# Confined fission track revelation in apatite: how it works and why it matters

Richard A. Ketcham[1], and Murat T. Tamer[1]

[1]Department of Geological Sciences, Jackson School of Geoscience, University of Texas, Austin, TX 78712, USA

*Correspondence to*: Richard A. Ketcham (ketcham@jsg.utexas.edu)

**Abstract.** We present a new model for the etching and revelation of confined fission tracks in apatite based on variable along-track etching velocity, $v_T(x)$. Insights from step-etching experiments and theoretical energy loss rates of fission fragments suggest two end-member etching structures: Constant-core, with a central zone of constant etching rate that then falls off toward track tips; and Linear, in which etching rates fall linearly from the midpoint to the tips. From these, we construct a

characterization of confined track revelation that encompasses all relevant processes, including penetration and widening of semi-tracks etching in from the polished grain surface, intersection with and expansion of confined tracks, and analyst selection of which tracks to measure and which to bypass. Both etching structures are able to fit step-etching data from five sets of paired experiments of fossil tracks and unannealed and annealed induced tracks, supporting the correctness of our approach and providing a series of insights into the theory and practice of fission-track thermochronology. Etching rates for annealed

induced tracks are much faster than those for unannealed induced and spontaneous tracks, impacting the relative efficiency of both confined track length and density measurements, and suggesting that high-temperature laboratory annealing may induce a transformation in track cores that does not occur at geological conditions of partial annealing. The model quantifies how variation in analyst selection criteria, summarized as the ratio of along-track to bulk etching velocity at the etched track tip ($v_T/v_B$), likely plays a first-order role in the reproducibility of confined length measurements. It also accounts for and provides

an estimate of the large proportion of tracks that are intersected but not measured, and shows how length biasing is likely to be an insufficient basis for predicting the relative probability of detection of different track populations. The $v_T(x)$ model provides an approach to optimizing etching conditions, linking track length measurements across etching protocols, and discerning new information on the underlying structure of fission tracks.

## 1 Introduction

Apatite fission-track confined lengths remain of great interest because of their capacity to record detailed thermal histories (Malusa and Fitzgerald, 2019; Gallagher 2012; Ketcham et al. 2018). However, our understanding of them remains incomplete in ways that are likely to be consequential for thermal history analysis. Measurements of laboratory-annealed spontaneous and induced fission tracks designed to test the principle of equivalent time, which posits that track annealing behavior is

determined by length alone and not prior thermal history (Duddy et al., 1988), indicate that their behavior subtly but certainly differs (Wauschkuhn et al., 2015). This divergence leads to continuing uncertainty on the fidelity of induced tracks annealed in the laboratory as proxies for spontaneous ones annealed at geological conditions over geological time scales. Additionally, the reproducibility of length measurements among laboratories has been disappointing (Ketcham et al., 2015;Ketcham et al., 2018). The first concern questions the theoretical basis for thermal history inversion, the second its practice.

Fission tracks form due to the transfer of energy from fission fragments into the surrounding crystal lattice, creating a zone that is up to 9 nm in diameter and ~21 µm long in apatite (Jonckheere, 2003;Paul and Fitzgerald, 1992;Li et al., 2012). The crucial property of fission tracks is their etching structure; the only reason we can detect and measure them at all is that they etch differently from surrounding, relatively undamaged material. The most influential aspect of this structure is the etching velocity along the track.

From the earliest days of fission-track dating (e.g., Fleischer and Price, 1964), the model emerged of a fast etching velocity along the track ($v_T$) versus a slow etching velocity of the bulk grain ($v_B$, also called $v_G$). This contrast allows, among other things, for the calculation of a counting efficiency to quantify what proportion of surface-intersecting tracks become unobservable due to bulk etching of the polished surface obscuring shallow-dipping tracks. At the same time, it carries the implication that $v_T$ is constant along a track; the efficiency equation put forth by Fleischer and Price (1964) and repeated many

times since (e.g., Fleischer et al., 1975;Hurford, 2019;Tagami and O'Sullivan, 2005) presumes single values for $v_B$ and $v_T$. A similar simplification is embedded in the characterization of confined track revelation by Laslett et al. (1984), in which fast along-track etching compared to bulk etching leads to track tips being hard to observe or measure reliably while incompletely etched, but once the end of the track is reached bulk etching allows the tip to widen and become clear. A linked concept is that of maximum etchable length. Virtually all mathematical treatments of track revelation, biasing, and the relationship

between confined track length and track density portray latent tracks as line segments in space, and presume that the probability of a track being measured is equivalent to its probability of being intersected by an etchant pathway (Galbraith and Laslett, 1988;Galbraith et al., 1990;Laslett et al., 1984;Laslett et al., 1982;Dakowski, 1978;Jonckheere and Van Den Haute, 1999;Ketcham, 2003). This simplification does not consider time, and effectively assumes that all tracks are etched to their full extents, or at least that all tracks are equally likely to become fully etched once they are intersected.

In this contribution, we demonstrate that these assumptions are incorrect, and that this shortcoming impacts apatite fission-track (AFT) thermochronology in multiple ways, from reproducibility of confined track length measurements to the efficiency of track revelation that underlies age determinations. We do this by constructing the first quantitative depiction of confined track-in-track (TINT; Lal et al., 1969) revelation, incorporating their along-track etching structure as constrained by a set of recently reported step etching experiments (Tamer and Ketcham, 2020a). The model incorporates both the etching of the

surface-intersecting semi-track channels and the confined tracks themselves, providing multiple insights into the nature of the confined track length distributions we measure and interpret.

## 2 Background

It has long been understood that etching velocity varies along ion tracks. Fleischer and Price (1964) mention the possibility
of etching slowing down toward track tips, and Fleischer et al. (1969) used diminishing along-track etching velocity to explain track geometries in track-recording plastics. Early work on using ion tracks for identification of cosmic ray particles (e.g., Green et al., 1978;Price and Fleischer, 1971;Price et al., 1967;Price et al., 1973) measured etch rates at varying locations along implanted tracks, linking them to the ionization rate, or the rate at which an ion transfers energy to the medium it is passing through. These studies established that if $v_T$ can be determined with sufficient precision at two points along a particle path,
the atomic number of the particle can be uniquely identified (Price and Fleischer, 1971).

As fission fragments pass through a solid and lose energy, their ionization rates fall. Figure 1 shows a sampling of possible fission pairs from induced fission of $^{235}$U in apatite, calculated using SRIM (Ziegler, 2013). The ionization rate is a function of the ion and its energy, and the enclosing mineral. The full range of the particle pairs varies from ~21-23 µm, but as the ionization rate falls below some limit the etching rate is no longer significantly enhanced, and so the etched track is shorter.
Although the details of each fission pair vary, with either the heavier or lighter fragment initially losing energy more quickly, there is a general pattern of relatively slow change in energy loss rate toward the center of a track, and faster as the enhanced etching limit is approached.

The inference that fission-track etching rates should vary along their length has been borne out by observation. TEM data show that ion tracks have diminishing cross-sectional area with increasing distance travelled through apatite and zircon (Li et
al., 2011), and it is reasonable to infer that etching rate is influenced by the latent channel width. Jonckheere et al. (2007) reported evidence for diminishing etching velocity toward confined track tips in apatite. More recent work has documented enhanced but continuously diminishing etching velocity in the region along tracks beyond that reached by a typical 20-second etch (Jonckheere et al., 2017).

Recent step-etching experiments (Tamer and Ketcham, 2020a) demonstrate that variations in etching rates extend well into the
interiors of tracks, and suggest that etching velocity should be treated as a continuous function, $v_T(x)$, where $x$ is along-track distance, rather than a single value $v_T \gg v_B$. Moreover, preliminary analysis of these data suggests that spontaneous tracks in Durango apatite have a significantly different etching structure than induced tracks lightly annealed to have a similar mean length.

Converting step-etching measurements of confined tracks into quantitative estimates of $v_T(x)$ is challenging, however, because
TINT revelation is a complex process. First, surface-intersecting semi-tracks must etch into the solid crystal. As they penetrate they also widen, and this widening is the process by which confined tracks are encountered and etched (Galbraith et al., 1990;Jonckheere et al., 2007;Ketcham, 2003); the latent tracks themselves have a maximum diameter of about 9 nm (Li et al., 2010;Paul and Fitzgerald, 1992), and thus only intersect when track density is extremely high (Ketcham et al., 2013). A confined track may be intersected by an expanding semi-track anywhere along its length, and its revelation rate will be a
function of the etching structure in both directions from that point. Finally, to be measured, a confined track must be etched

sufficiently to be seen and judged by the analyst as suitable for measuring. For routine AFT analysis, standard practice dictates that the track tips need to "fully etched," although this evaluation is analyst-specific. For measuring tracks in early steps of step-etching experiments, when all tracks are under-etched, the criterion is simply that a track and its tips be visible enough to make a reasonable measurement.

## 3 Data

The data analyzed in this study (Figure 2, Table 1) primarily consist of confined fossil and unannealed and annealed induced track lengths from a series of step-etch experiments in Durango apatite in which tracks were individually followed through each etching step (Tamer and Ketcham, 2020a). Step etching data of this sort, enabled by automated image capture measurement systems (Gleadow et al., 2009), provide much clearer information on track etching than traditional step-etch experiments in which tracks are randomly selected after each etching step (Aslanian et al., 2021;Jonckheere et al., 2017). We also use one single-etch-step measurement of spontaneous track lengths in Durango apatite from Tamer and Ketcham (2020b). The Tamer and Ketcham (2020a) data also include three "etch-anneal-etch" experiments on annealed induced tracks in which apatites were fully annealed (48 h at 450°C) after an initial etching step and then re-etched to obtain a robust estimate of the average bulk etching rate ($v_B$) at track tips of 0.022±0.004 µm/s/tip. There was no clear indication of $v_B$ varying with track orientation in these data. While perhaps unexpected in view of the anisotropy of etch figures (Burtner et al., 1994), this result is consistent with dissolution theory because confined track tips are concave and thus limited by the slowest crystallographic etching planes, as opposed to etch figures, whose boundaries are convex and limited by the fastest (Aslanian et al., 2021). We only use the first step in the etch-anneal-etch experiments in this study, as a 10-second etching experiment for their respective annealing states.

All experiments have been renamed from their original sources to make them easier to follow here. In the revised naming scheme, the first symbol (S or I) indicates spontaneous or induced tracks, and the second (U or A) indicates unannealed versus annealed by laboratory heating. For the annealed experiments, the following number indicates the temperature of 24-hour isothermal heating (235, 270, 280°C). The final hyphen and number indicate the duration in seconds of the first etching step in a series.

In addition to generally showing falling etching rates with increasing etch time and etched length for each track type, a cursory examination of the data in Table 1 and Figure 2 reveals a number of seemingly surprising results. The bulk etching velocity, indicated by the slopes of the dashed lines for the etch-anneal-etch experiments, is only achieved after 25 seconds in experiment IU-10, and not even after 30 seconds for IU-20. For both spontaneous and unannealed induced tracks, mean track lengths are 0.4 µm longer at 20 seconds in step etch experiments that began at 10 seconds compared to a single 20-second etch. Measured tracks in the three annealed induced experiments are much longer than fossil and unannealed induced tracks after 10 seconds, even though they are shorter once fully etched. Tracks annealed for 24 hours at 235 °C are only 0.4 µm longer when measured after 15 seconds compared to 10, but then grow by 1.4 µm between 15 and 20 seconds. In this study, we show that all of these

observations can be explained by a simple etching rate structure in the context of a full model of TINT revelation and measurement.

We have added to these data new measurements of the intersection points between the confined tracks and impinging semi-tracks, with the goal of evaluating model predictions for where TINT intersections occur. These measurements were made using the "cross section" tool in FastTracks software (v3), measuring the distance from each intersection to the uppermost tip of the confined track, and then converting that to true distance using track dip. In cases where it was difficult to determine if an intersection truly occurred due to interfering features, we included it. We took this as the conservative choice, as the

principal mechanism by which our etch rate determinations can be wrong is if multiple intersections lead to artificial, apparent acceleration of etching due to different parts of the track etching separately, and we wanted to examine the worst possible case.

## 4 The Model

This section describes our model for TINT revelation and measurement. Figure 3 shows our functional descriptions for $v_T(x)$ etching structure, and Figure 4 schematically outlines how we incorporate etching structure into our overall model of semi-

track penetration and expansion, followed by revelation, growth, detection, and measurement of confined tracks. Each aspect of the model is described in detail below.

### 4.1 Etching structure

The energy loss rates of fission particles (Fig. 1) suggest that etching rates decline continuously toward track tips, with the possibility of a region of slower change in the center of the track. Accordingly, we consider two simple end-member

possibilities for etching velocity structure (Figure 3), "Constant-core" and "Linear." Both are encompassed in Equation 1:

$$
v_T(x) = \begin{cases} v_{T_{max}} & x_{T_1} \leq x \leq x_{T_2} & (1a) \\ v_{Tmax} - A\left|x - x_{T_1 \text{ or } T_2}\right|;\ A = \left|\dfrac{\Delta v_{T_{max}-B}}{\Delta x_{T_{max}-B}}\right| & x_{B_1} \leq x < x_{T_1};\ x_{T_2} < x \leq x_{B_2} & (1b) \\ v_B & x < x_{B_1};\ x_{B_2} < x & (1c) \end{cases}
$$

The latent track is defined by a set of etching rates along $x$, with the starting point for etching, or point where the etchant pathway intersects the latent track, denoted as $x_{int}$. In the Constant-core model, the track middle has a constant etching rate $v_{Tmax}$ over length $\Delta x_{Tmax}$, beyond which etching rate falls at linear rate $A$ over distance $\Delta x_{T_{max}-B}$ until it drops by $\Delta v_{T_{max}-B}$ to $v_B$. Defining our coordinate system such that the track extends in the positive direction from one tip at $x=0$, we define

coordinates $x_{B_1}$ and $x_{B_2}$ to be the track tips, beyond which etching occurs at the bulk rate; and $x_{T_1}$ and $x_{T_2}$ demarking the central zone of maximum track etch rate. The Linear model is simply the special case where $\Delta x_{Tmax}$ is zero. We define the latent track length as the zone of enhanced etching velocity:

$$L_{lat} = \Delta x_{Tmax} + 2\Delta x_{Tmax-B} \tag{2}$$

Only simple models are justifiable at this point because of our limited number of experiments. The Linear model is simpler, but the Constant-core model includes the simplest possibility of constant etch rate for the entire track if $\Delta x_{Tmax-B}$ equals zero.

These models may be considered end-members of a more complex model in which reduction in etch rate slowly accelerates as the track tips are approached. For this initial effort, we neglect length and etching anisotropy, as well as other complexities such as the asymmetric nature of true tracks due to the unequal energies and masses of the fission particles, and the possibly discontinuous structure of tracks toward their tips. Figure 1 suggests that there are many etching structures depending on which atoms were generated by fission, so any single model can only be considered an average. Additionally, the etching

structure implied by Figure 1 only strictly applies to unannealed tracks, and we have no indication of how it evolves with annealing at various conditions, other than overall shortening.

To convert the etching structure to the time required to etch out to a certain length starting from a given $x_{int}$, we integrate Equation (1). For simplicity, we do this only for etching in the positive direction toward $x_{B_2}$, denoting that semi-length as $L_2$; its counterpart $L_1$ is calculated using the same set of equations by changing $x_{int}$ to $L_{lat} - x_{int}$.. Etching of the grain mount

commences at time $t=0$, and the confined track starts to etch at a later time $t_s$, to account for the time necessary to propagate and widen the impinging semi-track sufficiently to intersect the latent confined track. There are three cases to consider depending on where $x_{int}$ is: in the tip nearest $x_{B_2}$, in the constant-rate core, or in the tip nearest $x_{B_1}$. If $x_{int}$ is in the constant-rate core, the second case, then the time required to etch to a semi-length $L_2$ is:

$$t(L_2) = \begin{cases} t_s + \dfrac{L_2}{v_{T_{max}}} & L_2 \le x_{T_2} - x_{int} & (3a) \\[3mm] t_s + \dfrac{x_{T_2} - x_{int}}{v_{T_{max}}} + \displaystyle\int_{x_{T_2}}^{x_{int}+L_2} \dfrac{dx}{v_T(x)} & x_{T_2} - x_{int} < L_2 \le x_{B_2} - x_{int} & (3b) \\[3mm] t_s + \dfrac{x_{T_2} - x_{int}}{v_{T_{max}}} + \displaystyle\int_{x_{T_2}}^{x_{B2}} \dfrac{dx}{v_T(x)} + \dfrac{L_2 - (x_{B2} - x_{int})}{v_B} & x_{B_2} - x_{int} < L_2 & (3c) \end{cases}$$

After integrating, the solution becomes:

$$t(L_2) = \begin{cases} t_s + \dfrac{L_2}{v_{T_{max}}} & L_2 \le x_{T_2} - x_{int} & (4a) \\[3mm] t_s + \dfrac{x_{T_2} - x_{int}}{v_{T_{max}}} - \dfrac{1}{A}\ln\left(1 + \dfrac{x_{T_2} - (x_{int} + L_2)}{\frac{v_{T_{max}}}{A}}\right) & x_{T_2} - x_{int} < L_2 \le x_{B_2} - x_{int} & (4b) \\[3mm] t_s + \dfrac{x_{T_2} - x_{int}}{v_{T_{max}}} - \dfrac{1}{A}\ln\dfrac{v_B}{v_{T_{max}}} + \dfrac{L_2 - (x_{B_2} - x_{int})}{v_B} & x_{B_2} - x_{int} < L_2 & (4c) \end{cases}$$

Solving these equations for semi-length as a function of etching time then gives:

$$L_2(t) = \begin{cases} 0 & t \le t_s & (5a) \\[2mm] v_{T_{max}}(t - t_s) & t_s < t \le t_s + \dfrac{x_{T_2} - x_{int}}{v_{T_{max}}} & (5b) \\[2mm] x_{T_2} - x_{int} + \dfrac{v_{T_{max}}}{A}\left[1 - e^{-A\left(t - t_s - \frac{x_{T_2} - x_{int}}{v_{T_{max}}}\right)}\right] & t_s + \dfrac{x_{T_2} - x_{int}}{v_{T_{max}}} < t \le t_s + \dfrac{x_{T_2} - x_{int}}{v_{T_{max}}} - \dfrac{1}{A}ln\left(\dfrac{v_B}{v_{T_{max}}}\right) & (5c) \\[2mm] x_{B_2} - x_{int} + v_B\left[t - t_s - \dfrac{x_{T_2} - x_{int}}{v_{T_{max}}} + \dfrac{1}{A}ln\left(\dfrac{v_B}{v_{T_{max}}}\right)\right] & t_s + \dfrac{x_{T_2} - x_{int}}{v_{T_{max}}} - \dfrac{1}{A}ln\left(\dfrac{v_B}{v_{T_{max}}}\right) < t & (5d) \end{cases}$$

Solutions to the other two cases are provided in the Appendix.

Figure 5 illustrates lengthening curves for example Constant-core and Linear structures for unannealed induced tracks. Lengthening is mostly nonlinear, accelerating and decelerating depending on local etching structure and asymptotically approaching the bulk etch rate toward track tips. An immediately interesting outgrowth of this model is that etched track length after a given amount of time varies depending on where the track is intersected by the etchant pathway. Figure 5C,D shows the development of total confined track length with time etching depending on intersection point. Starting etching from the track center is most efficient, as equal etching can occur in both directions, and if intersection occurs toward one end the result is a shorter etched track. Variation in intersection point alone is likely responsible for some component of the observed variation in track lengths.

## 4.2 Semi-track penetration and confined track revelation

Once the track etching structure is defined, we can then describe how semi-tracks penetrate into the solid grain, etching downward and then expanding outward to intersect and reveal confined tracks. For internal surfaces (i.e. grains mounted and polished to remove ~10 µm or more of material), tracks will originate from fission events both above and below the polished surface, and will cross the polished surface at all possible angles. Although track orientations are completely random, their crossings are subject to two biases (Dakowski, 1978). First, the relative probability of a track of latent half-length $L$ crossing the surface plane will depend on track dip $\delta$ as $L \sin \delta$. Second, the relative abundance of tracks at dip $\delta$ will vary as $\cos \delta$, by analogy of the area between latitude lines on a globe, which diminishes toward the poles. Thus, the probability of a semi-track occurring will be proportional to $L \sin \delta \cos \delta$. The $x_{int}$ point at which the track intersects the polished surface is evenly distributed along its length. The semi-track penetration calculation thus consists of randomizing some number (typically $10^5$) of lengths, dips, and $x_{int}$ points (Fig. 4A), and then using the etching model to trace each semi-track's etching downward into the grain surface (Fig 4B-E).

We also consider the case of implanted ion tracks, whether from [252]Cf or a particle accelerator. Here the surface intersection angles and $x_{int}$ points are not random, but constant. As [252]Cf irradiation was used for our spontaneous track data, we use a mean [252]Cf semi-track length of 5.9 µm with standard deviation of 1.4 µm, based on unpublished measurements at the University of Texas. We also assume a dip of 75°; Jonckheere et al. (2007) point out that, to maximize etching efficiency, ion tracks should not be normal to the polished surface.

Confined track revelation requires that, once the semi-track has reached a given depth, it then begins to etch outward into the undamaged crystal at that depth (Fig. 4B), at the bulk etching rate, even as it continues to propagate downward. The probability of a semi-track encountering a confined track is proportional to the diameter of this etched zone, and increases as the track widens. Thus, the net probability of intersection of a confined track at a given depth $z$ during a given time step $t$ is proportional to the net growth in semi-track diameters $D_s$ normal to the confined track (Fig. 4C):

$$P(intersection|z,t) \propto \sum_s D_s(z,t) - \sum_s D_s(z,t-1) \tag{6}$$

Revelation is actually anisotropic based on the shape of the etch figures (Galbraith et al., 1990;Ketcham, 2003) or internal crystallographic planes (Jonckheere et al., 2019), but we omit this consideration for our simplified initial model, which neglects track crystallographic orientation.

Figure 6 shows examples of calculated penetration and revelation rate. For randomly oriented tracks on an internal surface (Fig. 6A, B), net penetration and revelation develop relatively slowly, as the majority of semi-tracks are at intermediate angles to the grain surface. It is also noteworthy that penetration is limited, and that ~10% of semi-tracks reach a depth of only 1 μm or less. Although the [252]Cf tracks are shorter and thus penetrate less deeply, as they reflect only one fission particle, penetration and revelation are relatively fast because of their consistent and efficient orientation (Fig. 6C, D).

**4.3 Confined track intersection**

The relationships represented in Figure 6B or 6D are used to calculate a cumulative distribution function (CDF) for intersecting possible confined latent tracks as a function of time and depth. We generate some number of latent tracks with either a single length, or a normal distribution defined by a mean and standard deviation (σ). Where we include the latter variation, the track etching structure scales linearly with length ($\Delta x_{T_{max}}$ and $\Delta x_{T_{max}-B}$), but $v_{Tmax}$ is left unchanged. For models shown here we use σ = 0.8 μm. We generate a distribution of dip angles ranging from 0 to δ$_{max}$, weighted as cos δ. For this study we use a δ$_{max}$ of 25°, near to the maximum measured in the Tamer and Ketcham (2020a) data set. We also randomize the impingement point ($x_{int}$) along each latent track, but exclude regions near the track ends, which can obscure the etched tip and make it unmeasurable. For the present work we set this exclusion region to be within 2 μm of each latent track tip.

Based on the length, dip and impingement point, some of the tracks generated intersect the grain surface, or in other words are semi-tracks themselves (Fig. 4E). We thus cull all tracks with an upper endpoint 0.5 μm or less below the surface to account for bulk etching down from the original surface and vertical widening of the track.

Figure 7 shows the result from an example model of $10^7$ track intersections generated over 20 s of etching. The contour plot of all intersections (Fig. 7A) shows that latent tracks are more likely to be intersected near the surface, and more likely to be intersected later in the etch than earlier. This result is a direct outgrowth of the revelation rate calculation (Fig. 6B); as time goes on, more semi-tracks are penetrating deeper and getting wider, increasing the probability of intersecting other tracks at depth. Because we generate interior tracks with dips, not all tracks intersected are confined tracks, however. Figure 7B shows that 58% of the generated interior tracks remain after surface-intersecting tracks are excluded, though the calculated proportion

should not be taken as an absolute, as it varies with $\delta_{max}$ because more strongly-dipping tracks are more likely to not be confined (e.g., Li et al., 2018).

## 4.4 Confined track selection

Once the impingement time and along-track location ($t_s$ and $x_{int}$) are generated for a confined track, its etched length through time is calculated. Because track intersection occurs continuously, etched lengths will range from negligibly short up to the longest track etched. However, not all of them will be selected for measurement. In order to test and calibrate the model against measured confined track lengths, is it necessary to estimate which confined tracks the analyst will see and measure, and which ones the analyst will miss or reject. We are not aware of any previous work on this topic, and so we created a set of criteria intended to broadly describe the two modes of track selection used by Tamer and Ketcham (2020a,b).

For experiments where the first or only etching step lasted 20 seconds, Tamer and Ketcham (2020a,b) measured confined track lengths as would be done for normal AFT analysis. In standard practice, the analyst aims to measure only tracks that are "fully etched" or, as proposed by Tamer and Ketcham (2020a), "sufficiently etched" according to the analyst's judgement. Making this determination is a matter of training and experience, however, and it is to a significant degree arbitrary, although a good analyst tries to maintain consistent criteria. It is based primarily on the appearance of the track tips, which will develop as a function of the etching velocity along the track, or $v_T(x)$.

For convenience, we define the tip etching state in terms of the ratio $v_T(x)/v_B$ (or, more briefly, $v_T/v_B$) at the etched tip, though the actual tip state will also depend somewhat on the slope of $v_T(x)$ leading up to the tip. We further propose that, to first order, each analyst has a characteristic $v_T/v_B$ for tracks that they decide are sufficiently versus insufficiently etched. In our model each track is evaluated based on the less-etched tip, with the larger $v_T/v_B$ value.

Figure 8 shows a rough approximation of the evolution of a track tip through a series of etching times, using $v_T(x)$ for along-track etching and $v_B$ for etching perpendicular to the track. The calculation is simplified compared to recent work that endeavors to incorporate the detailed etching structure caused by internal crystal lattice planes and the angle of the track (Aslanian et al., 2021;Jonckheere et al., 2019), but it is a reasonable depiction of a track at ~45° to the **c**-axis (see. Fig. 9). As $v_T(x)$ falls along the track, the tip widens and becomes more distinct well before the true end of the track is reached. When $v_T/v_B$ finally falls to a value of 1.0, the track is bulbous by normal AFT analysis standards, and most, perhaps all, analysts would judge the track as sufficiently distinct to measure at some earlier stage. Given the large length change with even subtle changes in track tip shape, more than 2.2 µm as $v_T/v_B$ goes from 11 to 1, it is evident that analyst judgement in tip evaluation can be a first-order factor in explaining inter-analyst variation in track measurement.

For step etching experiments with first steps shorter than 20 s, precise location of the tip is not a prerequisite for track selection, which is instead a matter of simple visibility. In the earliest stages of etching, tracks will be too thin to be observable in visible light. As they grow, they become more efficient at reflecting light, making them more detectable. However, precisely when this occurs in practice is unclear.

The photomicrographs of developing etched tracks in Figure 9 illustrate both cases. After a 10-second etch, most tracks are barely visible in still images, and are mostly found by careful searching and racking the microscope focus to look for linear features with discernible tips (see supplemental information for Tamer and Ketcham, 2020a, for an animation). Although such tracks are prone to a larger measurement uncertainty, our etch-anneal-etch experiments, which found constant etching rates in two etch steps after annealing tracks at the 10-s etching stage (Fig. 2), found no significant evidence of a slightly etched but

still invisible track beyond the visible tips. At later etching steps, whether a particular track is clear enough to measure is subject to analyst judgement, although they continue to get significantly longer with each step.

Lacking a physical basis for determining when briefly-etched tracks begin to become visible, and how likely they are to be seen by the analyst, we examined the shortest tracks measured during the initial etching steps, and by trial and error constructed an empirical two-component operator bias function for the probability of an etched track being measured. Etching time must

be greater than 3 s, after which the probability of selecting a track is $((L - 2)/7)^3$; no tracks are selected below 2 µm and all tracks are selected by 9 µm, with a power-law increase in probability between those lengths. We stress that this latter formulation only applies to tracks in their initial stages of etching, and not to truly short tracks that etch for enough time to widen.

Both observability criteria and their outcomes are demonstrated in Figure 10. Figure 10A shows the short-first-step probability

of measurement, and Fig. 10B shows the predicted length distributions for unannealed induced tracks after 10 s (corresponding to experiment IU-10), and the subset of tracks that are selected, which is very similar to the measured distribution (Fig. 10C). Figure 10D-F shows the corresponding case of standard track selection after a 20 s etch (experiment IU-20). The tracks have a range of tip development (Fig. 10D), and selecting only those with $v_T/v_B \leq 12$ (Fig. 10E) results in an excellent match to the measured data (Fig 10F). The model histograms (Fig. 10B, E) also provide an indication of how many, or few, confined tracks

are selected relative to how many are intersected and etched in total.

In addition to lengths, the model also predicts the intersection depth and etching time distribution of selected tracks. Figure 11 shows the distribution of modelled track lengths and etching times with depth below the surface of the grain mount for unannealed induced tracks. It is clear that most selected tracks are intersected close to the surface, as that is the area best sampled by widening semi-tracks, and substantial time is required to sufficiently etch a track once it is intersected, which will

be a function of the along-track etching structure (Fig. 11A). There is a modest decrease in mean length with depth (Fig. 11B) owing to deeper tracks on average taking longer to be intersected.

**4.5 Fitting step-etching data**

Finding an etching structure that allows model predictions to match the experimental data consists of posing model parameters $(v_{T_{max}}, L_{lat}, \Delta x_{Tmax})$ and using them to construct first a set of penetrating semi-tracks and then a distribution of confined track

lengths. For multi-step experiments, confined tracks selected after the first etching step are then allowed to lengthen through subsequent etching steps. The resulting mean track lengths are calculated, and compared with the measured mean lengths. The reduced chi-squared ($\chi^2_v$) value is used to evaluate sets of model results against measurements for a given track type.

The step etch experiments only consist of 3-5 steps, making it difficult to meaningfully constrain models with 2-3 variables defining etching structure (Eq. 1), in addition to ancillary factors such as analyst selection criteria, with a single experiment.

Accordingly, we simultaneously fit pairs of data sets with equivalent tracks (e.g., IU-10 and IU-20), so that a single etching structure would have to explain two etching schedules. To minimize the effect of the ancillary factors, we used the same ones for all model fitting. After several trials, we settled on a $v_T/v_B$ of 12, as well as the brief-etch analyst bias function and values for minimum tip depth and near-tip exclusion zone for impingement, based on their ability to fit the IU data. The $v_T/v_B$ value is broadly constrained (range $\pm$ ~2) by the mean length measured after the 20-s etch (15.8 µm) versus the length when etching rate reaches $v_B$ (~17 µm). The analyst biasing criteria essentially truncate the short part of the track length distribution for the first etching step, while the etching structure and latent track length standard deviation combine to define the longer part of the distribution (Fig. 10B, E). Both have to fit, or compensate each other, to match the measured mean track length data. The set of tracks selected after the first etching step must then evolve appropriately to match all subsequent etching steps. Other ancillary factors such as $x_{int}$ and $\delta_{max}$ generally have small effects, changing mean confined lengths by 0-0.2 µm when varied within reasonable limits.

During fitting of the unannealed induced track data, it became apparent that one pair of data points exerted a disproportionate control on the result. The 15-20-s and 20-25-s steps for experiment IU-10 feature very similar mean length increase (Fig. 2), and thus almost the same etching rate, which in turn was much slower than the rate for the 10-15-s etching step. Because our etching structure equations assume that once etching rate begins to fall, it decreases linearly, fits were forced to minimize the inevitable misfit with these two measurements to the exclusion of closely fitting the rest of the data. We thus excluded the 15-s measurement for IU-10, effectively making the second etching step go from 10 s to 20 s. Doing so lowered the fitted $\chi^2_v$ values for IU from ~7 to ~2, while predicting a mean length 0.8 µm shorter than the omitted measurement. We also tried instead excluding the 10-s measurement for IU-10, but that provided a smaller improvement in $\chi^2_v$ to ~5.6, with a marginally higher maximum etching rate of ~2 µm/s.

Model fitting is complicated by each calculation of predicted track lengths incorporating several randomizations: semi-track orientation and surface intersection points; internal track dips, depths, and intersection points; and which short tracks are observed. As a result, the same set of model parameters generates slightly different predictions with each model run, and $\chi^2_v$ values vary by several percent even when simulating $10^6$ tracks. There is thus no true minimum to converge to, making it difficult for iterative search methods to avoid temporary local minima. We thus fitted the models using the downhill simplex method (Press et al., 1988) from multiple starting points, stopping each run after testing 50 models, which by inspection was a sufficient number for the algorithm to converge toward a local minimum and no longer significantly change parameter values being attempted. For Constant-core models, the three model parameters (latent track length, core zone length, and maximum etching velocity) showed a high degree of correlation and broad $\chi^2_v$ minima, requiring running many more simplex instances from different starting points to trace out these correlations.

The parameter sets we report are those that achieved the lowest $\chi^2_\nu$ for each data set, but as discussed in the previous paragraph they are unlikely to be the true optima, and different parameter sets may be able to reach lower scores. Due to these complexities, we define confidence intervals by running 20 repetitions of the lowest-$\chi^2_\nu$ parameter set for each data set, determining the mean and relative deviation of $\chi^2_\nu$. The confidence intervals include all parameter sets tested during the simplex runs with $\chi^2_\nu$ values within two deviations of the mean. However, any result with a $\chi^2_\nu$ of around 1 or less can be

considered a reasonable fit to the data.

## 5 Results

Table 2 lists the fitted etching structure model parameters and confidence intervals, along with the mean $\chi^2_\nu$ values and relative deviations, and the right-hand columns of Table 1 provide the model predictions. The standardized residuals for each fit to each data set (Figure 12) show that almost all experimental results were reproduced to within 2 standard errors, and there are

335 no indications of any systematic patterns in the misfits. Figures 13 and 14 show the results of parameter fitting for Constant-core and Linear models, respectively. Constant-core models tend to support a long stretch of solutions trading off core length for maximum etching velocity that in most cases approaches or reaches to a core length of zero, the Linear model case. Constant-core model results for latent track length are relatively stable with respect to the other parameters, however, and predict ~0-0.1 µm shorter latent track length than Linear models, with higher divergences for longer cores. Linear model fits

show a slight correlation between maximum etching velocity and mean latent track length for the unannealed experiments, but little to no correlation for the annealed experiments.

Only the fit to the SU data implies significant support for the Constant-core etching structure over the simpler Linear one, and even in that case the Linear model still achieves an $\chi^2_\nu$ of 0.5 indicating that it fits the data to well within the measurement uncertainties. We can thus say that, to within the resolution of our data, the Linear structure is adequate for all cases that we

tested, and that all of our data point to a decline in etching velocity from well into the track interior to the track tips.

The starkest difference in etching structure is evident in Figure 15, showing the best-fit results for both model types. Unannealed induced and fossil tracks both have slow-etching central regions, whereas all samples that underwent laboratory annealing have roughly double the maximum etching rates in their centers. Due to their also being shorter, the fall-off in etching velocity from track centers to tips occurs at a ~2.5x faster rate for the annealed experiments compared to the unannealed

ones.

Table 1 also lists the number of impingements per track after the final etching step in each experiment, and supplemental Figure S1 shows histograms of their measured depths. Generally, data corroborate modelling results that most TINT intersections occur close to the polished surface, with the mean depths in all cases being less than 5 µm. The number of intersections per confined track is unusually high due to the large induced track densities, and Cf irradiation for the SU

experiments. We do not know precisely when these intersections occurred, however, and in particular whether a given semi-track intersected a still-latent part of the confined track, or one that was already etched.

## 6 Discussion

Our model of track revelation incorporating a simple along-track etching structure is able to reproduce our step-etching data closely using a consistent set of assumptions about track detectability and selection, and explains the otherwise odd-appearing features of the data in Figure 2 mentioned previously. For example, our results show that the 0.4-µm increase in mean track length after 20 seconds of etching in experiment IU-10 versus IU-20, and SU-10 versus SU-20, is a direct result of tracks being selected after 10 seconds, leading to the measured tracks being etched an average of ~2 seconds longer than if they are selected after 20 (Table 1; Fig. 11A). Similarly, the unexpected observation that annealed tracks (IA235-10, IA270-10, IA280-10) are longer than unannealed ones (IU-10, SU-10) after 10 seconds of etching is explainable by a change in etching rates that is consistent with measurements after longer initial etching steps. The small increase in mean length for IA-235 after a 15-s versus 10-s initial etch, followed by a much larger increase between 15 and 20 seconds after step etching, is consistent with the analyst selection model employed across all experiments (Fig. 10). We take these successes as an indication of the overall validity of our characterization of confined track etching.

The primary mechanism by which our derived etching rates may be grossly incorrect is the possibility that multiple impingements to the same tracks caused faster etching. While this is a concern, we do not believe that our rates are significantly affected. Our slowest-etching sample, SU, has about that same number of impingements per track as the 10-second experiments that help establish the fast early etching rates for the annealed experiments, and more impingements per track than the faster-etching IU. For an additional impingement to accelerate track revelation substantially, it would probably need to intersect an as-yet-unetched part of the track, and thus must happen within a few seconds of the first one, whereas impingements with no effect can occur at any time. Impingements also become more likely when both the semi-track and confined track are widening, in essence growing to meet each other. Additionally, our intersection numbers are probably high, as we counted all likely impingements, but limited optical data resolution can make it difficult to determine whether there is a true impingement or a near miss. When we regress the number of intersections against mean track length (Fig. S2), the annealed experiments range from 0.04 to 0.21 µm of additional length per intersection; even after removal of such an effect, our annealed track measurements would be significantly longer than unannealed ones after 10 seconds of etching. In sum, our high intersection numbers may have resulted in some over-estimation of the etch rates of our annealed track experiments, but not enough to change the broad conclusions we draw from those data.

Another possible weakness in our results is that the 10-s measurements by Tamer and Ketcham (2020a) are more likely to be erroneous due to the tracks' poor visibility, as the true etched ends of the tracks may not have been wide enough to be seen with optical microscopy. In the experiments with annealed induced tracks that underwent etch-anneal-etch examination, we are confident that this was not an issue: only a handful showed evidence of enhanced etching after the annealing step, which were not enough to affect the average bulk etch rates. However, the longer unannealed induced and fossil tracks could have had a greater incidence of this effect; such a phenomenon might partially underlie our removal of the 15-s experiment from

fitting, although we tested for this by trying to remove the 10-s result instead. Even in this case, the result remains consistent with all of the outcomes discussed below.

### 6.1 Etching structure model form

The success of both the Constant-core and Linear etching structures in fitting the data clearly indicates that etching rates start diminishing well before the tips are reached. For both unannealed and annealed induced-track experiments, there is little to no improvement of fit with the more complex Constant-core structure, whereas for fossil tracks the Constant-core structure provides a better fit, but a Linear one still fits to well within the uncertainty of the measurements. We take these results as an indication the Linear structure adequately describes our data, but further or better measurements may reveal an etching structure similar to that implied by the theoretical energy loss profiles (Fig. 1) for some track types. Etching rate may be linked similarly to latent track cross section diameter, which gradually shrinks from the fission site to the tips (Li et al., 2012). The TEM data suggest the existence of an inflection point from slower to faster diameter reduction, which a linear combination of the Linear and Constant-core models would roughly approximate. However, such a model would take four parameters to define (maximum etch rate, etch rate at inflection point, width of interior zone, latent length), which is too complex for the limited amount of data we have, and thus we prefer simple models at this stage. Smooth functions are of course also possible, and arguably preferable, but would also require as many or more degrees of freedom. We finally note that the variety of fission products implies that there will also be a variety of etching structures (Fig. 1), so any single structure can only represent an average. It is possible that such variation is responsible for our poorer IU data fit, and that annealing leads to a convergence of etching structure.

It is also possible that falling directly to our measured $v_B$ is an oversimplification of the latent track tip, and that etching velocity could have a sigmoidal form that asymptotically approaches a limiting and lower bulk value due to sporadic damage remnants, as postulated by Jonckheere et al. (2017) and Aslanian et al. (2021). We cannot rule this possibility out, but consider it likely to have at most a minor influence on practical observations, as they needed to etch for more than 40 seconds to reach inferred etching rates for Durango apatite lower than our value for $v_B$.

### 6.2 Fossil versus induced tracks

Fossil and unannealed induced tracks have slow core etching rates in our experiments, while all of our annealed induced experiments feature far higher rates. This implies that the high temperatures used for laboratory annealing experiments may reorganize the atoms in the track core in a way that does not occur during geological low-temperature annealing. Such a phenomenon could be responsible for some component of the mismatch in annealing fossil versus induced track annealing behavior in laboratory experiments (Wauschkuhn et al., 2015), although we do not yet have data to gauge whether high-temperature annealing of fossil tracks has the same effect as on induced ones. Additionally, insofar as the annealing that affects lengths takes place at track tips, the significance of what happens in the core region is unclear. Nevertheless, our result corroborates that there are differences between fossil and annealed induced tracks, which may imply a violation of the principle

of equivalent time, that "a track which has been annealed to a certain degree behaves during further annealing in a manner which is independent of the conditions that led to the prior annealing" (Duddy et al., 1988).

Although Duddy et al. (1988) experimentally validated the principle of equivalent time, they could only do so on annealed induced tracks. Contemporaneous experiments (Green, 1988) also supported the equivalency of track types and annealing
modes, by providing evidence that spontaneous track lengths should be normalized against induced ones, to in turn provide a normalization for spontaneous track density. Contrary to these studies, the more specifically designed experiment by Wauschkuhn et al. (2015) documented a case that demonstrates non-independence of prior annealing conditions – tracks annealed geologically to ~14 µm do not behave during further annealing equivalently to tracks annealed in the laboratory to ~14 µm. However, this observation has only been made in a single apatite type, and the degree of disagreement is limited to
1 µm or less. Thus, the extent to which such a departure affects practical usage is unknown, and the generally observed agreement between interpretations based on apatite fission track and other thermal history indicators suggests it is minor, or within the noise of the method as it is currently practiced. As the community works to improve the method and reduce the noise, as well as to develop a better physical understanding of fission-track annealing, however, this divergence will deserve continued attention.

**6.3 The line segment model and length bias**

Mathematical treatments of fission-track lengths are all based on a line segment model (Laslett et al., 1982;Parker and Cowan, 1976;Galbraith, 2005), which posits latent tracks as line segments within a volume that are detected and become fully etched when intersected. One outgrowth of the line segment model is length biasing; the probability of intersecting and thus detecting and measuring a latent track is proportional to its length. This study shows this to be an oversimplification. Many more tracks
are intersected than measured (Fig. 10), because of the significant time required for a track to etch. The controlling parameter on detectability then becomes how quickly they become sufficiently etched to be accepted by the analyst, which in turn depends on length, etching velocity structure, and analyst criteria. Figure 16 shows example model prediction of tracks based on IU (unannealed) and IA-280 (most highly annealed) induced tracks, both modeled as being revealed using Cf semi-tracks to give a common baseline for track revelation. Only 19% of the confined unannealed tracks end up being selected, due to slow
etching requiring at least 8 s and on average 14 s for tips to be sufficiently revealed (for $v_T/v_B = 12$). Because the annealed tracks etch more quickly and don't have as far to etch, requiring as little as 4 s and averaging 11 s, 59% of confined tracks intersected are sufficiently etched to be selected. If we take into account the 2-µm exclusion zones at each track tip, the line segment model states that the annealed tracks are only 65% as likely to be intersected as the unannealed ones. This remains a source of bias, but even after incorporating this intersection probability, the annealed confined tracks are approximately twice
as likely to be selected for measurement as the unannealed ones.

This is an extreme example, as we have no data yet that indicates that fossil tracks with varying levels of annealing have this degree of etching rate variation, but it demonstrates that etching properties can be far more influential than length biasing in determining the relative probability of track measurement. Length biasing is a central assumption embedded into track length

modeling (Green et al., 1989;Ketcham, 2005;Willett, 1997), as it determines how to construct a combined track length distribution from individual populations of tracks generated throughout a geologic history. A substantial change in inter-population biasing will alter the shape of time-temperature paths by changing the proportion of time spent at different temperatures required to generate a given observed track length distribution. A thorough re-evaluation of this biasing based on measurements of fossil track etching structure is thus a necessity for setting such modeling on a firmer foundation.

## 6.4 Sources of analyst variability

Our model illustrates how the measured length distribution for a single population of tracks is controlled on the long side by the maximum extent of etching, and on the short side by the analyst's selection criteria. This idea leads to a very compelling explanation for the observed variability in inter-laboratory studies: different judgement of track tips. In their international, inter-laboratory exercise, Ketcham et al. (2015) reported variation in measured track length for unannealed induced tracks in Durango apatite across all levels of analyst experience, and without any obvious linkage to specific etching protocol (Fig. 17).

Table 3 and the histograms at the bottom of Figure 17 show the corresponding model predictions, varying only $v_T/v_B$. Virtually the entire range of the data can be explained by this one parameter; 68% of results lie between the predictions for 16 and 4, and 30% between 12 and 8. The median track length measurement was 15.9 µm, which corresponds to a $v_T/v_B$ value of ~11. As $v_T/v_B$ rises, the proportion of accepted tracks increases, and the mean track length falls; essentially, relaxing tip selection criteria makes it easier to measure a large number tracks. Conversely, using restrictive criteria can strongly reduce the number

of tracks measured, but also reduces scatter, making the information in the individual track lengths less ambiguous. Figure 18A shows the observed relationship of mean length to standard deviation for the Ketcham et al. (2015) exercise. The increase in standard deviation predicted by the model matches the data quite well. The model prediction, and the data, also indicate that scatter does not start increasing quickly until the mean length falls below 16 µm, or a $v_T/v_B$ value of about 8. Figure 18B and Table 3 show that the predicted number of tracks that meet a given $v_T/v_B$ standard begins to rapidly rise as the standard is

relaxed; for example, a $v_T/v_B$ standard of 12 provides ~2x as many tracks as a $v_T/v_B$ of 8, and ~14x as many as a $v_T/v_B$ of 4. Because having a high number of confined tracks is better for thermal history modeling, there is a practical incentive to adopt criteria that will provide reasonable numbers of tracks for a reasonable amount of effort. This may create a tension between having demanding tip criteria to ensure tracks are all "fully etched," versus accepting a lower level of etching that provides more tracks. One way to ameliorate this tension is with [252]Cf or ion track irradiation, which makes it easier to achieve high

numbers while retaining very demanding tip criteria. However, we note that even the longest measurements reported by Ketcham et al. (2015), likely reflecting the most rigid criteria, still do not reach the full mean latent track length of 17 µm indicated by our data and model.

Another option is to maintain very consistent criteria, even at a diminished level of tip etching. The considerable variation observed among even experienced analysts in the Ketcham et al. (2015) exercise likely represents this occurring, but at

differing degrees of etching, in various lab groups. The model introduced here potentially provides the capability to evaluate

the trade-offs between efficiency and dispersion. However, to do so, more information is required to better document effects of track angle, apatite solubility, and etching rates of fossil tracks.

Yet another avenue to improvement may be to utilize other measurements that can be made on tracks, such as tip shape (through image analysis) or track thickness (Aslanian et al., 2021), to evaluate degree of etching, and construct a suitable compensation factor. Indeed, ultimately it may be possible to use image analysis or machine learning to evaluate degree of track etching, and by so doing extract more precise length and thus thermal information. Again, many more measurements will be required, and the context provided by an etching model will be important to any such effort.

Analyst variability in tip evaluation may also underlie the different trends seen in length versus $\mathbf{c}$-axis angle ($\phi$) distributions among analysts (Ketcham et al., 2015;Ketcham et al., 2007a). Because etched track morphology changes with $\mathbf{c}$-axis angle, and because tip visibility diminishes in low-angle and very high-angle tracks, different choices made by analysts on which of these tracks to accept versus reject may be responsible for the different observed trends in $L_c$ vs. $L_a$ (the $\mathbf{c}$-axis and $\mathbf{a}$-axis intercepts for ellipses fit to length versus $\phi$). Moreover, changes in etching structure and thus tip revelation between annealed induced and fossil tracks could shift the appropriate $\mathbf{c}$-axis projection between these track types.

What is the best way to overcome analyst variability? One avenue may be improved training and community consultation so all agree on what a sufficiently etched track is. Such a prescription may be tricky, however, as tip appearance depends on multiple factors, especially apatite composition but also including mount preparation, polishing, and cleaning technique, etching protocol, microscope optics, and captured image quality, which will vary over the community and can be expected to improve over time. An alternative may be to use measurements of standards to estimate an analyst's $v_T/v_B$ (or some other indicator quantifying selection criteria), and use that as a more informative renormalization parameter than simply the unannealed induced track length. Whereas length normalization consists of simply dividing mean lengths by an initial length, utilizing $v_T/v_B$ may provide a means to account for how different tip evaluation criteria affect measurements differently at various levels of annealing.

There is also the question of whether and how etching procedure matters. Etchant strength affects etch figure shape, and must affect both along-track and bulk etching rates, perhaps by different factors for each. We note that the overall schema that we are proposing contradicts some aspects of our own recent interpretation of why two major etching protocols (5.5 M $HNO_3$, 21°C, 20 s vs. 5.0 M $HNO_3$, 20 °C, 20 s) produced different results in a detailed comparison by two analysts (Tamer et al., 2019). We interpreted this to reflect that the weaker etchant and lower temperature resulted in more under-etched tracks, leading to shorter mean track lengths. However, the $v_T(x)$ model makes clear that under-etched tracks are always present; it is merely a question of whether they are selected. Whether more under-etched tracks were chosen because better-etched tracks were more uncommon, or because the different etchant subtly affected their appearance, or because of some other factor, we cannot establish at this time. It may be reasonable to infer, however, that maintaining consistent selection criteria is more important than consistent etching procedures for making reproducible measurements.

## 6.5 Normalization and annealing models

Ideally, divergences in analyst measurements can be overcome by normalizing them to some standard, usually the measurements that underlie the annealing model used to interpret their data (Barbarand et al., 2003;Carlson et al., 1999). Typically, this normalization consists of dividing by a determination of initial track length ($L_0$; unannealed induced tracks) by the analyst, employing the same procedures used for unknowns (e.g., Green et al., 1986). The results of this study, however, suggest that this approach may be oversimplified and in need of improvement. Table 4 contains model predictions for three track types across three $v_T/v_B$ selection criteria. The predicted mean track length for unannealed induced tracks (IU) ranges from 15.7 to 16.3 µm after 20 s of etching as selection criteria grow more strict. Using these values to normalize the unannealed spontaneous track measurements works very well ($L/L_0$ = 0.912-0.915). However, due to their faster etching rates, the predicted lengths of annealed sample IA280 are much more stable (12.4-12.5 µm), and the normalization actually destabilizes them ($L/L_0$ = 0.766-0.790). There is some indication of this phenomenon in the sample from the Ketcham et al. (2015) inter-laboratory study with the most similar level of annealing, DUR-1. Agreement in raw length measurements was arguably best for that sample among the four in that study, and normalizing by initial length alone actually increased the scatter among several laboratories (compare Fig. 3B and 7B in Ketcham et al., 2015). Convergence was maximized only after also normalizing for track angle using **c**-axis projection (and omitting results for which no angle measurements were provided).

It may be that etching rates remain low in annealed fossil tracks, reducing this divergence in behavior and improving the performance of regular normalization for geological investigations. However, this remains to be established via measurements. Even so, an improvement in cross-normalization of measurements of annealed induced tracks would be valuable, as a means of increasing the inter-compatibility of length measurements across different experimental annealing studies. Mostly minor but occasionally consequential differences in annealing temperatures predicted by the Ketcham et al. (1999) annealing model based on Carlson et al. (1999) measurements, versus the Ketcham et al. (2007b) model which combined these with the Barbarand et al. (2003) results, may be due to oversimplified normalization.

The clearer picture of track structure provided by the $v_T(x)$ model may improve our understanding of annealing and annealing models in other ways as well. Our derived true mean latent track lengths in Durango apatite (~17.0 µm induced and ~15.6 µm fossil) are significantly longer than measurements obtained with 20-second protocols; more than 1 µm for fossil and unannealed induced tracks measured in this study, and 0.7-1.2 µm in Durango apatite measurements in the experimental data sets used for annealing models.

These differences in latent versus measured lengths in unannealed and lightly geologically annealed track populations highlight a potential shortcoming in the etching procedures employed for the past few decades. Few if any such tracks are fully etched, as etching is halted when track revelation is still somewhere in the decelerating zone. This in turn amplifies the consequences of analyst disagreement about etching extent. By stopping etching as soon as the curve of length versus time in step-etch experiments is passed and a linear zone assumed to reflect the bulk etch rate is reached, the community has essentially set up

camp on the edge of a cliff.  A change in etching procedure to allow tracks at all levels of annealing to etch more completely may be worth considering.

## 6.6 Optimizing the etching protocol

The $v_T(x)$ model provides a quantitative framework for evaluating whether the etching procedures used today are the most effective for the goal of providing high numbers of reproducible and informative confined length measurements to constrain
thermal histories.  For example, the model makes clear that the longer the etch, the more tracks that fulfill a given tip clarity criterion.  As etching continues, fully-etched tracks may become over-etched, though at a rate defined by $v_B$, which may be obtainable from etch figure measurements parallel and perpendicular to the **c** axis, $D_{par}$ and $D_{per}$ (Tamer and Ketcham, 2020a). For $v_B$=0.022 µm/s, five extra seconds of etching leads to only a 0.22 µm increase in length for a previously fully-etched track, which is close to the resolution limit of individual track length measurements.  Non-fully-etched tracks will lengthen somewhat
more, depending on the etching rates toward the tips.  However, because the "short side" of the track length distribution is defined by analyst selection criteria, if these criteria are held consistent, the change in mean length will always be diminished. This suggests that over-etching may not be a significant concern, except to the extent that it makes a grain more difficult to measure due to the enlargement of multiple etched features, and that it has the advantage of increasing the number of well-etched confined tracks available.

We test this proposition using the modeling shown in Table 4 and Figure 19 to estimate what occurs with 25 s of etching rather than 20 s, for various annealing states and selection criteria.  Using the same selection criterion ($v_T/v_B = 12$), mean unannealed track lengths rise by up to 0.3 µm, but the number of selectable tracks more than doubles for the slowest-etching tracks (spontaneous).  Standard deviations rise by less than 0.1 µm. One can even become more restrictive with etching criteria ($v_T/v_B = 8$) and still have more tracks to measure.  Of course, changes in etching protocol cannot be considered from the standpoint
of length measurements alone, as there are also potential effects on track density measurements.  In particular, it would necessitate a change in zeta calibration factor, and could also impact the ability to measure tracks in apatite with higher solubility and larger etch figures.

The proportion of selectable annealed induced tracks (IA-280) increases only modestly with additional etching time, because the faster etching rate and shorter lengths combine to make the baseline selection efficiency much higher.  However, if the
faster etch rates are a result of laboratory annealing, and naturally annealed tracks have low etching rates more similar to SU, then the potential benefit of increasing etching time will be larger.  Furthermore, if that is indeed the case, the fact that a thermal treatment may significantly increase etching rates and thus revelation efficiency suggests that a carefully controlled preheating step might also be a means of greatly increasing confined track numbers, potentially without affecting lengths and thus paleothermal information.

## 6.7 Counting efficiency

The semi-track penetration component of the $v_T(x)$ model (Fig. 6A) also provides some insights into track counting efficiency, and thus age determination. Efficiency in this context is the measured track density divided by the true track density, or the proportion of tracks crossing the polished surface that are detected. Jonckheere and Van den Haute (1996) denote efficiency with two variables, $\eta q$, to reflect that it contains both geometric ($\eta$) and observer ($q$) components that are difficult to disentangle. Generally, track etching rates relative to bulk etching rates are too high in apatite to apply the geometric critical angle equation of Fleischer and Price (1964), $\delta_c = \sin^{-1}(v_B/v_T)$, where $\delta_c$ is the dip below which tracks become undetectable because surface etching is faster than track etching. The implied efficiency, $\eta = 1 - \sin^2\delta_c \approx 1$ for $v_T \gg v_B$, is far higher than the generally observed factor of ~0.9. Jonckheere and Van Den Haute (2002) propose instead the concept of a critical depth, $z_c$, defining the degree of penetration into the polished surface a track must achieve to be observed, distinguished from other features, and counted with confidence by the analyst. They estimate $z_c$ values of 0.8 μm for apatite and 0.5 μm for muscovite detectors in their data, and propose that efficiency may vary with track length in a way that compounds the effect of length in the first-principles fission track age equation.

The $v_T(x)$ model corroborates and extends these conclusions. Figure 20 shows the near-surface portion of the penetration model (Fig. 6A) for fossil and unannealed and annealed induced tracks, and compares them all after 20 seconds. The rapid falloff in penetration reflects tracks that originated above the polished surface, and only extend a short distance into the grain below it. The spacing of the one-second contours reflects the mean etching rate of track tips at depth through time; the slower etching unannealed tracks (Fig. 20A, B) penetrate more slowly than the fast-etching annealed tracks (Fig. 20C, D), but eventually all cases converge to closely-spaced contours reflecting bulk etching. The divergence of the 20-second lines (Fig. 20E) reflects a combination of the different mean track lengths and the etching velocities. If we assume a $z_c$ of 0.8 μm, the implied $\eta q$ factors are 0.918 for IU and IA235 ($\bar{L}$ = 15.8 μm and 15.0 μm, respectively), 0.912 for SU ($\bar{L}$ = 14.4 μm), and 0.902 for IA280 ($\bar{L}$ = 12.3 μm). With further annealing the disparity in efficiency will grow, with estimated $\eta q$ falling to 0.8 at 7.7 μm mean track length (Jonckheere and Van Den Haute, 2002). Essentially, shorter track lengths are less efficiently counted because a higher proportion of those that cross the polished surface do not penetrate it sufficiently to be detected reliably. This effect is superimposed on the already understood reduction in track density due to shorter tracks being less likely to cross the polished surface in the first place (e.g., Fleischer et al., 1975).

Variable etching rates further affect this picture. The faster etch rates of IA235 and IA280 increase their track detection efficiencies by allowing the tracks to penetrate more deeply. This is why the $\eta q$ for IA235 matches IU at 0.8 μm despite lengths being shorter, and is even larger than IU at shallower depths. Likewise, if rates for IA280 were more comparable to IU and SU, as may be the case for geologically annealed tracks, the discrepancy in penetration between them, and thus $\eta q$, would be larger. On the other hand, when etching continues after tracks have reached bulk etching rates, as reflected by the closely spaced lines in the latter stages of etching (Fig. 20C,D), shallow etched features may begin to widen and become less

distinct, possibly becoming less likely to be recognized as tracks. Deeper penetration might then be required for detection, essentially increasing $z_c$ for faster-etching tracks, which would have a similar effect on efficiency.

In practice, lower counting efficiency for shorter tracks could mean that the ages of older, more annealed grains may be under-estimated because more tracks are missed. Insofar as zeta calibration is based on measurement of standards with low levels of annealing (Durango, Fish Canyon), this effect may make old samples appear younger, leading, for example, to an increased possibility of inversion of apatite fission-track and (U-Th)/He ages (e.g., Flowers et al., 2009). It may also affect the way such samples are quantitatively interpreted using thermal history modeling. The currently used relationship between length and density is based on the data and normalizations put forward by Green (1988), and attempts at first principles derivations of this relationship are ground-truthed against those data (e.g., Ketcham, 2003). If, for example, geological annealing results in different etching rates than laboratory annealing, the laboratory-measured relation may be biased.

Finally, we note the relatively steep slopes of the depth versus penetration curves (Fig. 20E), which under the $z_c$ model correspond to about 1% efficiency (0.01 relative penetration) per 0.1 µm depth. This highlights the critical role of consistency in mount preparation and polishing, and short semi-track identification, in achieving reproducible ages.

## 6.7 Outlook

Although the $v_T(x)$ model provides a range of insights into the fission track revelation and measurement process, these results should be only viewed as preliminary, and far more data are required to construct a complete picture that can fully inform practical apatite fission track analysis. Detailed step-etching measurements of fossil tracks at various stages of natural annealing, and induced tracks at more advanced stages of annealing, are required to ascertain how etching velocities evolve, including the advent of unetchable gaps. We particularly note that apatites have a range of solubilities, which will affect both etching rates and tip appearances and thus selection biases; work on apatites beyond Durango is thus a necessity. In addition, the effects of track **c**-axis angle need to be incorporated into the modeling, which would be aided by larger step-etching data sets better documenting a range of angles.

Such efforts can be combined with further community-level work to verify the extent to which analytical procedure and analyst criteria are responsible for the disappointing lack of consistency in fission-track length data between research groups. If $v_T/v_B$ or something like it can be established as the primary driver of divergence, it will empower the community to make its data both more reproducible and more plentiful. It is likely that etching procedures can be optimized to provide more abundant confined tracks, while creating an improved, quantitative link to the experimental data sets that underlie annealing models. Follow-on rewards will also include quantitative linkages between experimental data sets across laboratory groups, etching protocols, and even apatite varieties, as well as a more complete picture of track structure, all of which will improve our understanding of annealing.

Ultimately, as image capture, storage, and processing become more commonplace and more powerful, the possibility of using image analysis to evaluate the degree of track etching will grow. Development of the requisite capabilities promises to not

only make length data more consistent, but also more information-rich by characterizing etching extent, and thus true underlying latent length, on an individual track level.

## 7 Conclusions

A new, comprehensive model of confined fission track etching successfully fits a range of detailed step-etching data for Durango apatite, and illuminates details of track structure and the nature of the measurement process. Specific findings include:

- Along-track etching velocity, $v_T(x)$, varies within and among fission tracks, and affects all length measurements as executed with current protocols.

- Fission tracks that only experienced relatively low, near-Earth-surface temperatures etched more than twice as slowly as tracks subjected to high temperatures in the laboratory ($\geq 235°C$, 24 h).

- For many track populations, especially at low levels of annealing, a limited proportion of tracks that are intersected are seen and selected for measurement. Etching extent and analyst decision-making are more influential than length biasing in determining which track populations are more likely to be measured, which in turn affects the fidelity of thermal history modeling using track lengths.

- The fall-off in track etching velocity toward track tips, and variability among analysts in how they judge fission tracks to be sufficiently etched for measurement, is likely to be the major factor underlying poor reproducibility. Most variation in a major inter-laboratory measurement experiment can be explained by varying only the threshold for track selection, characterized as the ratio of along-track and bulk etching velocities at the etched track tip ($v_T/v_B$).

- A normalization procedure that accounts for analyst decision-making (e.g., $v_T/v_B$), in the context of an overall etching model, will be more robust than one based on mean track length measurements alone.

- The $v_T(x)$ model has the potential to allow optimization of etching protocols to maximize both confined track yield and information content, while retaining a quantitative link to the experimental annealing data sets that underlie thermal history modeling.

- Variable along-track etch rates may also influence the efficiency of semi-track counting for age determinations, and our understanding of the length-density relationship that underlies thermal history modeling.

## Appendix: $v_T(x)$ Etching Model Equations

We characterize the latent track as a set of etching rates along the track central axis, $x$, with the starting point for etching, or point where the impinging etchant pathway intersects the latent track, denoted as $x_{int}$. In the Constant-core model, the track middle section is assumed have to a constant etching rate $v_{Tmax}$ over length extent $\Delta x_{Tmax}$, beyond which etching rate falls at linear rate $A$ over distance $\Delta x_{Tmax-B}$ until it drops by $\Delta v_{Tmax-B}$ to $v_B$. In the Linear model, $\Delta x_{Tmax-B} = 0$. Defining our coordinate system such that the track extends in the positive direction from one tip at $x=0$, we define coordinates $x_{B_1}$ and $x_{B_2}$ to be the track tips, beyond which etching occurs at the bulk rate; and $x_{T_1}$ and $x_{T_2}$ demarking the central zone of maximum track etch rate (see Figure 1):

$$v_T(x) = \begin{cases} v_{Tmax} & x_{T_1} \leq x \leq x_{T_2} & \text{(A1a)} \\[2mm] v_{Tmax} - A\left|x - x_{T_1 \text{ or } T_2}\right|; \; A = \left|\dfrac{\Delta v_{Tmax-B}}{\Delta x_{Tmax-B}}\right| & x_{B_1} \leq x < x_{T_1}; \; x_{T_2} < x \leq x_{B_2} & \text{(A1b)} \\[2mm] v_B & x < x_{B_1}; \; x_{B_2} < x & \text{(A1c)} \end{cases}$$

The full latent length is

$$L_{lat} = \Delta x_{Tmax} + 2\Delta x_{Tmax-B} \tag{A2}$$

We next derive the time required to etch a confined semi-track starting at point $x_{int}$ and going toward one end. To begin, we etch in the positive direction toward $x_{B_2}$, and denote the etched semi-length $L_2$. To solve for the other half-length, $L_1$, we use the same set of equations and simply change the value of $x_{int}$ to $L_{lat} - x_{int}$. Etching of the grain mount commences at time $t=0$, and the confined track starts to etch at a later time $t_s$, to account for the time necessary to etch the impinging semi-track and then widen it sufficiently to intersect the confined track. We back-step though the three possible zones where etching may begin. If $x_{int}$ is in the right-hand zone between $x_{T_2}$ and $x_{B_2}$, then the time required to etch to a half-length $L_2$ is:

$$t(L_2) = \begin{cases} t_s + \displaystyle\int_{x_{int}}^{x_{int}+L_2} \frac{dx}{v_T(x)} & L_2 \leq x_{B_2} - x_{int} \qquad\qquad\text{(A3a)} \\[2em] t_s + \displaystyle\int_{x_{int}}^{x_{B2}} \frac{dx}{v_T(x)} + \frac{L_2 - (x_{B_2} - x_{int})}{v_B} & x_{B_2} - x_{int} < L_2 \qquad\qquad\text{(A3b)} \end{cases}$$

Expanding the integral term:

$$\int \frac{dx}{v_T(x)} = \int \frac{dx}{[v_{T_{max}} - A(x - x_{T_2})]} = \frac{1}{A}\int \frac{dx}{v_{T_{max}}/A + x_{T_2} - x} = \frac{-1}{A}\ln\left(\frac{v_{T_{max}}}{A} + x_{T_2} - x\right) \qquad\qquad\text{(A4)}$$

using

$$\frac{-1}{A}\int \frac{du}{u} = \frac{-1}{A}\ln(u)\ with\ u = a - x;\ a = \frac{v_{T_{max}}}{A} + x_{T_2};\ du = -dx$$

leads to the solution:

$$t(L_2) = \begin{cases} t_s - \dfrac{1}{A}\ln\left(1 - \dfrac{L_2}{\dfrac{v_{T_{max}}}{A} + x_{T_2} - x_{int}}\right) & L_2 \leq x_{B_2} - x_{int} \qquad\text{(A5a)} \\[3em] t_s - \dfrac{1}{A}\ln\left(\dfrac{\dfrac{v_{T_{max}}}{A} + x_{T_2} - x_{B_2}}{\dfrac{v_{T_{max}}}{A} + x_{T_2} - x_{int}}\right) + \dfrac{L_2 - (x_{B2} - x_{int})}{v_B} & x_{B_2} - x_{int} < L_2 \qquad\text{(A5b)} \end{cases}$$

If $x_{int}$ is in the central zone with maximum etching rate, then:

$$t_s + \frac{L_2}{v_{T_{max}}} \qquad\qquad L_2 \le x_{T_2} - x_{int} \tag{A6a}$$

$$t(L_2) = \qquad t_s + \frac{x_{T_2} - x_{int}}{v_{T_{max}}} + \int_{x_{T_2}}^{x_{int}+L_2} \frac{dx}{v_T(x)} \qquad\qquad x_{T_2} - x_{int} < L_2 \le x_{B_2} - x_{int} \tag{A6b}$$

$$t_s + \frac{x_{T_2} - x_{int}}{v_{T_{max}}} + \int_{x_{T_2}}^{x_{B2}} \frac{dx}{v_T(x)} + \frac{L_2 - \left(x_{B_2} - x_{int}\right)}{v_B} \qquad\qquad x_{B_2} - x_{int} < L_2 \tag{A6c}$$

After integrating, the solution becomes:

$$t_s + \frac{L_2}{v_{T_{max}}} \qquad\qquad L_2 \le x_{T_2} - x_{int} \tag{A7a}$$

$$t(L_2) = \qquad t_s + \frac{x_{T_2} - x_{int}}{v_{T_{max}}} - \frac{1}{A} \ln\left(1 + \frac{x_{T_2} - (x_{int} + L_2)}{\frac{v_{T_{max}}}{A}}\right) \qquad\qquad x_{T_2} - x_{int} < L_2 \le x_{B_2} - x_{int} \tag{A7b}$$

$$t_s + \frac{x_{T_2} - x_{int}}{v_{T_{max}}} - \frac{1}{A} \ln \frac{v_B}{v_{T_{max}}} + \frac{L_2 - \left(x_{B_2} - x_{int}\right)}{v_B} \qquad\qquad x_{B_2} - x_{int} < L_2 \tag{A7c}$$

Finally, if $x_{int}$ is in the left-hand zone between $x_{T_1}$ and $x_{B_1}$:

$$t_s + \int_{x_{int}}^{x_{int}+L_2} \frac{dx}{v_T(x)} \qquad\qquad L_2 \leq x_{T_1} - x_{int} \qquad\qquad \text{(A8a)}$$

$$t(L_2) =$$

$$t_s + \int_{x_{int}}^{x_{T_1}} \frac{dx}{v_T(x)} + \frac{L_2 - (x_{T_1} - x_{int})}{v_{T_{max}}} \qquad\qquad x_{T_1} - x_{int} < L_2 \leq x_{T_2} - x_{int} \qquad\qquad \text{(A8b)}$$

$$t_s + \int_{x_{int}}^{x_{T_1}} \frac{dx}{v_T(x)} + \frac{x_{T_2} - x_{T_1}}{v_{T_{max}}} + \int_{x_{T_2}}^{x_{int}+L_2} \frac{dx}{v_T(x)} \qquad\qquad x_{T_2} - x_{int} < L_2 \leq x_{B_2} - x_{int} \qquad\qquad \text{(A8c)}$$

$$t_s + \int_{x_{int}}^{x_{T_1}} \frac{dx}{v_T(x)} + \frac{x_{T_2} - x_{T_1}}{v_{T_{max}}} + \int_{x_{T_2}}^{x_{B_2}} \frac{dx}{v_T(x)} + \frac{L_2 - (x_{B_2} - x_{int})}{v_B} \qquad\qquad x_{B_2} - x_{int} < L_2 \qquad\qquad \text{(A8d)}$$

Integrating leads to:

$$t_s + \frac{1}{A} \ln\left( 1 + \frac{L_2}{\frac{v_{T_{max}}}{A} + x_{int} - x_{T_1}} \right) \qquad\qquad L_2 \leq x_{T_1} - x_{int} \qquad\qquad \text{(A9a)}$$

$$t(L_2) =$$

$$t_s - \frac{1}{A} \ln\left[ 1 + \frac{A(x_{int} - x_{T_1})}{v_{T_{max}}} \right] + \frac{L_2 - (x_{T_1} - x_{int})}{v_{T_{max}}} \qquad\qquad x_{T_1} - x_{int} < L_2 \leq x_{T_2} - x_{int} \qquad \text{(A9b)}$$

$$t_s - \frac{1}{A} \ln\left[ 1 + \frac{A(x_{int} - x_{T_1})}{v_{T_{max}}} \right] + \frac{x_{T_2} - x_{T_1}}{v_{T_{max}}} - \frac{1}{A} \ln\left( 1 + \frac{A(x_{T_2} - x_{int} - L_2)}{v_{T_{max}}} \right) \qquad x_{T_2} - x_{int} < L_2 \leq x_{B_2} - x_{int} \qquad \text{(A9c)}$$

$$t_s - \frac{1}{A} \ln\left[ 1 + \frac{A(x_{int} - x_{T_1})}{v_{T_{max}}} \right] + \frac{x_{T_2} - x_{T_1}}{v_{T_{max}}} - \frac{1}{A} \ln \frac{v_B}{v_{T_{max}}} + \frac{L_2 - (x_{B_2} - x_{int})}{v_B} \qquad\qquad x_{B_2} - x_{int} < L_2 \qquad\qquad \text{(A9d)}$$

Solving each set of equations for length as a function of etching time, and transforming the length boundaries to time boundaries, in the right-hand zone:

$$0 \qquad t \leq t_s \qquad (A10a)$$

$$L_2(t) = \left(\frac{v_{Tmax}}{A} + x_{T_2} - x_{int}\right)\left[1 - e^{-A(t-t_s)}\right] \qquad t_s < t \leq t_s + \frac{1}{A}\ln\left(\frac{v_{Tmax} + A(x_{T_2} - x_{int})}{v_B}\right) \qquad (A10b)$$

$$v_B\left[t - t_s - \frac{1}{A}\ln\left(\frac{v_{Tmax} + A(x_{T_2} - x_{int})}{v_B}\right)\right] + (x_{B2} - x_{int}) \qquad t_s + \frac{1}{A}\ln\left(\frac{v_{Tmax} + A(x_{T_2} - x_{int})}{v_B}\right) < t \qquad (A10c)$$

In the central zone:

$$0 \qquad t \leq t_s \qquad (A11a)$$

$$v_{Tmax}(t - t_s) \qquad t_s < t \leq t_s + \frac{x_{T_2} - x_{int}}{v_{Tmax}} \qquad (A11b)$$

$$L_2(t) = x_{T_2} - x_{int} + \frac{v_{Tmax}}{A}\left[1 - e^{-A\left(t - t_s - \frac{x_{T_2} - x_{int}}{v_{Tmax}}\right)}\right] \qquad t_s + \frac{x_{T_2} - x_{int}}{v_{Tmax}} < t \leq t_s + \frac{x_{T_2} - x_{int}}{v_{Tmax}} - \frac{1}{A}\ln\left(\frac{v_B}{v_{Tmax}}\right) \qquad (A11c)$$

$$x_{B_2} - x_{int} + v_B\left[t - t_s - \frac{x_{T_2} - x_{int}}{v_{Tmax}} + \frac{1}{A}\ln\left(\frac{v_B}{v_{Tmax}}\right)\right] \qquad t_s + \frac{x_{T_2} - x_{int}}{v_{Tmax}} - \frac{1}{A}\ln\left(\frac{v_B}{v_{Tmax}}\right) < t \qquad (A11d)$$

And in the left-hand zone:

$$L_2(t) = $$

$$0 \qquad\qquad t \le t_s \qquad\qquad (A12a)$$

$$\left(\frac{v_{Tmax}}{A} + x_{int} - x_{T_1}\right)\left[e^{A(t-t_s)} - 1\right] \qquad t_s < t \le t_s - \frac{1}{A}\ln\left[1 + \frac{A\left(x_{int} - x_{T_1}\right)}{v_{Tmax}}\right] \qquad (A12b)$$

$$x_{T_1} - x_{int} + v_{Tmax}\left\{t - t_s \right.$$
$$\left. + \frac{1}{A}\ln\left[1 + \frac{A\left(x_{int} - x_{T_1}\right)}{v_{Tmax}}\right]\right\} \qquad \begin{aligned}&t_s - \frac{1}{A}\ln\left[1 + \frac{A\left(x_{int} - x_{T_1}\right)}{v_{Tmax}}\right] < t \\ &\le t_s - \frac{1}{A}\ln\left[1 + \frac{A\left(x_{int} - x_{T_1}\right)}{v_{Tmax}}\right] \\ &+ \frac{x_{T_2} - x_{T_1}}{v_{Tmax}}\end{aligned} \qquad (A12c)$$

$$x_{T_2} - x_{int}$$
$$-\frac{v_{Tmax}}{A}\left(e^{-A\left\{t - t_s + \frac{1}{A}\ln\left[1 + \frac{A(x_{int} - x_{T_1})}{v_{Tmax}}\right] - \frac{x_{T_2} - x_{T_1}}{v_{Tmax}}\right\}} - 1\right) \qquad \begin{aligned}&t_s - \frac{1}{A}\ln\left[1 + \frac{A\left(x_{int} - x_{T_1}\right)}{v_{Tmax}}\right] + \frac{x_{T_2} - x_{T_1}}{v_{Tmax}} < t \\ &\le t_s - \frac{1}{A}\ln\left[1 + \frac{A\left(x_{int} - x_{T_1}\right)}{v_{Tmax}}\right] \\ &+ \frac{x_{T_2} - x_{T_1}}{v_{Tmax}} - \frac{1}{A}\ln\frac{v_B}{v_{Tmax}}\end{aligned} \qquad (A12d)$$

$$x_{B_2} - x_{int} + v_B\left\{t - t_s + \frac{1}{A}\ln\left[1 + \frac{A\left(x_{int} - x_{T_1}\right)}{v_{Tmax}}\right]\right.$$
$$\left. - \frac{x_{T_2} - x_{T_1}}{v_{Tmax}} + \frac{1}{A}\ln\frac{v_B}{v_{Tmax}}\right\} \qquad \begin{aligned}&t_s - \frac{1}{A}\ln\left[1 + \frac{A\left(x_{int} - x_{T_1}\right)}{v_{Tmax}}\right] + \frac{x_{T_2} - x_{T_1}}{v_{Tmax}} - \frac{1}{A}\ln\frac{v_B}{v_{Tmax}} \\ &< t\end{aligned} \qquad (A12e)$$

To solve for the other half-length, $L_1$, we use the same set of equations and simply change the value of $x_{int}$ to $L_{lat} - x_{int}$.

**Code Availability**

The modelling introduced in this contribution was coded in the IDL programming language. The GitHub repository containing the IDL code used to generate our results can be accessed at https://doi.org/10.5281/zenodo.4779128. It is made available under the GPL-3.0 License.

**Data Availability**

The data are available as an Excel spreadsheet in the Texas Data Repository, at https://doi.org/10.18738/T8/MPWDZI.

**Author Contributions**

RAK derived and coded the model, interpreted the results, wrote the text, and drafted most figures. MTT planned and executed the measurements, drafted Figure 9, and aided with interpretation. RAK and MTT jointly initiated the study, which is an extension of the research program begun with MTTs dissertation.

**Competing Interests**

The authors declare that they have no competing interests.

**Acknowledgements**

This work was supported by the Geology Foundation of the Jackson School of Geosciences. We are grateful for thorough and constructive reviews by R. Jonckheere, B. Wauschkuhn, P. Green, A. Gleadow, and E. Sobel, which helped us to improve this work, as well as editorial handling by C. Spiegel.

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

**Table 1**: Data and model fits

| Fission tracks | Exp.[1] | Src.[2] | Etch Time (s)[3] | N | $l_m$ (μm) | $dl_m$ (μm) | σ (μm) | $z_{Int}$[4] (μm) | Int./ trk.[5] | $l_m$ (μm) | σ (μm) | %sel[6] | $t_s$[7] (s) | $z_{Int}$[4] (μm) | $l_m$ (μm) | σ (μm) | %sel[6] | $t_s$[7] (s) | $z_{Int}$[4] (μm) |
|---|---|---|---|---|---|---|---|---|---|---|---|---|---|---|---|---|---|---|---|
| | | | | | | | Measured | | | | | Constant-core model | | | | | Linear model | | |
| Spontaneous Unannealed Cf-irradiated | SU-10 | SE3 | 10 | 47 | 9.11 | 0.30 | 2.08 | | | 9.1 | 1.7 | 26% | 4.4 | 2.4 | 9.7 | 1.3 | 29% | 4.5 | 2.4 |
| | | | 15 | 47 | 13.03 | 0.28 | 1.95 | | | 13.3 | 0.8 | | | | 13.3 | 0.6 | | | |
| | | | 20 | 47 | 14.89 | 0.11 | 0.73 | | | 14.9 | 0.7 | | | | 14.9 | 0.4 | | | |
| | | | 25 | 47 | 15.43 | 0.11 | 0.76 | | | 15.4 | 0.7 | | | | 15.5 | 0.4 | | | |
| | | | 30 | 47 | 15.69 | 0.11 | 0.76 | 2.7 | 3.5 | 15.7 | 0.8 | | | | 15.8 | 0.5 | | | |
| | SU-20 | TK20 | 20 | 87 | 14.43 | 0.08 | 0.78 | | | 14.4 | 0.8 | 18% | 6.1 | 2.8 | 14.3 | 0.6 | 20% | 6.3 | 2.9 |
| Induced Unannealed | IU-10 | SE2 | 10 | 127 | 9.89 | 0.18 | 1.97 | | | 9.9 | 1.8 | 12% | 4.2 | 2.3 | 10.1 | 1.5 | 29% | 4.8 | 2.9 |
| | | | *15** | *127* | *15.31* | *0.08* | *0.92* | | | *14.5* | *0.8* | | | | *14.5* | *0.8* | | | |
| | | | 20 | 127 | 16.19 | 0.07 | 0.80 | | | 16.2 | 0.7 | | | | 16.1 | 0.4 | | | |
| | | | 25 | 127 | 16.99 | 0.07 | 0.82 | | | 16.8 | 0.7 | | | | 16.8 | 0.4 | | | |
| | | | 30 | 127 | 17.18 | 0.08 | 0.86 | 4.4 | 2.5 | 17.1 | 0.8 | | | | 17.2 | 0.5 | | | |
| | IU-20 | SE1 | 20 | 72 | 15.77 | 0.08 | 0.67 | | | 15.8 | 0.8 | 15% | 6.5 | 3.4 | 15.6 | 0.6 | 15% | 6.4 | 3.4 |
| | | | 25 | 72 | 16.35 | 0.08 | 0.72 | | | 16.6 | 0.8 | | | | 16.6 | 0.5 | | | |
| | | | 30 | 72 | 16.92 | 0.09 | 0.77 | 3.9 | 1.7 | 16.9 | 0.8 | | | | 17.0 | 0.5 | | | |
| Induced Annealed (235°C, 24h) | IA235-10 | EAE3 | 10 | 81 | 13.38 | 0.12 | 1.12 | 1.6 | 4.2 | 13.2 | 1.4 | 54% | 4.8 | 4.0 | 13.2 | 1.3 | 54% | 4.8 | 4.0 |
| | IA235-15 | SE6 | 15 | 105 | 13.77 | 0.13 | 1.32 | | | 14.1 | 1.4 | 73% | 7.7 | 4.5 | 14.0 | 1.3 | 73% | 7.7 | 4.5 |
| | | | 20 | 105 | 15.13 | 0.08 | 0.78 | | | 15.1 | 0.8 | | | | 15.1 | 0.5 | | | |
| | | | 25 | 105 | 15.34 | 0.08 | 0.79 | | | 15.4 | 0.8 | | | | 15.4 | 0.5 | | | |
| | | | 30 | 105 | 15.65 | 0.08 | 0.78 | 1.4 | 6.3 | 15.6 | 0.8 | | | | 15.6 | 0.5 | | | |
| Induced Annealed (270°C, 24h) | IA270-10 | EAE2 | 10 | 94 | 11.76 | 0.14 | 1.36 | 1.5 | 4.1 | 11.8 | 1.3 | 53% | 4.9 | 3.6 | 11.8 | 1.1 | 53% | 4.9 | 3.6 |
| | IA270-15 | SE5 | 15 | 113 | 12.58 | 0.09 | 1.00 | | | 12.6 | 1.3 | 72% | 7.7 | 4.1 | 12.6 | 1.2 | 72% | 7.7 | 4.1 |
| | | | 20 | 113 | 13.48 | 0.09 | 0.96 | | | 13.6 | 0.8 | | | | 13.6 | 0.5 | | | |
| | | | 25 | 113 | 13.87 | 0.09 | 0.95 | | | 13.8 | 0.8 | | | | 13.8 | 0.5 | | | |
| | | | 30 | 113 | 14.15 | 0.09 | 0.95 | 1.3 | 5.8 | 14.1 | 0.8 | | | | 14.1 | 0.5 | | | |
| Induced Annealed (280°C, 24h) | IA280-10 | EAE1 | 10 | 57 | 11.25 | 0.12 | 0.93 | 2.0 | 3.4 | 11.2 | 1.0 | 55% | 4.8 | 3.4 | 11.2 | 0.9 | 60% | 4.8 | 3.5 |
| | IA280-20 | SE4 | 20 | 146 | 12.33 | 0.07 | 0.87 | | | 12.4 | 0.8 | 65% | 8.8 | 4.0 | 12.4 | 0.5 | 65% | 8.8 | 4.0 |
| | | | 25 | 146 | 12.64 | 0.07 | 0.85 | | | 12.6 | 0.8 | | | | 12.7 | 0.5 | | | |
| | | | 30 | 146 | 12.93 | 0.07 | 0.83 | 2.8 | 5.5 | 12.9 | 0.8 | | | | 12.9 | 0.5 | | | |

[1]Experiment code: [S or I: spontaneous or induced][U or A###: unannealed or lab-annealed at ###°C for 24h][-##: Seconds in first etch step]
[2]Data source: SE and EAE data from Tamer and Ketcham (2020a); TK20 data from Tamer and Ketcham (2020b)
[3]All etching done using 5.5 M $HNO_3$ at 21°C.
[4]Mean depth of intersection by semi-track (all intersections for measured data, first intersection for models)

[5]Semi-track intersections per confined track
[6]Percent of confined tracks that pass selection criteria
[7]Mean time selected tracks start etching (are intersected by semi-tracks)
[*]Not used for model fitting

**Table 2**: Best-fit model parameters, ranges of comparable fits, and goodness of fit

| Data | Constant-core model | | | | | Linear model | | | |
|------|---------------------|---|---|---|---|--------------|---|---|---|
| | $v_{Tmax}$ (µm/s) | $\Delta x_{Tmax}$ (µm) | $L_{lat}$ (µm) | $\chi^2_v$ | $\chi^2_v$ r.v.[1] | $v_{Tmax}$ (µm/s) | $L_{lat}$ (µm) | $\chi^2_v$ | $\chi^2_v$ r.v.[1] |
| SU | 1.054 (0.981 - 1.111) | 5.48 (4.68 - 6.50) | 15.57 (15.52 - 15.61) | 0.21 | 6.4% | 1.495 (1.469 - 1.545) | 15.67 (15.60 - 15.70) | 0.50 | 8.8% |
| IU | 1.639 (1.332 - 1.705) | 0.74 (0.02 - 4.22) | 17.02 (16.94 - 17.10) | 2.32 | 3.9% | 1.700 (1.638 - 1.752) | 17.00 (16.95 - 17.15) | 2.43 | 5.2% |
| IA235 | 3.592 (2.532 - 3.642) | 0.26 (0.01 - 5.71) | 15.10 (15.04 - 15.14) | 1.76 | 4.3% | 3.600 (3.576 - 3.700) | 15.11 (15.04 - 15.14) | 1.84 | 3.6% |
| IA270 | 3.083 (2.590 - 3.088) | 0.26 (0.23 - 2.60) | 13.58 (13.57 - 13.59) | 0.66 | 0.5% | 3.128 (3.118 - 3.157) | 13.58 (13.57 - 13.59) | 0.67 | 1.3% |
| IA280 | 3.109 (2.492 - 3.281) | 0.78 (0.06 - 3.42) | 12.40 (12.39 - 12.42) | 0.42 | 1.8% | 3.317 (3.273 - 3.353) | 12.41 (12.39 - 12.42) | 0.41 | 2.1% |

[1]Relative variation in reduced chi-squared over 20 replicate runs.


**Table 3**: Constant-core model predictions of induced unannealed fission track length measurements in Durango apatite based on user selection criterion

| $v_T/v_B$ | $l_m$ (µm) | $\sigma$ (µm) | %sel | $t_s$ (s) | $z_{int}$ (µm) |
|-----------|-----------|----------------|------|-----------|----------------|
| 40 | 14.1 | 1.8 | 51% | 9.9 | 4.2 |
| 30 | 14.6 | 1.4 | 41% | 9.1 | 4.0 |
| 20 | 15.3 | 1.0 | 28% | 7.9 | 3.8 |
| 16 | 15.6 | 0.9 | 22% | 7.3 | 3.6 |
| 12 | 15.9 | 0.8 | 15% | 6.5 | 3.4 |
| 10 | 16.0 | 0.8 | 11% | 5.9 | 3.3 |
| 8 | 16.2 | 0.8 | 7.5% | 5.2 | 3.0 |
| 4 | 16.6 | 0.8 | 1.03% | 3.3 | 2.3 |
| 2 | 16.8 | 0.8 | 0.003% | 1.2 | 1.0 |


**Table 4**: Predicted changes in confined track length, standard deviation, and selection efficiency with etching time, selection criteria

| Sample | Etch time (s) | $v_T/v_B = 12$ | | | | | $v_T/v_B = 8$ | | | | | $v_T/v_B = 4$ | | | | |
|---|---|---|---|---|---|---|---|---|---|---|---|---|---|---|---|---|
| | | $l_m$ (µm) | σ (µm) | %sel | $t_s$ (s) | $z_{int}$ (µm) | $l_m$ (µm) | σ (µm) | %sel | $t_s$ (s) | $z_{int}$ (µm) | $l_m$ (µm) | σ (µm) | %sel | $t_s$ (s) | $z_{int}$ (µm) |
| SU-20[1] | 20 | 14.3 | 0.8 | 13% | 6.5 | 3.0 | 14.6 | 0.7 | 6.3% | 5.4 | 2.6 | 14.7 | 0.7 | 0.9% | 3.7 | 2.0 |
| | 25 | 14.7 | 0.8 | 28% | 9.4 | 3.6 | 14.9 | 0.8 | 19% | 8.3 | 3.4 | 15.1 | 0.8 | 8.1% | 6.4 | 2.9 |
| IU-20 | 20 | 15.7 | 0.8 | 15% | 6.5 | 3.4 | 16.0 | 0.8 | 7.7% | 5.3 | 3.1 | 16.1 | 0.7 | 1.2% | 3.5 | 2.4 |
| | 25 | 16.1 | 0.9 | 30% | 9.3 | 4.1 | 16.4 | 0.8 | 21% | 8.2 | 3.8 | 16.5 | 0.8 | 9.1% | 6.2 | 3.3 |
| IA280-20 | 20 | 12.4 | 0.8 | 65% | 8.8 | 3.9 | 12.4 | 0.8 | 60% | 8.4 | 3.9 | 12.5 | 0.8 | 52% | 7.7 | 3.9 |
| | 25 | 12.5 | 0.8 | 72% | 11.4 | 4.1 | 12.5 | 0.8 | 69% | 11.0 | 4.0 | 12.6 | 0.8 | 63% | 10.4 | 4.0 |

[1]Model does not include Cf-irradiation, making predictions different from Table 1.


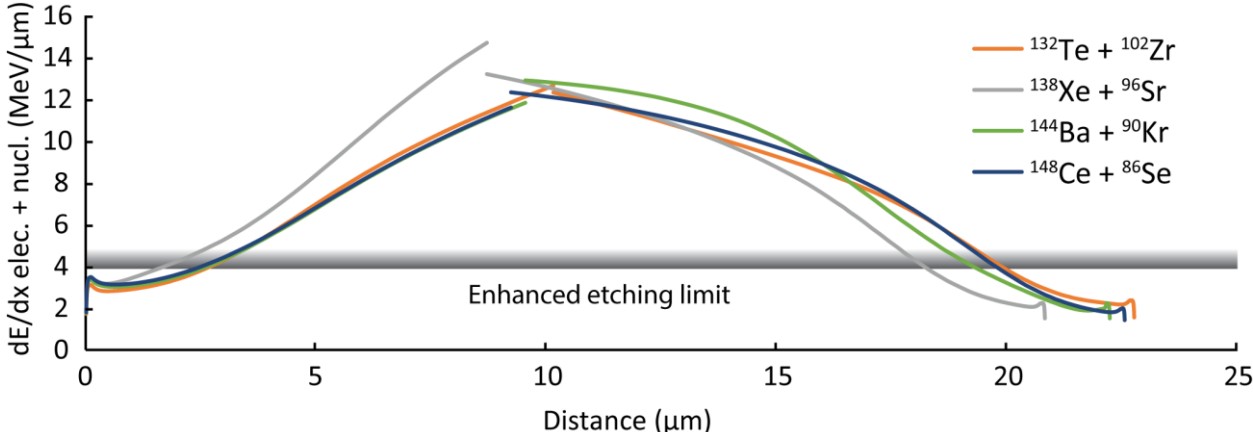

**Figure 1: Energy loss profiles from electronic and nuclear interactions for a sampling of possible products from induced fission of $^{235}$U in apatite, calculated using SRIM (Zeigler 2008). For each fission pair, the left curve segment is the heavier, lower-energy fragment. The shaded horizontal line represents the approximate limit below which energy loss no longer results in enhanced etching rates.**

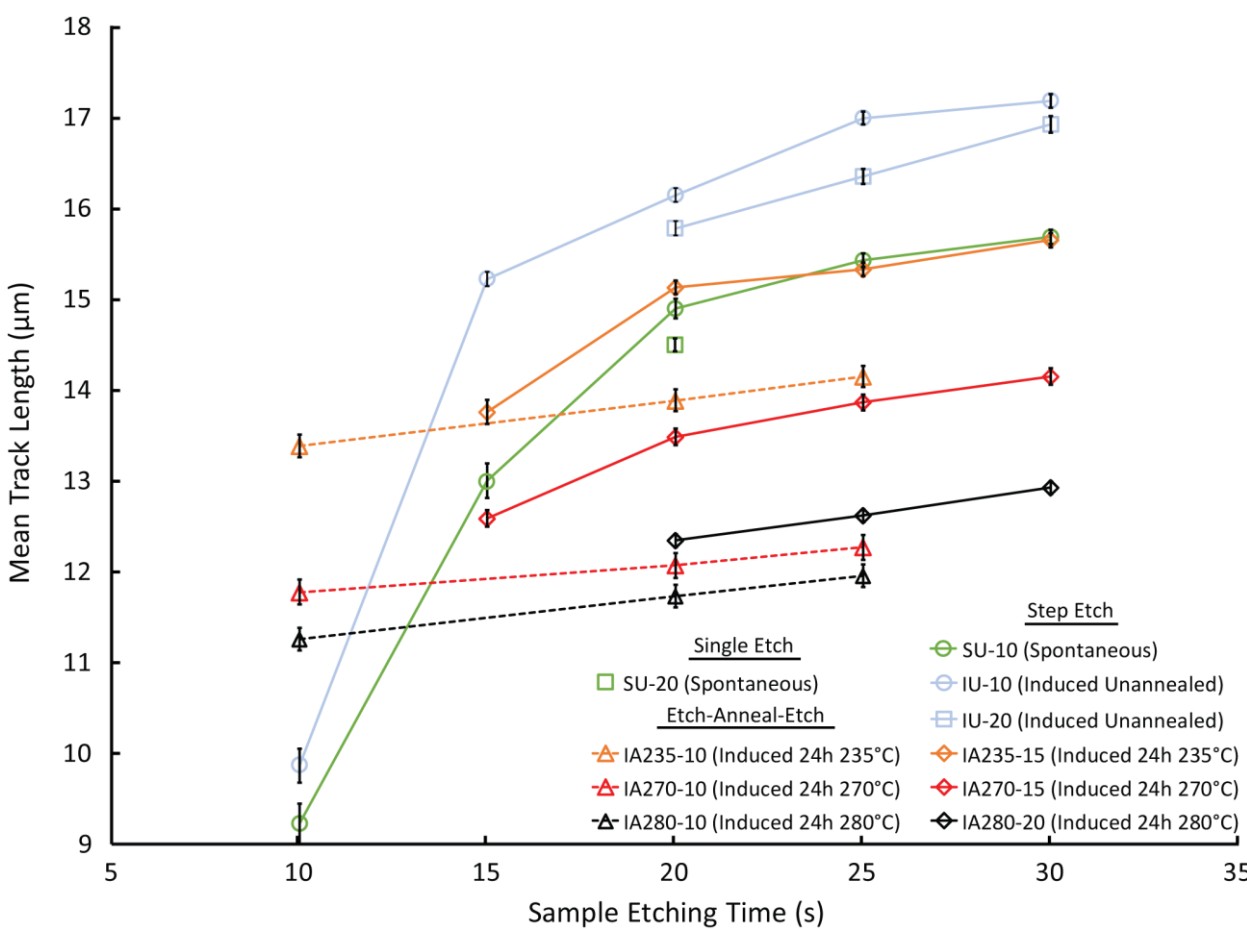

**Figure 2: Mean track length data used in this study, from Tamer and Ketcham (2020a,b); see Table 1.  Error bars show**
**1 SE.  Only the first step of etch-anneal-etch experiments are used here, but post-complete-annealing etching steps at 20 and 25 seconds, connected by dashed lines, indicate bulk etching rate, $v_B$.**

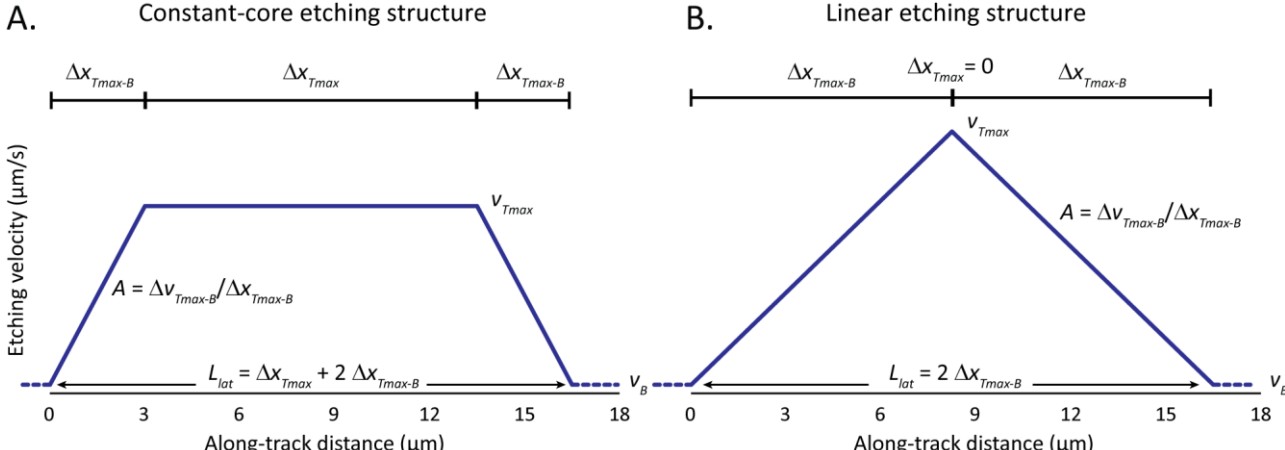

**Figure 3: Model schemas for fission-track etching structure.  A. Constant-core; B. Linear.**  $v_{Tmax}$ **is maximum along track etching velocity and** $v_B$ **is bulk grain etching velocity.**  $\Delta x_{Tmax}$ **is the width of the constant-etching-rate core, and** $\Delta x_{Tmax-B}$ **is the width of the zone from the core to the track tip.**  $A$ **is the track etch rate gradient as it falls toward the tip, and** $L_{lat}$ **is the full latent track length.  The intersection point of the etchant pathway with the latent track,** $x_{int}$ **, can occur anywhere along its length; the etching of the track will begin from that point, and follow the etching structure in each direction.**


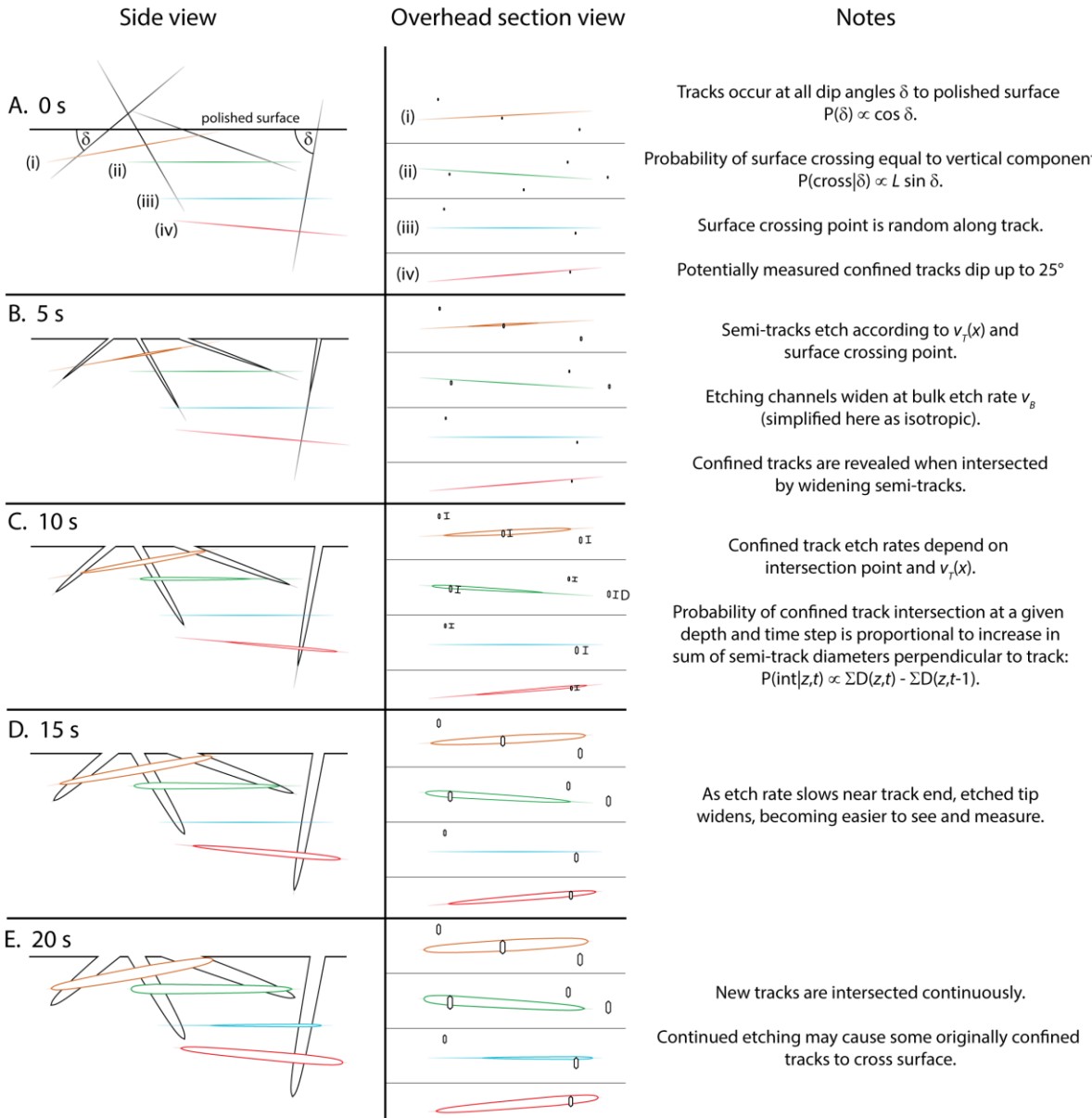

**Side view**  **Overhead section view**  **Notes**

**A. 0 s**
polished surface
(i)  (ii)  (iii)  (iv)

(i) Tracks occur at all dip angles $\delta$ to polished surface $P(\delta) \propto \cos \delta$.

(ii) Probability of surface crossing equal to vertical component $P(\mathrm{cross}|\delta) \propto L \sin \delta$.

(iii) Surface crossing point is random along track.

(iv) Potentially measured confined tracks dip up to 25°

**B. 5 s**

Semi-tracks etch according to $v_T(x)$ and surface crossing point.

Etching channels widen at bulk etch rate $v_B$ (simplified here as isotropic).

Confined tracks are revealed when intersected by widening semi-tracks.

**C. 10 s**

Confined track etch rates depend on intersection point and $v_T(x)$.

Probability of confined track intersection at a given depth and time step is proportional to increase in sum of semi-track diameters perpendicular to track: $P(\mathrm{int}|z,t) \propto \Sigma D(z,t) - \Sigma D(z,t-1)$.

**D. 15 s**

As etch rate slows near track end, etched tip widens, becoming easier to see and measure.

**E. 20 s**

New tracks are intersected continuously.

Continued etching may cause some originally confined tracks to cross surface.


**Figure 4: Schematic overview of TINT revelation model, through successive etching times of 0 through 20 seconds (A-E). Semi-tracks are shown in gray, and confined tracks are shown in colors to facilitate matching tracks in side view versus overhead section view. Overhead sections are in the sometimes-oblique plane of the confined track, as it is the expansion of the semi-track in that plane that leads to intersection. Variations in latent track brightness indicate relative**
**etching velocity. Etchant pathways in overhead view are depicted as anisotropic prisms, but the present model effectively simplifies them as circles because track orientation is not considered.**

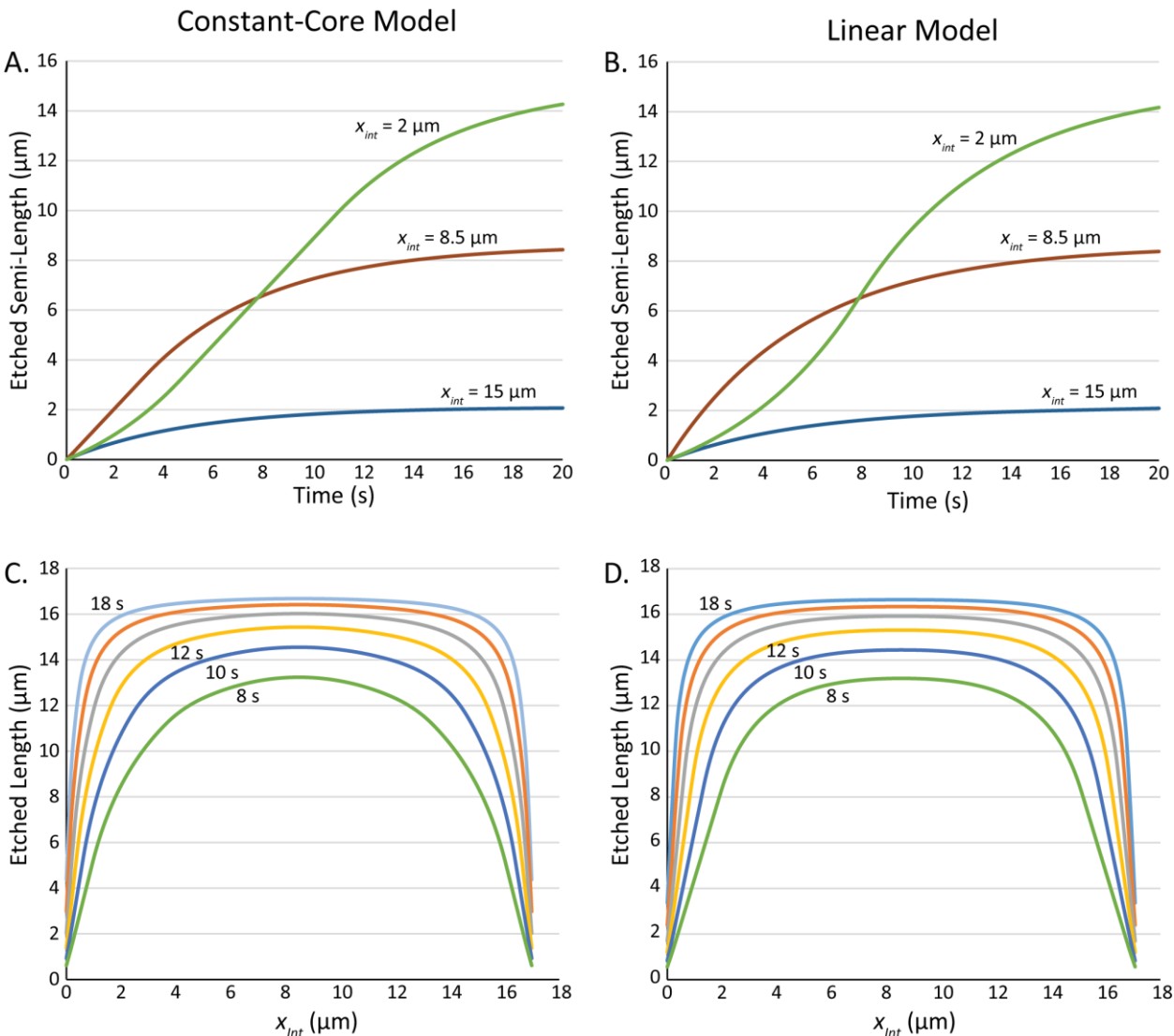

**Figure 5:** Top row: lengthening along unannealed induced fission tracks (latent length 17 μm) starting from midpoint and near each tip, etching from etchant pathway intersection point toward one tip. A. Constant-core etching structure with 8-μm core and 1 μm/s maximum $v_T$. B. Linear model with a 1.55 μm/s maximum $v_T$. Bottom row: evolution of total track length as a function of time, depending on intersection point, at 2-second intervals after etching of the latent track commences; C. Constant-core model, D. Linear model.

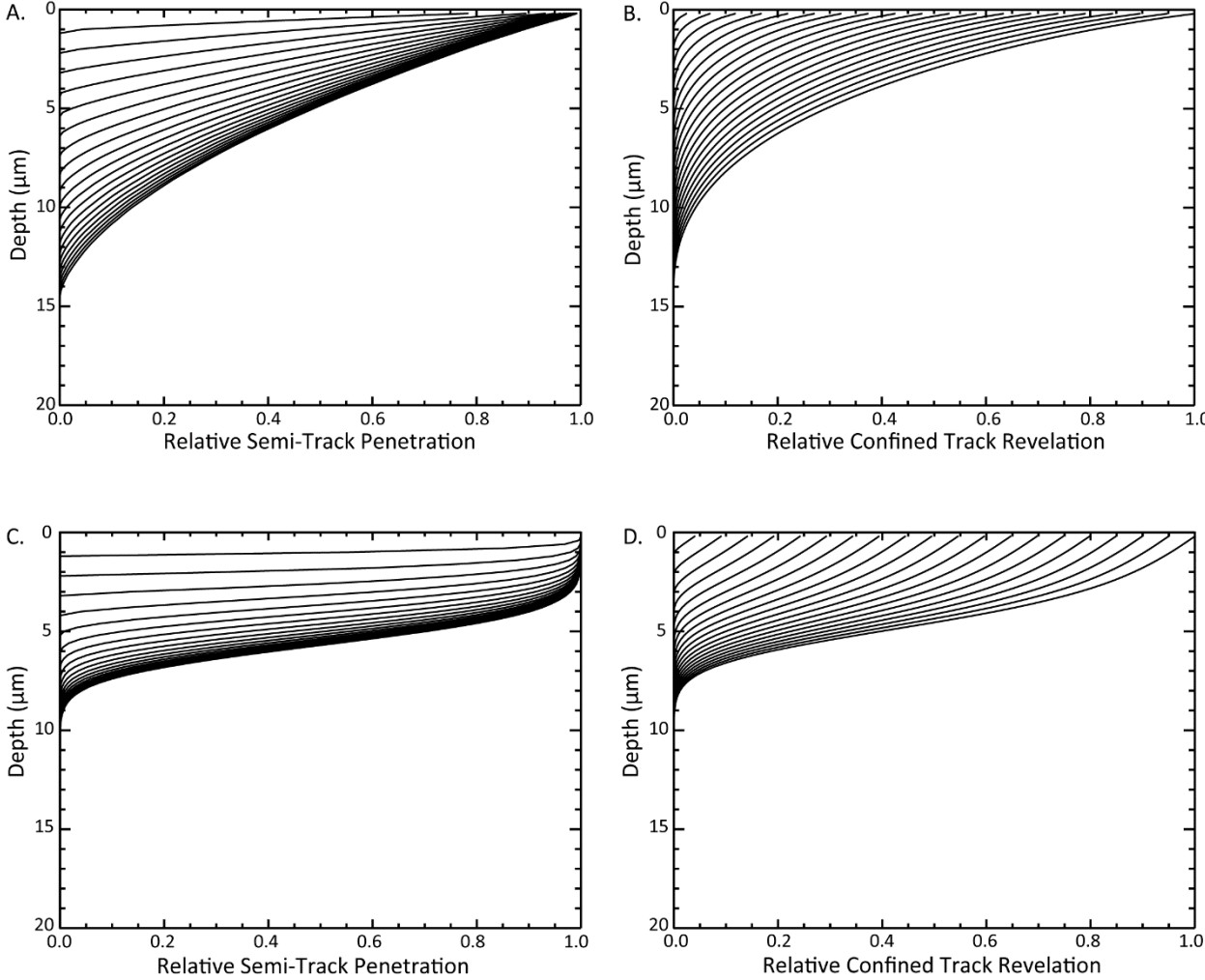

Figure 6: Semi-track penetration and confined track revelation. Semi-track penetration shows the relative number of surface-intersecting semi-tracks that have penetrated to a given depth by a given time. As semi-tracks penetrate, they also widen, allowing them to intersect confined tracks (Fig. 4). Confined track revelation reflects the probability of confined tracks being revealed by this process as a function of time and depth. Lines in figure correspond to relative penetration of semi-tracks and revelation of confined tracks at etching times every second from 1 to 20 s, with the upper left line in each diagram being 1 s and the lower right line being 20s. A, B: Penetration and revelation based on randomly oriented unannealed induced tracks. C, D: Penetration and revelation based on [252]Cf tracks oriented at 75° to the grain surface. Calculation performed at time steps of 0.2 s and depth steps of 0.2 μm.

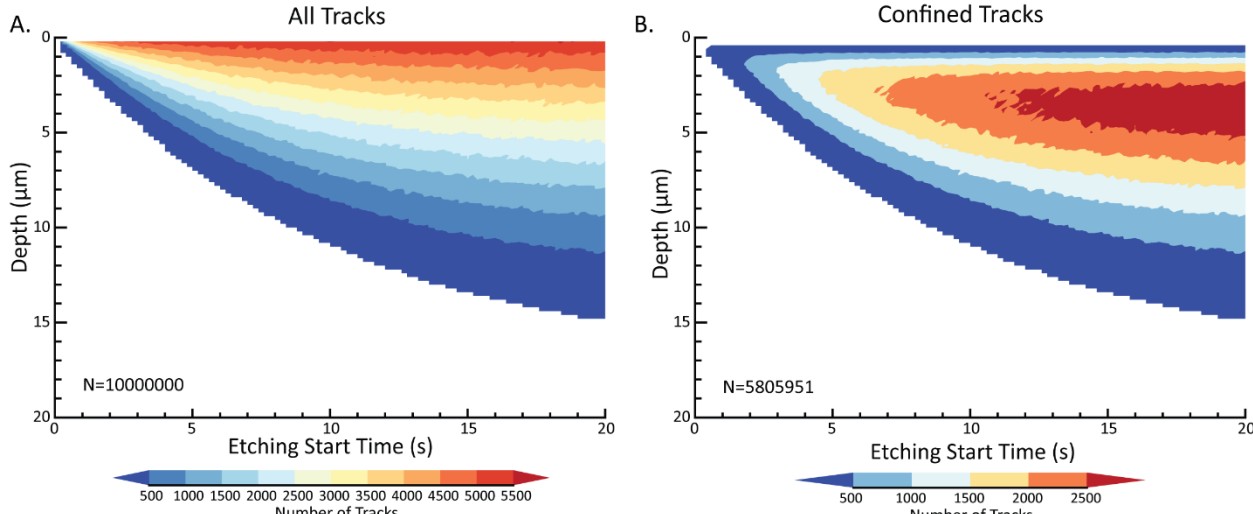

Figure 7: Track intersection results for $10^7$ unannealed induced tracks after 20 s of etching, using the Constant-core model for unannealed induced tracks. A. Contour diagram of all intersections; B. contour diagram of intersections after excluding tracks that reach the surface (semi-tracks).

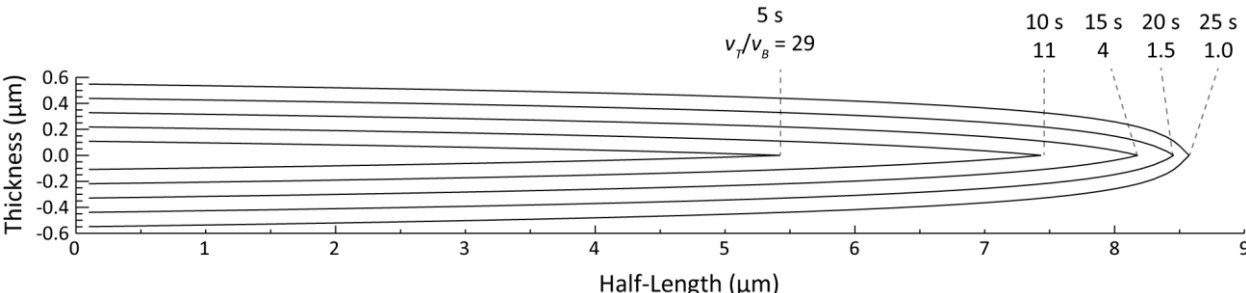

Figure 8: Approximate model of tip evolution and $v_T/v_B$ for constant-core model of an unannealed induced fission track. Each profile starts at the track center, starting from 5 s after the beginning of etching for the innermost profile and proceeding in 5 s increments to 25 s for the outermost profile.

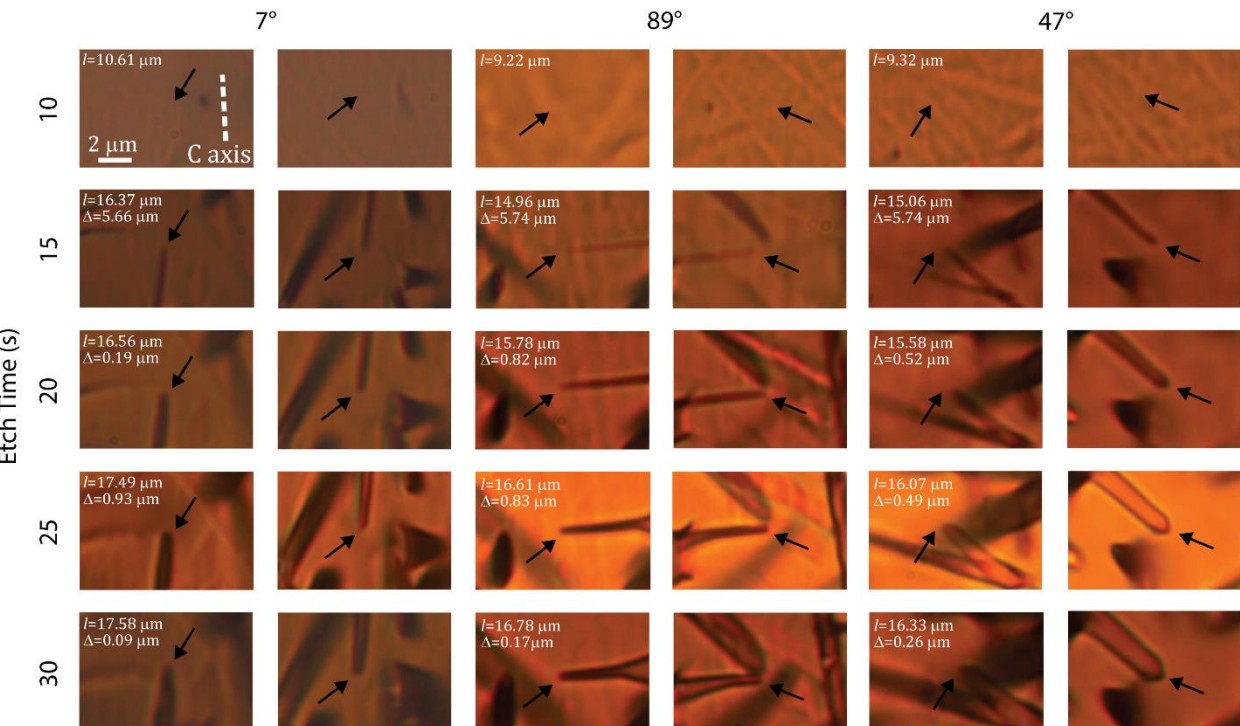

**Figure 9: Transmitted-light microscope images of tips of three unannealed induced fission tracks at different crystallographic angles through progressive etching steps; pictures are paired left-right, showing each tip of the same track separately. Scale and orientation in top left applies to all images. Etch times are since the beginning of etching, although it is unknown exactly when each track was intersected and began to etch.**

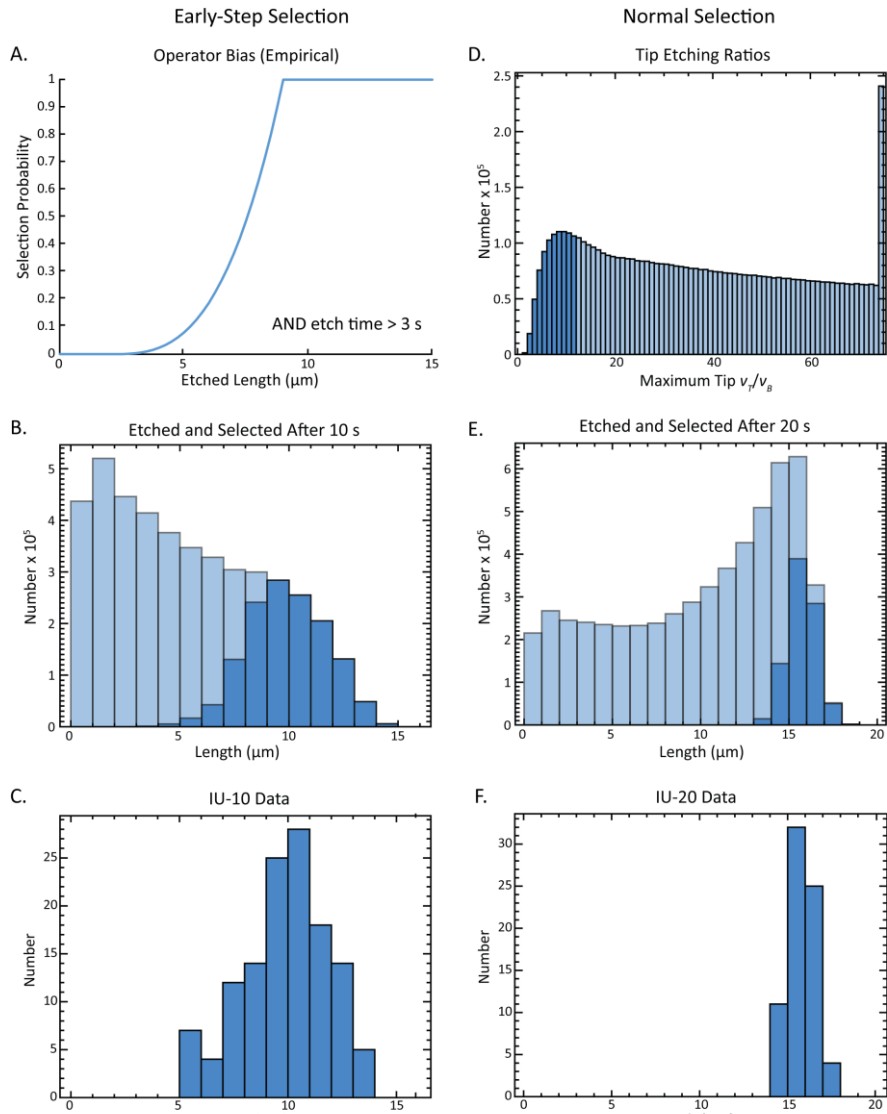

**Figure 10: Illustration of how selection criteria for confined fission tracks affects length distributions. In all histograms, faded bars show all etched tracks, and darkened bars show tracks that pass selection criteria. A. Empirical probability function for track selection when first etching step is below 20 s, when analyst is accepting most tracks found. B. Model histogram for all etched tracks after 10 s (unannealed induced). C. Measured track lengths after 10 s of etching (experiment IU-10). D. Model $v_T/v_B$ distribution of unannealed induced tracks after 20 s of etching, with darker bars showing tracks with $v_T/v_B \leq 12$. E. Model histogram of all etched tracks after 20 s, when analyst is using selection criteria employed for standard fission-track analysis. F. Measured track lengths after a single 20s etch (experiment IU-20).**

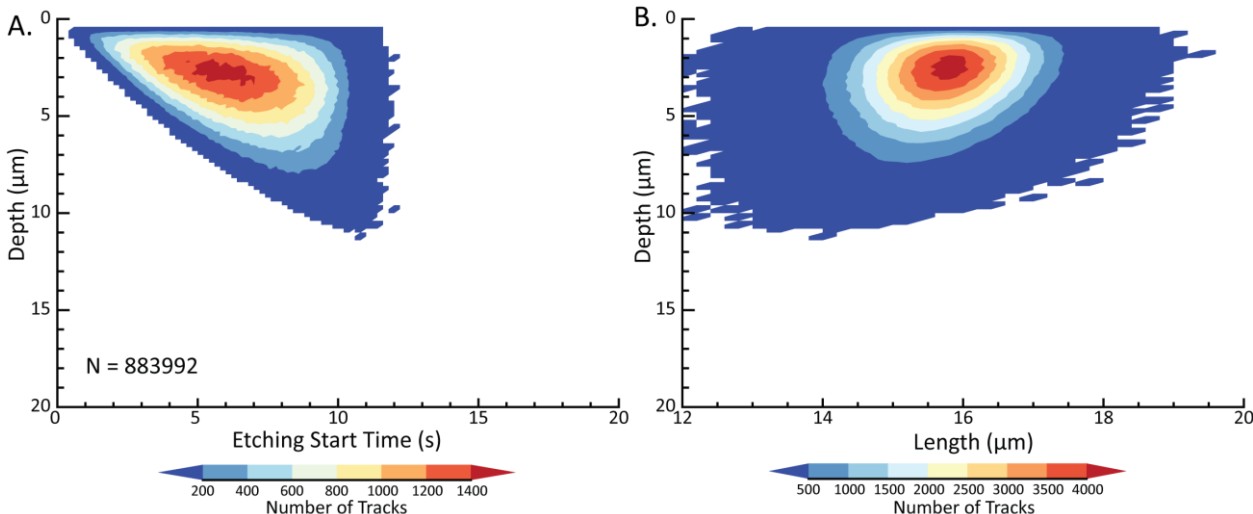

**Figure 11. Contour diagrams of model predictions of unannealed induced fission tracks selected by the analyst for measurement after etching the grain mount for 20 seconds. The selected tracks are the subset of the total population of confined tracks shown in Fig. 7B with tips that are sufficiently etched according to analyst criteria. The time-depth contours (A) indicate that, for this model, selected tracks began etching an average of ~6.5 seconds after etching commenced, and thus etched for an average of ~13.5 seconds before being measured. The contours of length versus depth (B) indicate that etched tracks are on average slightly shorter with increasing depth below the polished surface, as on average they have had less time to etch.**

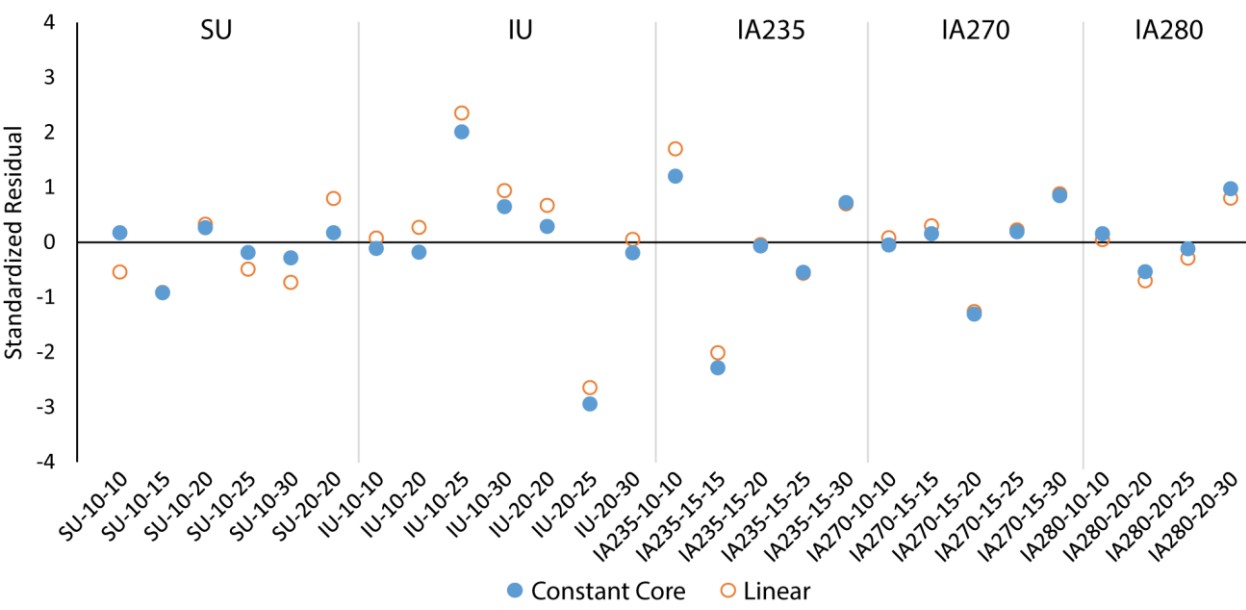

**Figure 12: Standardized residuals $(\bar{l} - \bar{l}_{est})/\sigma$ of model fits to each data set.**

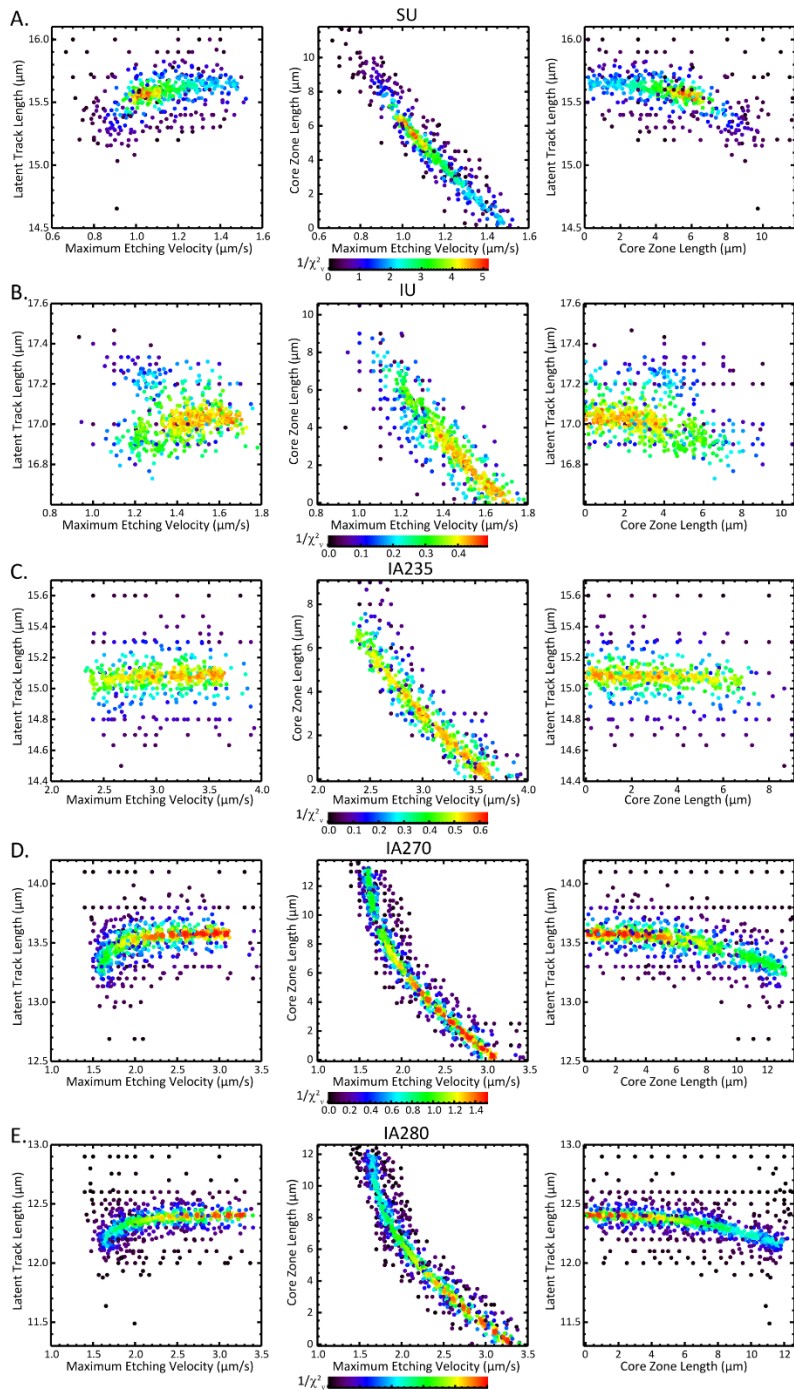

**Figure 13: Constant-core model parameter fits for each track type, showing correlations between variables. Each point represents a set of parameters tested during the fitting process, and colors indicate goodness of fit as $1/\chi^2_\nu$. Warmer colors indicate better fits, and any value near or above 1 indicates a fit to within the resolution of the measurements.**

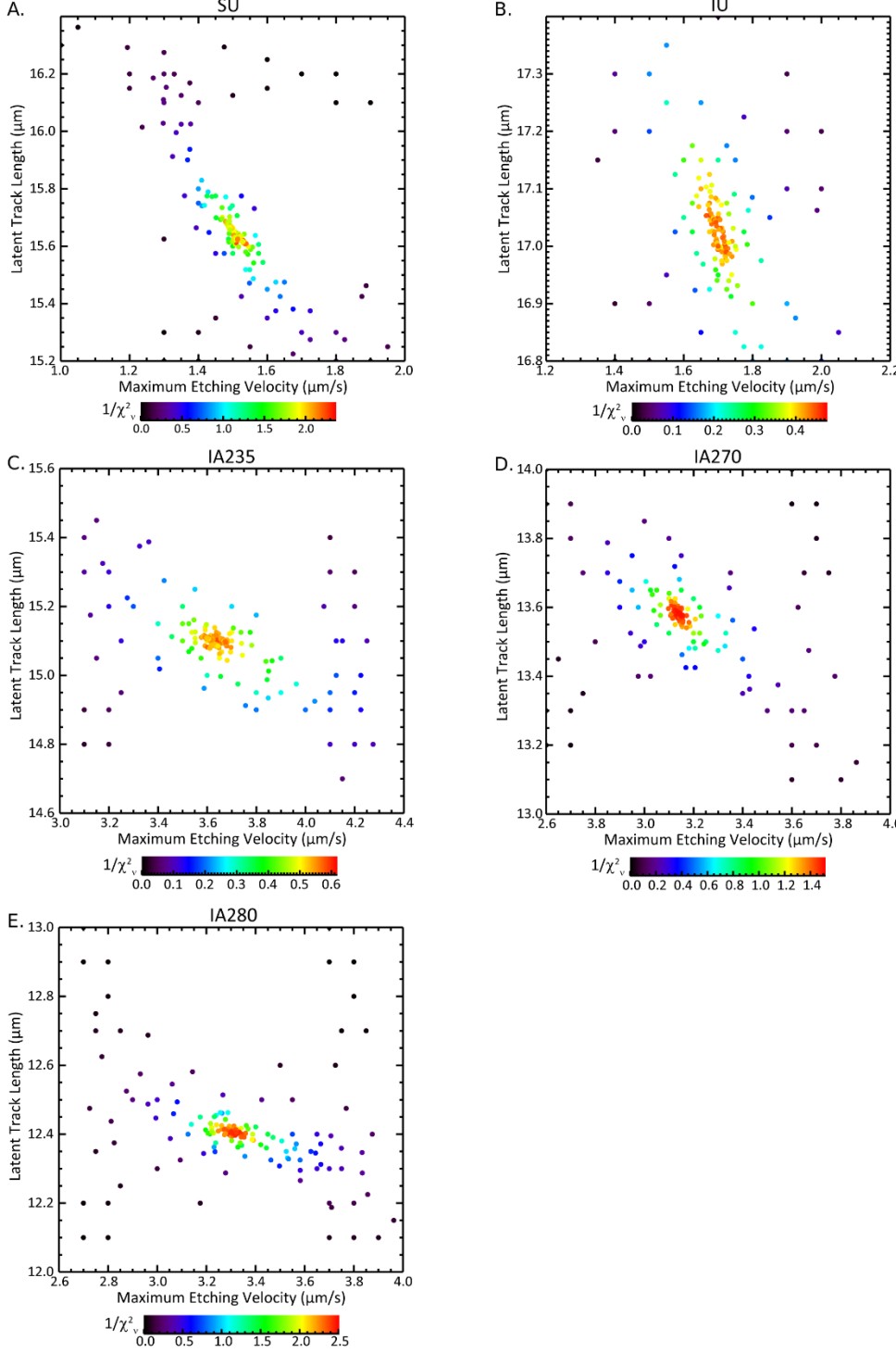

**Figure 14: Linear model parameter fits; see Figure 13 caption for explanation.**

945

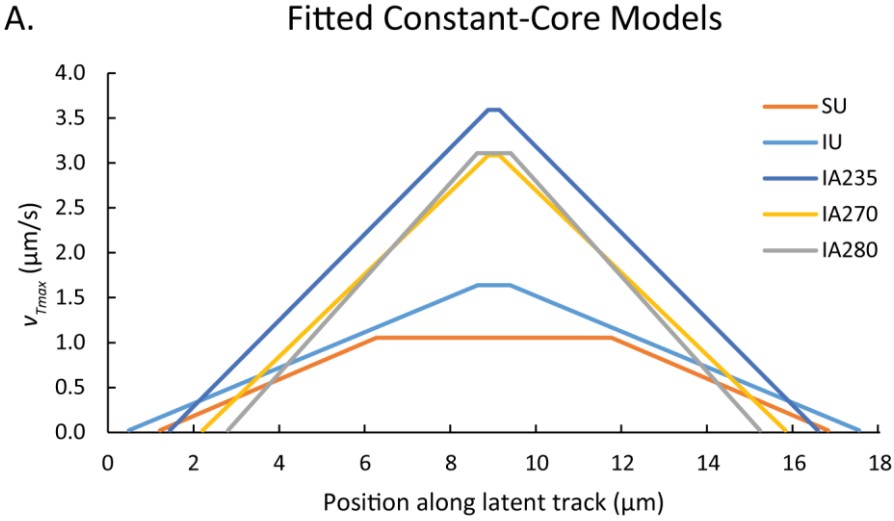

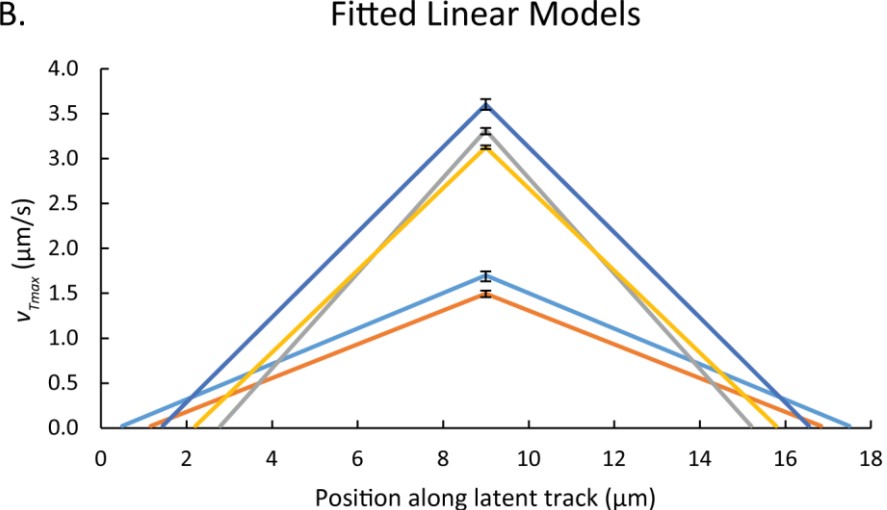

**Figure 15: Best-fit results for (A) Constant-core and (B) Linear etching structure models, showing that laboratory-annealed experiments feature maximum etching velocities ~2x higher than tracks without heat treatment. Error bars only shown for the maximum etching velocity in the Linear models, owing to the complex correlation between maximum velocity and core length in the Constant-core ones (Fig. 13).**

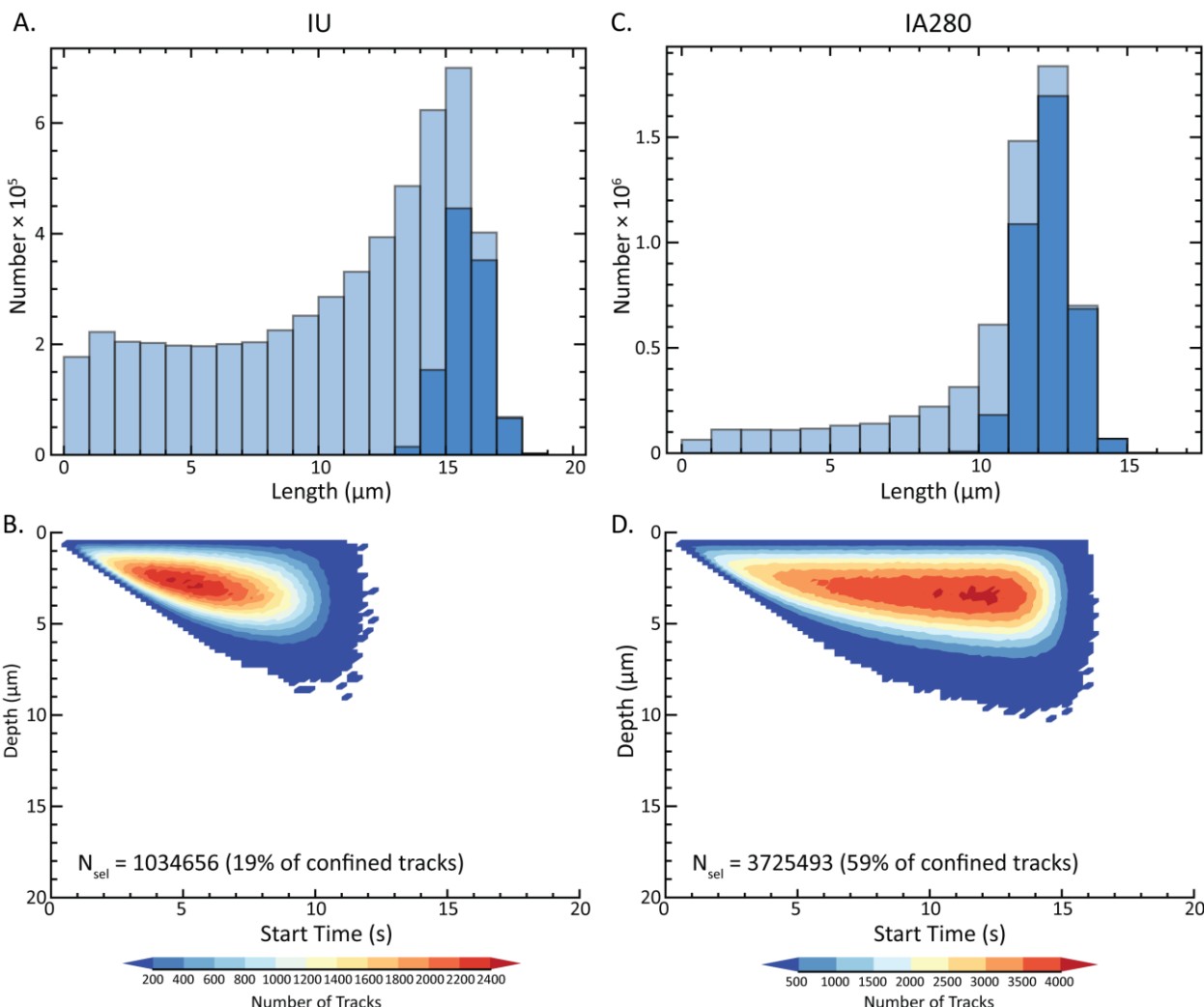

**Figure 16: Comparison of the relative efficiency of selecting different track types using a 20-second etching protocol.** In histograms (A and B), lighter bars show all etched tracks, and darker bars show tracks that pass selection criteria. For unannealed induced tracks, the great majority of confined tracks intersected have not been etched fully enough to be selected for measurement (A); those that are selected have etched for at least 8 seconds after being intersected (B). Annealed induced tracks are both shorter and faster-etching, leading to a much higher proportion of intersected tracks being selectable and smaller required etching times (C, D).

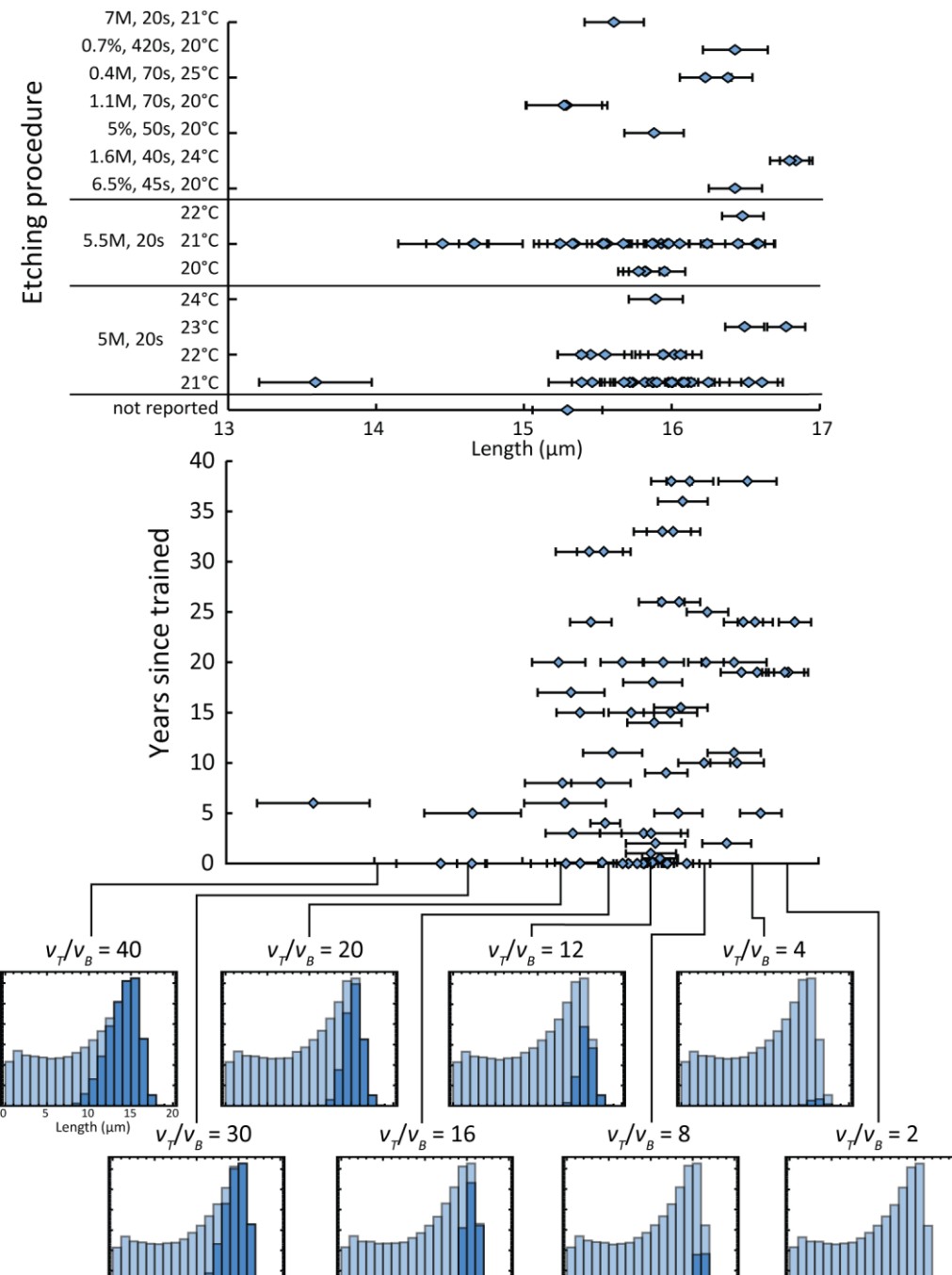

**Figure 17: Upper graphs (based on Ketcham et al., 2015, Figure 2C,D) show results of inter-laboratory exercise measuring unannealed induced tracks in Durango apatite, against etching procedure (upper graph) and years since trained in fission-track analysis (middle graph). Histograms below show prediction of Constant-core $v_T(x)$ model, varying only required $v_T/v_B$ for track selection; light bars are unselected tracks, dark bars are selected tracks.**

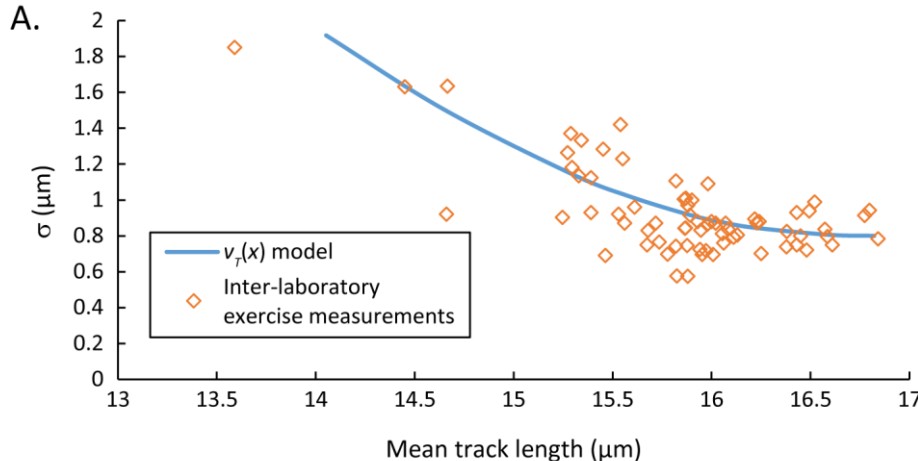

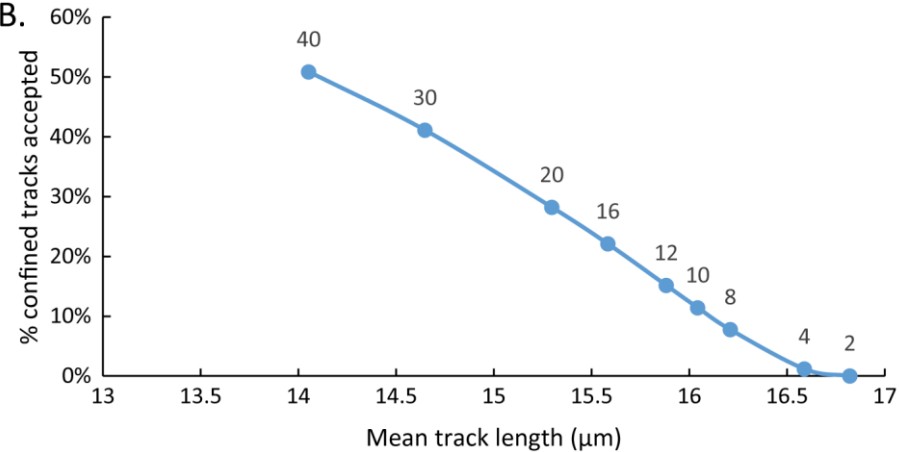

**Figure 18: Model relationship between analyst selection criteria (approximated as $v_T/v_B$ at the track tips) and dispersion and efficiency for unannealed induced tracks in Durango apatite. A. Points show the relationship between mean track lengths and standard deviation. Line shows prediction of $v_T(x)$ model assuming only variation in $v_T/v_B$, and a baseline 0.8 μm standard deviation of latent track length. B. Curve of efficiency (percent of all confined tracks intersected that are accepted) versus mean track length as it varies with $v_T/v_B$ (values above points).**

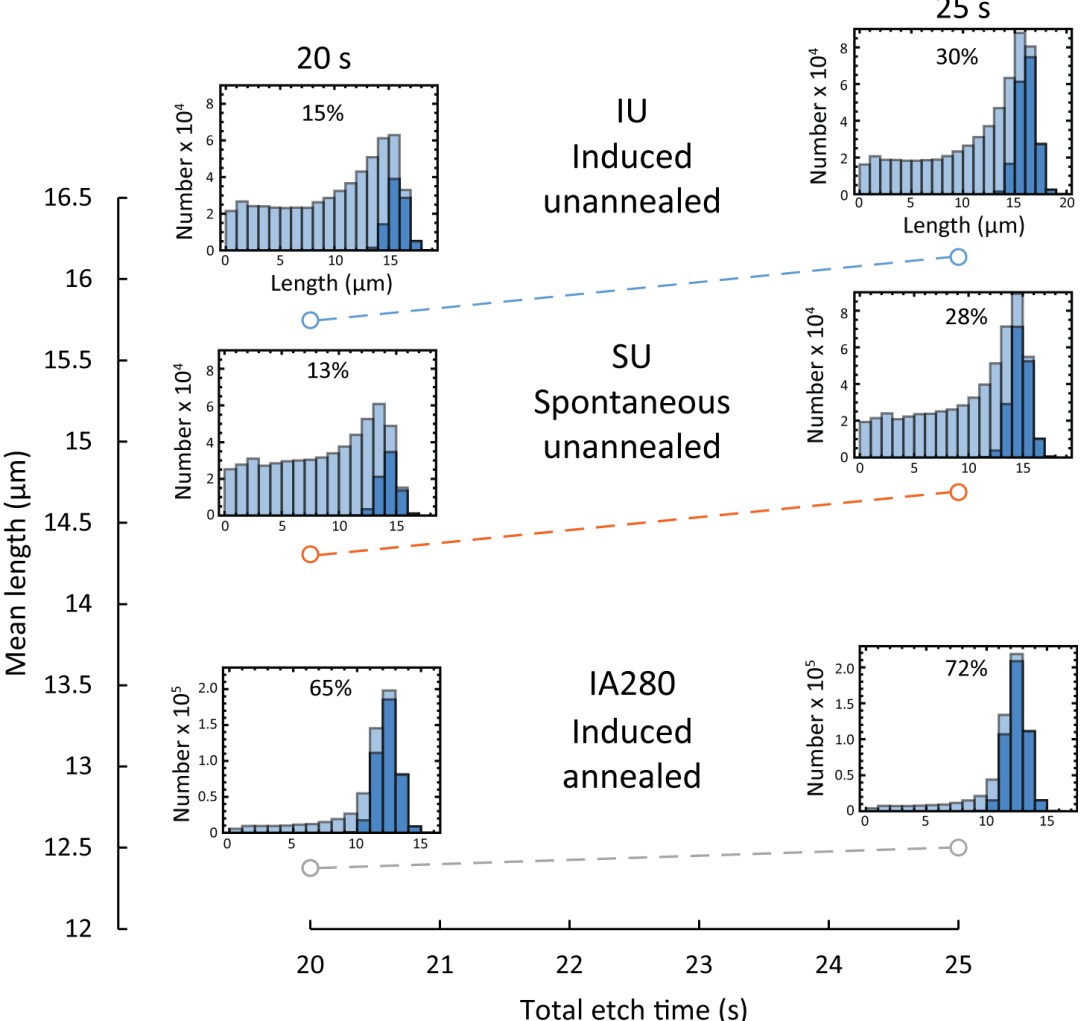

**Figure 19: Model predictions of effects of etching for 25 versus 20 seconds, for various track types, assuming selection criterion $v_T/v_B = 12$. Histograms show confined (light bars) and selected (dark bars) tracks for each point, and percentages reflect proportion of confined tracks selected. Note that SU model predictions do not presume Cf-irradiation, reducing selection efficiency compared to result reported in Table 1.**

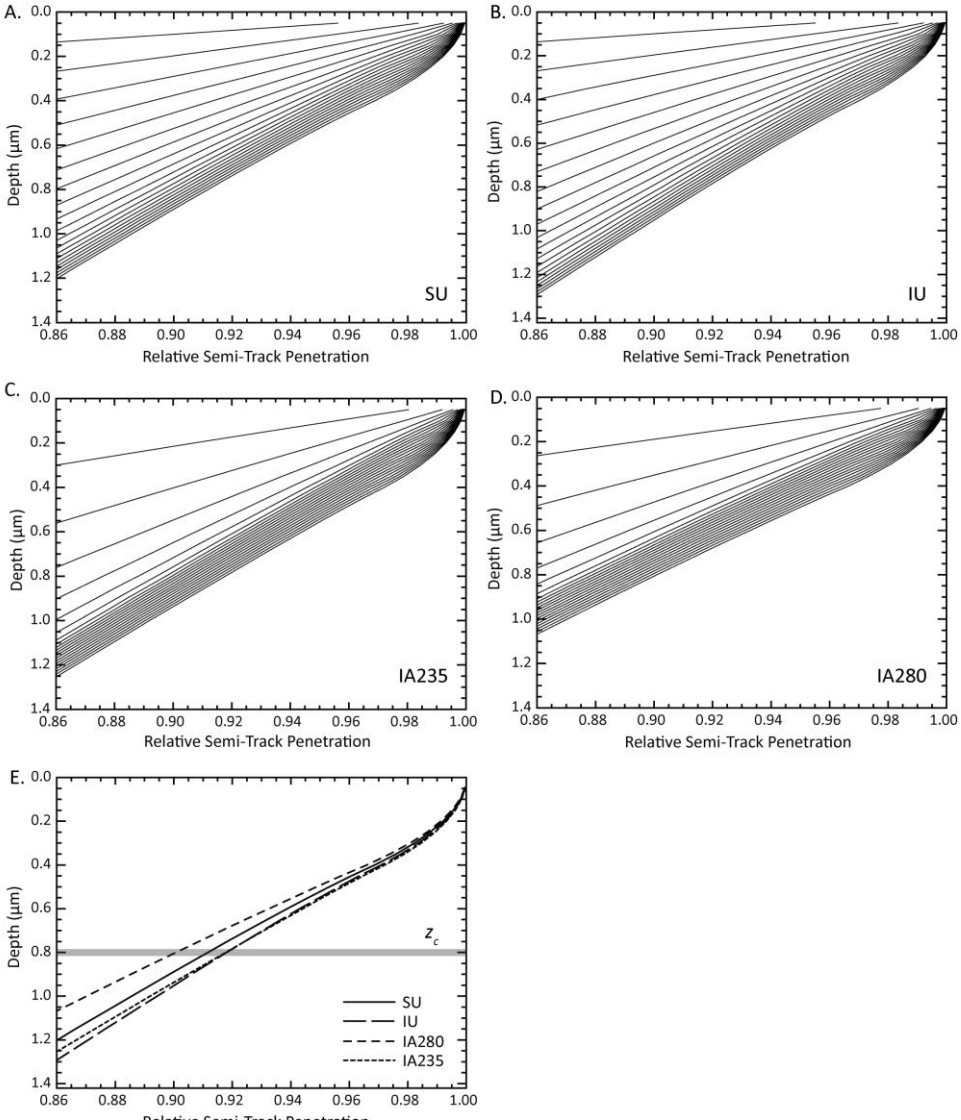

Figure 20: Penetration of randomly oriented semi-tracks into the grain interior from its polished surface, concentrating on the near-surface region (corresponding to upper right of Fig. 6A) for various track types: A. SU Unannealed spontaneous; B. IU Unannealed induced; C. IA235 Annealed induced (235°C, 24 h); D. IA280 Annealed Induced (280°C, 24 h). In A-D, each line represents penetration in 1-second increments from upper left to bottom right, from 1 to 20 seconds after etching begins. E. 20-second lines for each model, showing estimated degree of semi-track penetration after a standard 20-s etch. Depth of penetration may be the primary factor in determining whether a track is counted during age determination. Gray bar shows $z_c$, the estimated critical depth that a semi-track must penetrate to be confidently observed and counted (Jonckheere and Van Den Haute 2002).