# Peer review of "Confined fission track revelation in apatite: how it works and why it matters"

_Geochronology, 2020_

## Referee Comment (RC1) · Paul. Green (Referee) · 10 Nov 2020

**Review of Manuscript gchron 2020-31.  "Confined fission track revelation in apatite: how it works and why it matters" by Ketcham and Tamer, submitted to Gchron.**

**Review Accepted 21/10/2020**
**Review submitted 10/11/20**

Paul F. Green,
Director Technical
Geotrack International
10[th] November 2020

The paper under review (henceforth "the ms") presents results of a series of calculations and models regarding the revelation of fission tracks in apatite, with detailed discussion of the implications.  The subject matter falls within the remit of the Journal and would be of interest to a sub-set of readers.

However, I have to say that the ms is written in a style that I found very difficult to follow, and required several readings before I began to understand the overall aims of the paper and the reasoning behind the work described therein.  One of the main reasons for this is that the study relies on data and discussion presented in a companion paper (Tamer and Ketcham, 2020) and other works.  In addition, or perhaps as a result, many aspects of the work described in the ms are unexplained, or difficult to follow, simply being referenced to previous papers.  In fact, I only really began to get a clear idea of the rationale behind the work, together with the experimental design and significance of the analytical results after I had read a series of previous papers going back to Wauschkuhn et al (2015), which appears to form the foundation for these later studies.  Needless to say, I regard this as an unnecessary level of effort required to provide a review of a new contribution.  The text is also replete with vague expressions which have unclear implications, as well as inaccurate statements or misunderstandings and non-intuitive results that are accepted without question.  Some examples are cited below, but this is not an exhaustive list and the authors need to make an effort to explain their procedures and reasoning in a lot more detail.

Perhaps the biggest problem with the ms is that it is founded on a complete misconception of the application of apatite fission track thermochronology, emanating from the paper by Wauschkuhn et al (2015).  To quote from the manuscript under review (lines 26 – 28):

> *Measurements of laboratory-annealed spontaneous and induced fission tracks do not agree (Wauschkuhn et al., 2015b), leading to continuing uncertainty on the fidelity of induced tracks annealed in the  laboratory as proxies for spontaneous ones annealed at geological conditions over geological time scales.*

The paper by Wauschkuhn et al (2015) casts doubt on the use of kinetic models based on laboratory annealing studies of induced fission tracks in apatite to predict the behaviour of spontaneous tracks in geological conditions.  I note that if this doubt were to be well founded, then the very basis of the apatite fission track method, as implemented for example in the HeFTy software of the first author, would be invalid.  The Wauschkuhn et al (2015) study is based on the assumption that samples from the KTB borehole in Germany have undergone essentially isothermal annealing since the rock section cooled to temperatures at which tracks were retained.  This seems highly unlikely, given both the geological evolution of the region which contains a series of tectonic events during the Cenozoic (discussed in detail by Wauschkuhn et al., 2015) and also a number of previous studies which have revealed a more complex thermal history framework for samples from the borehole.  Wauschkuhn et al (2015) claim that the various aspects of the thermal history invoked to explain the apatite data in samples from shallow depths in the borehole cannot be believed, due to an absence of independent evidence.  But they then proceed to invoke a novel physical process for which there is also no independent evidence, to explain these data.  They thus simply replace one enigmatic process with another.  A borehole such as the KTB, where a range of factors may contribute to the thermal history of individual rock samples, cannot and should not be used to provide a geological test of annealing behaviour.

[Figure]

Thankfully, there is ample published evidence showing that the view expressed in the paragraph quoted above is without foundation. In my 1988 paper (cited in the ms) I described a series of experiments which showed that spontaneous tracks anneal in similar fashion to induced tracks once the induced tracks were shortened to a similar degree to the spontaneous tracks (this is not mentioned in the ms). This is an expression of the concept of equivalent time, originally introduced by Duddy et al. (1988) (not cited in the ms). A vitally important fact, that neither the paper under review nor any of the preceding papers have mentioned, is that Duddy et al. (1988) performed a series of careful laboratory experiments to validate this concept. In similar vein, a number of studies have shown that the predictions of fission track annealing models are consistent with measured data in a range of different geological conditions, e.g. Spiegel et al. (2007); Green and Duddy (2018), while other studies (e.g. Green and Duddy, 2010; Japsen et al., 2020; Green et al., 2013, 2018) have reported results from apatite fission track methods which are highly consistent with other paleo-thermal (e.g. vitrinite reflectance) and paleo-burial (e.g. sonic velocity) indicators. None of these studies confirming the equivalence of spontaneous and induced tracks and the validity of apatite fission track methods, have been cited in the ms or its preceding papers. Thus it is my view that the ms is based on a false premise and provides a biased view of the state of the technique. If spontaneous tracks did not behave in similar fashion to induced tracks then surely it would not be possible to use the technique to accurately predict track response in geological conditions. In my view, the failure of this and other papers to cite evidence which oppose their own views represents a serious failure to follow established practices in scientific publication.

The ms suffers from another serious shortcoming is that although not stated explicitly, it suggests that etching of confined tracks is the only consideration in determining appropriate etch times. In practice the etch time is also required to ensure that tracks which intersect the grain surface (i.e. those employed in determining a fission track age) are clearly revealed, to allow accurate counting without too many overlaps which might obscure some tracks. In addition, in routine application involving natural samples, particularly in the case of detrital grains in sandstones, grain mounts invariably contain apatites with a range of bulk etching rates, so that tracks in some grains will be etched to a greater degree than in others. Thus, the typical etch time of 20 seconds represents a balance between the need to sufficiently reveal tracks in grains with the lowest etch rates while not obscuring tracks in grains with higher etch rates. Accepting this view, and given the importance of measuring track lengths and ages in the same grains (although sadly this is often not done), it is difficult to see how the results of this study could find any practical application. Perhaps the authors should give some consideration as to how their work could be implemented in practical application.

Another issue, which the authors appear to discount, is that if measurements (including annealing experiments and data in field samples) are standardised to the same etching and measurement conditions, then the effects that they are discussing in this ms should not be a problem. This has certainly been our philosophy at Geotrack since 1982. In our early days, we performed several standardisation exercises with personnel from other laboratories to ensure that similar measurement protocols were adopted. However in recent years the explosion of laboratories around the word has resulted in a considerable number that have not shared that consistent approach.

Another motivation for the study described in the ms concerns the disappointing degree of reproducibility of track length measurements in inter-laboratory comparisons (lines 59-60). The authors may be better advised to focus on this aspect, rather than any perceived problems with the equivalence of spontaneous and induced tracks. A lack of standardisation between laboratories is likely to be a major feature in causing the observed lack of reproducibility.

Another important issue with one aspect of the analytical procedures adopted in the study concerns the measurement of under-etched tracks. The previous paper (Tamer and Ketcham, 2020b) discusses "a limited number of tracks that remained under-etched at 20 s but were measured anyway to adhere to the experiment design". This allows a certain degree of arbitrariness into the measurements which makes it difficult to know how to interpret differences from one etch step to the next. This issue is also referred to in the following list in relation to lines 74-77 and 194-196. The question arises of how the ends of under-etched tracks were measured, and what such eaurements mean. The resolution of the track end points becomes a critical issue and where the width of a track is less than 1 μm we are talking about measuring objects with a size close to the wavelength of light. So do these measurements indicate the actual etched length or only some part of that governed by resolution? Thus there is a question of how much the scatter in the measurements is due to problems in resolving track tips and how much is due to real differences in etch rates. This is also an issue with measurement of Dper, all of which are around the wavelength of light or less. This would account for the

degree of scatter in the Dper measurements in Figure 5c of Tamer and Ketcham (2020b). For these reasons, some discussion of how the authors dealt with this problem is required.

**Specific comments regarding the text:**

**Line 60:** *the region along tracks beyond where most current etching protocols reac**h***

The meaning of this is unclear.

**Lines 44-48:** *Virtually all mathematical treatments of track revelation, biasing, and the relationship between confined track length and track density (… … ……… …..) presume that tracks are line segments in space, all etched to their full extents once they are intersected.*

This is inaccurate, at least in terms of those papers involving Galbraith, Laslett and our group. These mathematical treatments are based on the assumption that only tracks that are etched to their full extent are measured. This is not the same as the statement quoted from the ms.

**Lines 73-74:** *Finally, to be measured a confined track must be etched sufficiently to be observed, the criteria for which will differ depending on the situation.*

What does the second part of that statement mean?

**Lines 74-77:** *For routine AFT analysis, where the analyst evaluates whether a track is sufficiently etched, the ends of the tracks need to be clearly visible, although this evaluation is analyst-specific. For measuring tracks in early steps of step-etching experiments, the criterion is simply that a track and its tips be visible enough to make a reasonable measurement.*

There is an underlying question here that pervades the ms as well as preceding papers; what criteria should be used in measuring tracks where the ends are not properly visible? This needs to be discussed – at least, the authors should discuss the approach that they used here.

**Lines 87-88:** *There was no clear indication of $v_B$ varying with track orientation (Tamer and Ketcham, 2020b)*

I find this puzzling. Surely the shape of the track openings in a prismatic surface shows that the bulk etch rate is higher in the direction of the c-axis than perpendicular to it. In this context I am puzzled as to why the authors have adopted the value of 0.022 µm/s for the bulk etch rate, when other samples from the same experiment define different growth rates, and this value is intermediate between the growth rate of Dpar and Dper.

**Lines 92-93:** *In cases where it was difficult to determine if an intersection truly occurred due to interfering features, we conservatively included it.*

I would say that a conservative approach (meaning careful, in my view) would be to exclude such values.

**Line 94:** *Section 4 heading. "4. The model"*

What model? Up to this point there has been no description of what is to be modelled or why. This needs to be explained and the basis of the model should be described. The reader should not have to work this out for him/herself.

**Line 121:** *Figure 2 illustrates the implied growth curves*

What is meant by "growth curves"? An informed reader can probably work this out but the content of the Figure should be explained more clearly. And in C,D, why are curves not shown for t=20 sec?

What values of VT, VB have been used in constructing this figure?

And in regard to Lines 129-130, It seems unlikely that differing intersection times would contribute significantly to differences between observers if other requirements of measurement (in terms of tip shape) are met, on the basis of this Figure.

**Lines 134-135:** *First, the relative probability of a track of latent length or half-length L crossing the surface*

Is it latent length or half length? The two terms suggest different meanings.

**Line 138:** *The semi-track penetration calculation…… ..*
**Line 150:** *Figure 3 shows examples of penetration and revelation rate.*

Both of these extracts from the text represent one of the most confusing aspects of the paper, at least as far as I am concerned. I cannot work out what Figure 3 is supposed to show, and the text provides no information that will help to work it out. Simply "penetration and revelation rate". These terms should be explained in detail.

**Line 165:**          *Figure 4 shows an example model of $10_7$ track intersections*

This Figure doesn't show a model. It shows results from a model. Maybe I'm old fashioned but I think text should be clearly written, describing what is shown accurately and clearly.

**Lines 194-196:**   *For step etching experiments with first steps shorter than 20 s, precise location of the tip is not a prerequisite for track selection, which is instead a matter of simple visibility. In the earliest stages of etching, tracks will be too thin to be observable in visible light. As they grow, they become more efficient at reflecting light, making them more detectable. However, when this occurs in the context of practical fission track analysis is unclear.*

This is not clear to me. To measure a track it is necessary to identify the end, so how did the authors decide where the ends were?

**Line 197:**           *Lacking a physical basis for determining when etched tracks begin to become visible*

This follows on from above. I find this section very confusing. Constructing an empirical bias relationship to allow for this aspect seems a very imprecise solution. Why should the same bias function apply in all cases?

**Lines 210-211:**   *Figure 7 shows the distribution of track lengths and etching times with depth below the surface*

This Figure does not show the distribution of lengths and etch times. It shows model predictions in terms of contours. Once again, precise description is important. Plus more description

**Lines 218-219:**   *Fitting the experimental data consists of posing an etching structure and using it to construct first a semi-track distribution and then a distribution of etched lengths*

What is being fitted to what here? This should be explained clearly.

**Lines 221-222:**   *We also include the option to incorporate some innate variation in latent track length. For the modeling in this paper we chose $\sigma$ of 0.5 µm, based on the scatter of lengths after **c**-axis projection to remove anisotropy effects*

Wouldn't a value of 0.8 or so be more appropriate, based on the distribution of induced track lengths which show no anisotropy?

It would also be interesting to run models with a single latent length, to investigate how much of the observed distribution of measured track lengths can be attributed simply to etching processes.

**Lines 229-232:**   *The step etch experiments only consist of 3-5 steps, making it difficult to meaningfully constrain models with 2-3 variables defining etching structure, in addition to biasing factors. To increase the amount of data, we paired data sets based on the same or equivalent tracks. Thus, we simultaneously fit data sets SE1 and SE2, both of which analyzed unannealed induced tracks, but with different initial etching steps.*

This is another of the most ambiguous passages in the ms. In what sense were the data sets paired? What does that mean? Why try to fit models with 2-3 variables to data with only 3 to 5 steps? Why not try simpler models or use more etch steps to resolve these problems? And why combine data with different initial etch times? More control on experimental design would ameliorate these issues.

**Lines 241-245:**   *During fitting of the unannealed induced track data (SE1 and SE2), it became apparent that one pair of data points were exerting outsized control on the result. Since the 15-20 s and 20-25 s steps for experiment SE2 feature the very similar mean length reduction and thus almost the same rate, in the context of our model forms assuming a linear rate decrease fits were forced to split this difference to the exclusion of closely fitting the rest of the data for SE1 and SE2. We thus excluded the 15-s mean length for SE2, effectively making the second etching step go from 10 s to 20 s*

I wonder if this presages graver issues within the data set. Why should this dataset be problematical? To simply reject these data seems rather arbitrary.

**Line 268:** *Figure 9 shows the parameter fits for Constant-core models, and Figure 10 for Linear models*

Here is the biggest single problem that I had with reviewing the ms. What on earth does this mean? This Figure shows a series of cross plots, without any description of what is being plotted or why, and what they mean. What does each dot represent in each plot? I find it totally unacceptable that the authors leave it to the reader to work this out.

**Line 287:** *It is remarkable that we are able to reproduce the mean length data in each etching step of all experiments simultaneously using these simple models of etching structure*

Give the number of adjustable parameters, each of which is free to vary as required until a fit is obtained, I don't see this as remarkable at all. And regarding the statement that mean length is reproduced, I guess that Figure 8 is supposed to show this, but a plot of measured vs fitted parameters would seem to be a more graphic depiction of this.

**Line 329:** *Fossil and unannealed induced tracks have slow core etching rates, while all annealed induced tracks have far higher rates.*

It is counter-intuitive that annealing would result in an increase in etch rates within the residual track region, particularly when spontaneous tracks do not show enhanced track rates. Such behaviour would suggest that heating increases the degree of damage, when all previous experience with lattice damage suggests that heating leads to a decrease in damage intensity. This suggests that there is a flaw in either the experimental design or the model design from which this conclusion was derived. The authors seem to accept this observation at face value, but this unusual result deserves further investigation before any firm conclusions can be drawn.

The fitted values of $V_{Tmax}$ and $\Delta x_{Tmax}$ in Table 2 show no consistent trends that might suggest these results have any meaning. It would be reasonable to expect these values to vary in a consistent manner as the degree of annealing progresses, but for annealing temperatures of 235, 270 and 280°C, $V_{Tmax}$ varies from 4.0 to 1.77 and back to 3.36, while $\Delta x_{Tmax}$ changes from 0.19 to 11.35 and back to 0.76. It is hard to imagine why the distribution of damage should go through such contortions within a limited range of annealing temperatures. These results suggest that the fitted values are simply empirical numbers without any real meaning. For the Linear model, values of $V_{Tmax}$ change from 4.09 to 3.54 to 3.59 over the same range, compared to the value for induced tracks of 1.70, and again it is difficult to see how these numbers can be accepted as having any physical reality. This I turn raises questions regarding the significance of other aspects of the results.

Incidentally, it would be useful to show error bars in Figure 11, as per normal practice.

**Lines 331-332:** *This may be responsible for some component of the mismatch in annealing fossil versus induced tracks in laboratory experiments (Wauschkuhn et al., 2015a),*

As mentioned earlier, there is plenty of published evidence that there is no mis-match. The authors appear to have adopted this observation with no critical assessment. At the very least, the authors should mention that evidence to the contrary exists, and preferably also to explain why they discount that evidence.

**Line 337:** *treatments of fission-track lengths are all based on a line segment model*

Any mention of the line segment model should refer to Rex Galbraith's (2005) book.

**Lines 457-457:** *The $v_T(x)$ model framework provides a quantitative basis for evaluating whether the etching procedures used today are the most effective at their goal, which is to provide high numbers of reproducible and informative measurements to constrain thermal histories.*

I would dispute that the goal is simply to provide high numbers of measurements. Surely the goal should be to obtain data that provides the most accurate thermal history constraints possible. This depends not only on track length measurements but also on fission track age determinations and also on allowing for different etching rates in apatite grains on a grain mount. The $v_T(x)$ *model* described in the ms focusses on only one of these aspects.

[Figure]

**References not cited in the ms under review:**

Duddy, I.R., Green, P.F., Laslett, G.M., 1988. Thermal annealing of fission tracks in apatite 3. Variable temperature behaviour. Chem. Geol. 73, 25–38.

Green, P.F. and Duddy, I.R. 2018.  Apatite (U-Th-Sm)/He thermochronology on the wrong side of the tracks Chemical Geology 488, 21–33

Green, P.F., J Duddy, I.R. & Japsen, P., 2018.  Multiple episodes of regional exhumation and inversion identified in the UK Southern North Sea based on integration of palaeothermal and palaeoburial indicators. In: Bowman, M. & Levell, B. (eds) 2018. Petroleum Geology of NW Europe: 50 Years of Learning – Proceedings of the 8th Petroleum Geology Conference, 47–65.

Spiegel , C, Kohn, B.P., Raza, A., Raiuner, T., Gleadow, A.J.W. 2007. The effect of low temperature exposure on apatite fission track stability: a natural annealing experiment in the deep ocean. Geochimica et Cosmochimica Acta 71:4512- 4537.

---

## Referee Comment (RC2) · Raymond Jonkheere (Referee) · 18 Nov 2020

Review of gchron-2020-31

**Confined fission track revelation in apatite: how it works and why it matters**

Richard A. Ketcham and Murat T. Tamer

We should declare that we have a competing model, and thus a possible prejudice against, or in favour, of the reviewed manuscript. It was our intention not to review it, but we accepted at the insistence of the authors, and because of its interest to those in our group involved in related studies. Our evaluation is critical but we believe not unfair. The public comments will set us right, if need be.

**General comments**

This manuscript reports calculations and numerical simulations of fission-track etching in apatite based on a variable track etch rate $v_T(x)$, isotropic apatite bulk etch rate $v_B$, and certain track selection criteria. It concludes with a discussion of the implications for apatite fission-track dating and modelling.

An astounding fact considering the first author's impressive publication record is that the manuscript is almost unreadable. Even with foreknowledge of the concepts involved and of earlier publications, we struggled to understand certain sections. This was not helped by the constant use of ill-defined or undefined notions and meaningless word strings (e.g., semi-track penetration calculation). There appears to be no attempt to be understood, let alone to be exact, and readers less familiar with the subject can make heads nor tails of the manuscript, much less evaluate its significance.

A second striking fact is that the manuscript presents a track etch model that appears to exist in a vacuum, disconnected from all existing knowledge of latent tracks, etching of tracks and minerals, and from all published step-etch data, except for those of Tamer and Ketcham (2020), e.g., for the 5.5 M etch (Carlson et al., 1999; Jonckheere et al., 2007, 2017; Tamer et al., 2019), the comparable 5.0 M etch (Laslett et al., 1984; Green et al., 1986; Barbarand et al., 2003; Moreira et al., 2010), and, given that the model is a first-order approximation, for lower etchant concentrations as well.

One respect in which we found the manuscript lacking is a *post-factum* assessment of what it all signifies. The known statistical properties of surface tracks and confined tracks are a basic ingredient of the model. A second ingredient is the principle that to etch a track-in-track, the etchant penetrates down a host track, crosses to the confined track and etches it from their intersection point outwards. It seems not unlikely that this explains the broad traits of Figures 2-4, and, with scaling based on measured lengths, some experimental results as well, without the need for a specific $v_T(x)$ model. A foreseeable consequence is an excess of "under-etched" tracks which is culled down to ≪10% based on for the most part unexplained *ad hoc* selection criteria. This results in etch rates ($v_{Tmax}$) ranging by a factor of ~3 and core lengths ($\Delta x_{Tmax}$) by a factor of >50 between samples. Dazed by the mathematical acrobatics, the reader is left wondering what it all means. Is this a hard numerical result, and which of the several factors above weighed most on the outcome, the fundamental geometrical configuration, the etch rate model, or the selection criteria?

From our standpoint, the proposed model is too approximate and dependent on questionable assumptions to be confident about the numerical results. On the other hand, it highlights the paramount importance of etching protocols and selection criteria for apatite fission-track dating and modelling.

**Specific comments**

4.1. *Etching structure.* Given the title: "*Confined fission track revelation …*", and Figure 1, showing a confined track with two endpoints, one could be forgiven for thinking that the calculations refer to etching of confined tracks. It turns out, after a full page of baffling equations suspended in mid-air, that the calculation was not that of the lengths of confined tracks at all but of those of a semi-tracks. The confined track length is then obtained as the sum of complementary semi-tracks. One needs to inform the reader from the start about what is going on, i.e. how one is going about solving a problem. With that information and a full explanation of all the symbols (and simpler notation) one need not bother about (1)-(5) as the conclusions are rather obvious from the outset.

Figure 2 suggests a more interesting observation that is not discussed: the results for the "Constant-core model" (A, C) and the "Linear model" (B, D) are identical. It is possible that this is due to a conservation principle, in the sense that if A and B are far enough separated, then the time to etch from A to B is independent of the etch rate function $v_T(x)$ between A and B, but only depends on the area under $v_T(x)$. A consequence of this observation is that if the two end-member models predict the same evolution of (semi-)track length with etch time (cf. Figure 2), then how are step-etch mean-length data going to distinguish between them, or between them and other models?

4.2. *Semi-track penetration and confined track revelation.* Much as the preceding discussion of the track etch rate and apatite etching, the lead up to Figure 3 (line 150) reads as a perfunctory dismissal of the accomplishments of earlier scientist as too trivial to cite. This entire section could have read: "*Semi-tracks have a sin(2δ) dip angle distribution and a homogeneous etchable length distribution (Dakowski, 1978; Laslett et al., 1982)*". Then some modelling happens and something is plotted in Figure 3: *relative semi-track penetration* and *relative confined track revelation* against *depth* (below the initial or etched surface?). Even with foreknowledge it is not possible to guess at the significance of Figure 3. What is concluded from Figure 3 does not seem to require modelling at all: "*~10% of tracks reach ≲1 μm*" and "*$^{252}$Cf tracks are shorter, … but efficient*". Yes, see Dakowski (1978): ¾ of semi-tracks reach ≤½ a full track length deep, and most have near-zero depths.

4.3 *Confined track intersection.* Line 158: "*weighted as cos δ*": see Dakowski (1978). It is hard to comment; on the one hand the concept is simple (hence likely the shorthand reporting) but, on the other, it is impossible to puzzle out what *exactly* has been done or what *exactly* is shown in Figure 4 (CDF?). Its discussion again refers to issues that seem to require no modelling at all: "*only about half of the tracks remain after surface intersecting track are excluded*". Yes, a slab of thickness $L$ below a unit of surface contains $L\,N$ tracks of which ½ $L\,N$ intersect it (Fleischer et al., 1975).

4.4 *Confined track selection.* Fleischer et al. (1969), Paretzke et al. (1973) and a great number of other scientists published calculations of etch-time dependent track profiles for varying $v_T(x)$ and isotropic $v_B$.

Line 185: the isotropic-$v_B$ case is not "simplified" compared to Aslanian et al. (in press), but contradicts it, as well as fundamental etching theories (e.g., Heimann, 1975). The definition of "bulk etch rate" is different.

Line 198: "*we constructed … an operator bias function*". Despite the reserve one has to ask: where does it end? It brings to mind Murphy's variable constant, which multiplied with anything gives the desired result. How decisively does Figure 6A differ from a straight ramp, or even a step function? If there is an actual basis for it, please discuss it; if not, all that follows from 6A is a possible artefact.

Figure 5 is to all intents and purposes identical with Figure 4 of Fleischer et al. (1969); the equations are also not dissimilar. It is worth noting that the assumed gradual increase of track length is contradicted by the step-etch data of Jonckheere et al. (2017, in part measured by the present co-author), demonstrating that confined track lengths increase in fits and starts, reflecting a discontinuous latent-track structure, as reported in numerous apatite studies (Paul and Fitzgerald, 1992; Paul, 1993; Li et al., 2010; 2011; 2012; 2014, etc.), as well as nigh the entire literature on latent ion tracks (not counting the amusing contributions from the Canberra group who posit a cylindrical track and then shoot their ions straight through the crystal so that the track ends are

missing). One can of course think of $v_T(x)$ as averaging over a certain section, but this should be stated explicitly.

Figure 6. It is worth commenting on the true significance of the observed "excellent agreement". It would appear that the invasive, *ad hoc* filters (6A and 6D) would transform just about any underlying distributions (light blue in 6B and 6E) into almost negligible residual fractions (dark blue in 6B and 6E) with the general Gaussian appearance of the measurements presented for comparison.

Figure 7 is interesting, but, of course, underlies the combined reservations concerning the steps leading up to it.

A puzzling fact about section 4 is that 4.1 sets up two competing *ad hoc* models for the variation of the etch rate $v_T(x)$ along the tracks (in fact one end-member and an generalized member of a single model, which is assumed to be the same for tracks produced by all fission fragments and energies). Sections 4.2 to 4.4 then present modelling results, which seem not to require the equations in 4.1, and never again mention which model (or parameter fit) was used, much less how the models differ.

4.5 *Fitting step-etching data*. This section presents a complicated account of different modelling issues. It would help to reproduce Figure 1 of Tamer and Ketcham (2020) to present the reader with visual map of the data. It is worrying that there appears to be a need to exclude one data point, that the iterative fitting procedure does not seem to converge towards unique best-fit solutions, and that such solutions appear to not be strictly reproducible. Please discuss what that actually signifies.

5. *Results*. One cannot help thinking that the discussion of Figures 8-11 could be so much better structured if, instead of the artificial opposition of "two models", one of which is an end member of the other, there would have been just one constant-core model with two end members, a linear-$v_T(x)$ model ($\Delta x_{Tmax} = 0$) and a constant-$v_T(x)$ model ($\Delta x_{Tmax} = L_{lat}$). The latter is at least relevant to the ceremonial dismissal of the model unfairly attributed to Laslett et al. (1984) in Tamer et al. (2019).

6. *Discussion*. "*It is remarkable that we are able to reproduce ... using these simple models of etching structure*". Yes indeed, on condition (1) that the predictions in the first place depend on the assumed model, and in negligible measure on the geometrical framework and assumptions concerning track selection, (2) there is a negligible self-referral between the data used for calibration of the models and those adduced for testing them. Perhaps the careful term "reproduce" reflects concern about the fact that these overlap, which, in our view, affects the significance of the results.

"*We take these successes as an indication of the overall correctness of our characterization of confined track etching*". This is too facile; it could be a Ptolemaic model, which, admittedly based on false assumptions, nevertheless manages to account for isolated appearances (individual experiments) given suitable eccentricities and epicycles. The authors need to make a serious effort to make transparent the network of mutual influences between the geometrical framework, the actual etch model, the various selection criteria, calibration data, and the data used for verification, so as to to reassure the reader that their model is not in effect a self-sustaining mirage hovering in mid-air.

It could be argued in favour of the model that, following annealing and etching together, pre-annealed induced tracks are longer than fossil tracks (Wauschkuhn et al., 2015; Figure 15). Perhaps they etch faster?

Line 358: "*A thorough re-evaluation of this biasing based on measurements...*". This appears to overlook the fact that the (length) bias function implemented in modelling programs is based on experimental data.

Lines 370-380. "*As $v_T/v_B$ rises the proportion of accepted tracks increases ...*". Do we interpret this accurately as: "Including shorter tracks decreases the mean length and increases the standard deviation (range)"?

Line 386: "*... the mean latent track length of 17 μm indicated by our data and model*". More reliable estimates bracket the average latent track length in apatite between ~18 and ~21 μm (Jonckheere, 2003a; 2003b; Jonckheere et al., 2017). A discussion of established concepts such as *etch-*

*ing threshold* and *range deficit* relevant to this issue. On a related point: one could cite Figure 1 of Jonckheere (2003a) in support of the linear model, on the assumption that $v_T(x)$ is proportional to *dE/dx*.

Line 406: "*tip appearance depends on ... mount preparation, polishing, and cleaning technique, microscope optics, and captured image quality*". This should include etching protocol and apatite composition.

Lines 450-455: "*These differences in latent versus measured lengths ...*". The discussion omits established facts such as the *etching threshold* and *range deficit* (Fleischer et al., 1975; Iwano and Danhara, 1998).

Lines 450-455: "*By stopping etching as soon as the curve of length versus time in step-etch experiments was reached ...*". No, ... as soon as the data showed a constant rate of increase of the mean track length, taken to be the average bulk etch rate. The same principle used for calculating your bulk etch rate.

Lines 460-462: "*... a rate defined by $v_B$, which may be obtainable from etch figure measurements parallel and perpendicular to the c axis (Tamer and Ketcham, 2020a)*". Long-established etching theories hold that measurements of surface features are not suitable for estimating etch rates (Jonckheere et al., 2019).

Lines 497-498: "*Figure 17 shows the near-surface portion of the penetration model ...*": Figures 3 and 17 remain obscure; they could be etch-time-dependent depth distributions, but a better explanation is needed.

Lines 524-525: "*If ... geological annealing results in different etching rates than laboratory annealing ...*". It could be argued that, after simultaneous annealing, pre-annealed induced tracks are longer than fossil tracks (Wauschkuhn et al., 2015; Figure 15), perhaps because they do etch faster?

Lines 530-549: The outlook section reads as an appeal for more and better data and community involvement, but is rather short on scientific detail and tends to drift off into a dreamy utopia. It can be deleted. It is at this stage more important to look back and re-evaluate the results, than to look forward.

Line 551: "*a range of detailed step-etching data ...*". This is a rather exaggerated designation of a handful of mean track lengths (Table 1) and some intersection depths. The imbalance between the *"comprehensive"* model and the limited data for calibration and testing is a conspicuous weakness of this work.

Line 554: "Along-track etching velocity *... varies ... among fission tracks ...*". The intended meaning is probably among different *types of tracks* (spontaneous, induced, annealed), not between individual tracks.

Line 563: *"Most variation ..."*. Barring sample preparation accidents, does one need an etch model to conclude that lower mean lengths and higher standard deviations result from including shorter tracks? The $v_T/v_B$ criterion is not progress, because it applies to isotropic detectors, not to apatite.

Lines 575-613: "*Appendix: $v_T(x)$ Etching Model Equations*". This appendix contains a long list of repetitive piecewise integrals of simple functions, that are not used to obtain the results in this manuscript, instead produced by numerical simulations. The appendix thus seems superfluous to the manuscript, and may be either deleted or included as a supplement. We have not checked these equations. It is on reflection possible that the modelling uses the equations but this is nowhere made  clear.

Matters of detail relating to the writing, figures and captions are too numerous to be addressed at this stage.

Freiberg, 17 November 2020,

R. Jonckheere
B. Wauschkuhn

---

## Author Comment (AC1) · 13 Dec 2020

**Answers to comments on manuscript Confined fission track revelation in apatite: how it works and why it matters**

Richard Ketcham and Murat Tamer

Our answers are below in *red italics*.

**Answers to comments by Paul Green**

The paper under review (henceforth "the ms") presents results of a series of calculations and models regarding the revelation of fission tracks in apatite, with detailed discussion of the implications. The subject matter falls within the remit of the Journal and would be of interest to a sub-set of readers. However, I have to say that the ms is written in a style that I found very difficult to follow, and required several readings before I began to understand the overall aims of the paper and the reasoning behind the work described therein. One of the main reasons for this is that the study relies on data and discussion presented in a companion paper (Tamer and Ketcham, 2020) and other works. In addition, or perhaps as a result, many aspects of the work described in the ms are unexplained, or difficult to follow, simply being referenced to previous papers. In fact, I only really began to get a clear idea of the rationale behind the work, together with the experimental design and significance of the analytical results after I had read a series of previous papers going back to Wauschkuhn et al (2015), which appears to form the foundation for these later studies. Needless to say, I regard this as an unnecessary level of effort required to provide a review of a new contribution. The text is also replete with vague expressions which have unclear implications, as well as inaccurate statements or misunderstandings and non-intuitive results that are accepted without question. Some examples are cited below, but this is not an exhaustive list and the authors need to make an effort to explain their procedures and reasoning in a lot more detail.

*This contribution is indeed built upon and/or informed by recent studies, as well as older ones; this is the way things normally work. We have attempted to balance explanation with brevity, and will endeavor to do better during revisions.*

Perhaps the biggest problem with the ms is that it is founded on a complete misconception of the application of apatite fission track thermochronology, emanating from the paper by Wauschkuhn et al (2015). To quote from the manuscript under review (lines 26 – 28):

*Measurements of laboratory-annealed spontaneous and induced fission tracks do not agree (Wauschkuhn et al., 2015b), leading to continuing uncertainty on the fidelity of induced tracks annealed in the laboratory as proxies for spontaneous ones annealed at geological conditions over geological time scales.*

The paper by Wauschkuhn et al (2015) casts doubt on the use of kinetic models based on laboratory annealing studies of induced fission tracks in apatite to predict the behaviour of spontaneous tracks in geological conditions. I note that if this doubt were to be well founded, then the very basis of the apatite fission track method, as implemented for example in the HeFTy software of the first author, would be invalid. The Wauschkuhn et al (2015) study is based on the assumption that samples from the KTB borehole in Germany have undergone essentially isothermal annealing since the rock section cooled to temperatures at which tracks were retained. This seems highly unlikely, given both the geological evolution of the region which contains a series of tectonic events during the Cenozoic (discussed in detail by Wauschkuhn et al., 2015) and also a number of previous studies which have revealed a more complex thermal history framework for samples from the borehole. Wauschkuhn et al (2015) claim that the various aspects of the thermal history invoked to explain the apatite data in samples from shallow depths in the borehole cannot be believed, due to an absence of independent evidence. But they then proceed to invoke a novel physical process for which there is also no independent evidence, to explain these data. They thus simply replace one enigmatic process with another. A borehole such as the KTB, where a range of factors may contribute to the thermal history of individual rock samples, cannot and should not be used to provide a geological test of annealing behaviour.

*We agree that interpreting the KTB data set is very difficult, and we do not agree with all of the analysis and interpretations by Wauschkuhn et al (2015). At the same time, we appreciate the long-term project of that research group to improve the foundations of apatite fission-track analysis, which includes revisiting assumptions that are often taken for granted but may not hold up under closer scrutiny.*

*However, the reviewer is misdirected; we were not referring to the KTB data and its interpretation at all, but instead to the experimental data set reported in their Table 2 and Figure 15 (we will clarify this during revision). That small study constitutes an examination of the equivalent time principle not undertaken by Duddy et al. (1988) in comparing the annealing behavior of fossil to induced tracks, by first pre-annealing induced tracks to the length of fossil ones, and then annealing fossil and pre-annealed induced tracks side by side. In these experiments, fossil tracks annealed more quickly than induced ones in these subsequent steps, seeming to violate the equivalent time hypothesis that tracks of the same length behave the same way, "totally independent of the conditions of temperature and time which caused the prior annealing."*

*The reviewer appears to agree with Wauschkuhn et al. (2015) in claiming that any failure of the equivalent time hypothesis, or some other assumptions embedded into fission-track thermal history modeling by HeFTy and other software, invalidates the entire method. We believe that this is an overreaction. As we know, "all models are wrong, but some are useful" (George Box, co-creator of the Box-Cox transform used by Laslett et al. (1987)); even if an assumption is not precisely*

*correct, the question remains of how much this incorrectness affects model predictions. The best way to answer this question is to study the problems more closely.*

Thankfully, there is ample published evidence showing that the view expressed in the paragraph quoted above is without foundation. In my 1988 paper (cited in the ms) I described a series of experiments which showed that spontaneous tracks anneal in similar fashion to induced tracks once the induced tracks were shortened to a similar degree to the spontaneous tracks (this is not mentioned in the ms).

*The Green (1988) data do not include a pre-annealing step equivalent to that reported in Wauschkuhn et al. (2015), which is why we did not mention it. The experiment in Wauschkuhn was designed very specifically to test the equivalent time hypothesis in a way Green (1988) did not.*

This is an expression of the concept of equivalent time, originally introduced by Duddy et al. (1988) (not cited in the ms). A vitally important fact, that neither the paper under review nor any of the preceding papers have mentioned, is that Duddy et al. (1988) performed a series of careful laboratory experiments to validate this concept.

*We discussed our data with respect to Duddy et al. (1988) in our previous paper, but we can bring some of that discussion forward here to reinforce it. We agree that Duddy et al (1988) is an effective verification of the equivalent time hypothesis, but only on laboratory-annealed induced tracks. It did not, and could not, test the equivalent time hypothesis against geological annealing. In our previous paper, we showed evidence that the etching structure of spontaneous and equivalent-length-annealed tracks are different, and in this paper, we further quantify that laboratory annealing substantially accelerates etching.*

*Does these various observations invalidate the equivalent time hypothesis? In a sense, perhaps, but not necessarily catastrophically. Perhaps it is roughly true on both geological time scales and laboratory time scales, but not between the two, because some processes operate on one time or temperature scale and not the other, such as background radiation damage.*

In similar vein, a number of studies have shown that the predictions of fission track annealing models are consistent with measured data in a range of different geological conditions, e.g. Spiegel et al. (2007); Green and Duddy (2018), while other studies (e.g. Green and Duddy, 2010; Japsen et al., 2020; Green et al., 2013, 2018) have reported results from apatite fission track methods which are highly consistent with other paleo-thermal (e.g. vitrinite reflectance) and paleo-burial (e.g. sonic velocity) indicators. None of these studies confirming the equivalence of spontaneous and induced tracks and the validity of apatite fission track methods, have been cited in the ms or its preceding papers. Thus it is my view that the ms is based on a false premise and provides a biased view of the state of the technique. If spontaneous tracks did not behave in similar fashion to induced

tracks then surely it would not be possible to use the technique to accurately predict track response in geological conditions. In my view, the failure of this and other papers to cite evidence which oppose their own views represents a serious failure to follow established practices in scientific publication.

*We do not disagree that a range of studies have shown AFT data to be consistent with other techniques, to within their respective time and temperature resolutions, in those settings. This track record provides grounds for believing that the deleterious effects of some underlying assumptions being incorrect or inexact is limited. However, it is certainly not grounds for believing that all of our underlying assumptions are perfect. Sometimes we can be only approximately right, or right for the wrong reasons.*

*The reviewer misinterprets that we are trying to knock down the technique. We are not. We are making the case, however, that we don't know everything we thought we did, and it would be good to figure it out.*

The ms suffers from another serious shortcoming is that although not stated explicitly, it suggests that etching of confined tracks is the only consideration in determining appropriate etch times. In practice the etch time is also required to ensure that tracks which intersect the grain surface (i.e. those employed in determining a fission track age) are clearly revealed, to allow accurate counting without too many overlaps which might obscure some tracks. In addition, in routine application involving natural samples, particularly in the case of detrital grains in sandstones, grain mounts invariably contain apatites with a range of bulk etching rates, so that tracks in some grains will be etched to a greater degree than in others. Thus, the typical etch time of 20 seconds represents a balance between the need to sufficiently reveal tracks in grains with the lowest etch rates while not obscuring tracks in grains with higher etch rates. Accepting this view, and given the importance of measuring track lengths and ages in the same grains (although sadly this is often not done), it is difficult to see how the results of this study could find any practical application. Perhaps the authors should give some consideration as to how their work could be implemented in practical application.

*We did mention the competing consideration of density measurements at the end of section 6.6, but we agree that we can be more holistic in discussing what goes into selecting an etching technique, and will do this during revisions. We don't consider this to be a "serious shortcoming", however. First, it is worthwhile in and of itself to develop a quantitative understanding of how to maximize efficiency of confined track revelation. Second, we did give some consideration in our section 6.6, which we can further clarify and develop; for example doing a single, standard etch for density and preliminary length measurement can be followed by a short re-etch and revisiting the same grains in a second measurement step, which is made relatively easy by modern software.*

Another issue, which the authors appear to discount, is that if measurements (including annealing experiments and data in field samples) are standardized to the same etching

and measurement conditions, then the effects that they are discussing in this ms should not be a problem. This has certainly been our philosophy at Geotrack since 1982. In our early days, we performed several standardization exercises with personnel from other laboratories to ensure that similar measurement protocols were adopted. However in recent years the explosion of laboratories around the word has resulted in a considerable number that have not shared that consistent approach.

*We disagree. The weakness of relying on this approach is documented in this and previous studies. We see in inter-laboratory experiments that even identical etching conditions lead to divergent results in the community (Ketcham et al., 2015). Standardizing etching and measurement conditions does not resolve what we propose to be the decisive variable, which is the choices the analyst makes. This has long been suspected, or even informally understood, but not quantified. Before this study, the primary response has been for experienced workers to tut-tut the "explosion" of new and implicitly under-trained laboratories (and perhaps tut-tut similarly experienced workers who nevertheless get different answers). Fission-track length measurements should not be a morality play. We believe that a quantitative understanding of track etching can help in the development of a more productive response.*

Another motivation for the study described in the ms concerns the disappointing degree of reproducibility of track length measurements in inter-laboratory comparisons (lines 59-60). The authors may be better advised to focus on this aspect, rather than any perceived problems with the equivalence of spontaneous and induced tracks. A lack of standardization between laboratories is likely to be a major feature in causing the observed lack of reproducibility.

*We believe both problems deserve focus, and that they are not unrelated.*

Another important issue with one aspect of the analytical procedures adopted in the study concerns the measurement of under-etched tracks. The previous paper (Tamer and Ketcham, 2020b) discusses "a limited number of tracks that remained under-etched at 20 s but were measured anyway to adhere to the experiment design". This allows a certain degree of arbitrariness into the measurements which makes it difficult to know how to interpret differences from one etch step to the next. This issue is also referred to in the following list in relation to lines 74-77 and 194-196. The question arises of how the ends of under-etched tracks were measured, and what such measurements mean. The resolution of the track end points becomes a critical issue and where the width of a track is less than 1 μm we are talking about measuring objects with a size close to the wavelength of light. So do these measurements indicate the actual etched length or only some part of that governed by resolution? Thus there is a question of how much the scatter in the measurements is due to problems in resolving track tips and how much is due to real differences in etch rates. This is also an issue with measurement of Dper, all of which are around the wavelength of light or less. This would account for the degree of

scatter in the Dper measurements in Figure 5c of Tamer and Ketcham (2020b). For these reasons, some discussion of how the authors dealt with this problem is required.

*This is an excellent point, and we agree that this is a very important consideration that deserves more emphasis. However, we believe that the "etch-anneal-etch" experiments of Tamer and Ketcham (2020) resolve the question. In that experiment, following the initial 10-second etching step and annealing, there was another 10-second step, followed by a 5-second one. We assume that annealing will not affect an etched track. If there had been a significant extent of etched track not included in the length measurement after the initial 10s due to the track tip being too thin to resolve, we would expect that the etching rate would have been faster in the first post-annealing etching step than the second, as the invisible part of the etched track would have been revealed, in addition to bulk etching at the track tip. We did not observe this -- in all three experiments the track length increased at a near-constant rate after annealing, in both steps.*

Specific comments regarding the text:

Line 60: *the region along tracks beyond where most current etching protocols reac**h***

The meaning of this is unclear.

*We can clarify (it's because they start step etching after 20s).*

Lines 44-48: *Virtually all mathematical treatments of track revelation, biasing, and the relationship between confined track length and track density (… … ……… …..) presume that tracks are line segments in space, all etched to their full extents once they are intersected.*

This is inaccurate, at least in terms of those papers involving Galbraith, Laslett and our group. These mathematical treatments are based on the assumption that only tracks that are etched to their full extent are measured. This is not the same as the statement quoted from the ms.

*The reviewer is incorrect. All of the mathematics in those papers, and the papers they relied on (e.g. Parker and Cowan 1976), only encompass relative probability of intersection. None mention time in the etching process. They thus make the implicit assumption that probability of intersection is equal to probability of measurement. They omit the possibility that different tracks may be differently likely to be fully or sufficiently etched after intersection.*

*We also note that the reviewer has little quantitative basis for saying that tracks are etched to their full extent. If "full extent" means the entire region of enhanced etching rate, we document that the full extent of unannealed tracks in Durango apatite (years after irradiation) is about 17 μm.*

Lines 73-74: *Finally, to be measured a confined track must be etched sufficiently to be observed, the criteria for which will differ depending on the situation.*

What does the second part of that statement mean?

*This is clarified in the following sentences; we can clarify further ("to be observed" was admittedly clumsy; should have been "seen and judged suitable for measuring" or something like that).*

Lines 74-77: *For routine AFT analysis, where the analyst evaluates whether a track is sufficiently etched, the ends of the tracks need to be clearly visible, although this evaluation is analyst-specific. For measuring tracks in early steps of step-etching experiments, the criterion is simply that a track and its tips be visible enough to make a reasonable measurement.*

There is an underlying question here that pervades the ms as well as preceding papers; what criteria should be used in measuring tracks where the ends are not properly visible? This needs to be discussed – at least, the authors should discuss the approach that they used here.

*We can elaborate somewhat, but only to a limited degree. The principal criterion was to be confident that the end was visible, and that it was not a gradational fading into nothingness. As discussed previously in this response, follow-up measurements indicated that this resulted in no "missing length" beyond the track tip.*

Lines 87-88: *There was no clear indication of vB varying with track orientation (Tamer and Ketcham, 2020b)*

I find this puzzling. Surely the shape of the track openings in a prismatic surface shows that the bulk etch rate is higher in the direction of the c-axis than perpendicular to it. In this context I am puzzled as to why the authors have adopted the value of 0.022 µm/s for the bulk etch rate, when other samples from the same experiment define different growth rates, and this value is intermediate between the growth rate of Dpar and Dper.

*We found it puzzling as well, but the simple fact was that there was no clear signal in our measurements, even though we were expecting one. The signal was just not large enough to emerge from the noise; the fact that two of our experiments featured relatively few low-angle tracks. We agree with the reviewer that there probably really is a difference, but rather than assert some larger truth we decided to just stick with what our data said at this time.*

Lines 92-93: *In cases where it was difficult to determine if an intersection truly occurred due to interfering features, we conservatively included it.*

I would say that a conservative approach (meaning careful, in my view) would be to exclude such values.

*The reviewer misinterprets what we mean by "conservative." The principal mechanism by which our etch rate determinations can be wrong is if multiple intersections lead to an artificial, apparent acceleration of etching due to different ends of the track etching simultaneously. We thus wanted to challenge the veracity of our etch rates as strongly as possible by including ambiguous cases.*

Line 94: *Section 4 heading. "4. The model"*

What model? Up to this point there has been no description of what is to be modelled or why. This needs to be explained and the basis of the model should be described. The reader should not have to work this out for him/herself.

*This critique is incorrect. We introduce "the model" on lines 50-54.*

Line 121: *Figure 2 illustrates the implied growth curves*

What is meant by "growth curves"? An informed reader can probably work this out but the content of the Figure should be explained more clearly. And in C,D, why are curves not shown for t=20 sec? What values of VT, VB have been used in constructing this figure? And in regard to Lines 129-130, It seems unlikely that differing intersection times would contribute significantly to differences between observers if other requirements of measurement (in terms of tip shape) are met, on the basis of this Figure.

*We can clarify by calling them "lengthening curves." Part of the reviewer's question is answered in the very next part of the sentence he quotes ("… using Constant-core model rates for unannealed tracks calculated later in this paper (Table 2)"). We can also include similar text in the caption, if that helps. We did not include 20 seconds because, simply, no confined track is etched for 20 seconds in a 20-second etch, because first a semi-track has to etch down to reach it. (We feel that this work requires a certain amount of unlearning by even experienced fission trackers; shortcut like "confined tracks etch for 20s in a 20s etch" are wrong). Concerning lines 129-130, we are not talking about variation among observers, but simply that tracks of the same true length will have a range of etched lengths after a given amount of etching, based solely on where they are impinged. We are putting it out there that this simple, inevitable mechanism explain part of the scatter observed in induced lengths.*

Lines 134-135: *First, the relative probability of a track of latent length or half-length L crossing the surface*

Is it latent length or half length? The two terms suggest different meanings.

*We agree that our phrasing is confusing, and can fix it. We were trying to convey that the statement is mathematically correct for both lengths and half-lengths. Half-length is used for the fundamental fission-track age equation (Fleisher, Price and Walker, 1975) because only one of the two fission particles can cross a given plane, and some readers may be used to thinking of things from that perspective.*

Line 138: *The semi-track penetration calculation…… ..*

Line 150: *Figure 3 shows examples of penetration and revelation rate.*

Both of these extracts from the text represent one of the most confusing aspects of the paper, at least as far as I am concerned. I cannot work out what Figure 3 is supposed to show, and the text provides no information that will help to work it out. Simply "penetration and revelation rate". These terms should be explained in detail.

*We can do that. Penetration refers to the etching of semi-tracks, revelation refers to intersection and etching of internal tracks by the etching semi-tracks.*

Line 165: *Figure 4 shows an example model of 107 track intersections*

This Figure doesn't show a model. It shows results from a model. Maybe I'm old fashioned but I think text should be clearly written, describing what is shown accurately and clearly.

*We will endeavor to improve.*

Lines 194-196: *For step etching experiments with first steps shorter than 20 s, precise location of the tip is not a prerequisite for track selection, which is instead a matter of simple visibility. In the earliest stages of etching, tracks will be too thin to be observable in visible light. As they grow, they become more efficient at reflecting light, making them more detectable. However, when this occurs in the context of practical fission track analysis is unclear.*

This is not clear to me. To measure a track it is necessary to identify the end, so how did the authors decide where the ends were?

*We addressed this in our initial comments; we will clarify.*

Line 197: *Lacking a physical basis for determining when etched tracks begin to become visible*

This follows on from above. I find this section very confusing. Constructing an empirical bias relationship to allow for this aspect seems a very imprecise solution. Why should the same bias function apply in all cases?

*We agree that this is an imprecise solution, but we are trying to quantify something nobody has quantified before: predict which barely-etched tracks an analyst will see and select and which ones will either be missed or rejected. It's a difficult problem, and we certainly welcome improvements. The same bias function should apply in all cases here because it is the same analyst using the same instrumentation, attempting to do the same thing each time.*

Lines 210-211: *Figure 7 shows the distribution of track lengths and etching times with depth below the surface*

This Figure does not show the distribution of lengths and etch times. It shows model predictions in terms of contours. Once again, precise description is important. Plus more description

*We can insert "predicted".*

Lines 218-219: *Fitting the experimental data consists of posing an etching structure and using it to construct first a semi-track distribution and then a distribution of etched lengths*

What is being fitted to what here? This should be explained clearly.

*Mean track lengths for each etching step, as explained in the sentence start in line 220.*

Lines 221-222: *We also include the option to incorporate some innate variation in latent track length. For the modeling in this paper we chose $\sigma$ of 0.5 μm, based on the scatter of lengths after c-axis projection to remove anisotropy effects*

Wouldn't a value of 0.8 or so be more appropriate, based on the distribution of induced track lengths which show no anisotropy?

*No, because some variation in etched length arises from variation in the impingement point (Fig. 2C, D), not variation in latent length. We will clarify that we removed an estimate of this component of error in selecting this value.*

It would also be interesting to run models with a single latent length, to investigate how much of the observed distribution of measured track lengths can be attributed simply to etching processes.

*Our initial models were like this, but we did not want to go into this particular detail in the interest of keeping the paper shorter. We report on line 224 that the effect is minor.*

Lines 229-232: *The step etch experiments only consist of 3-5 steps, making it difficult to meaningfully constrain models with 2-3 variables defining etching structure, in addition to biasing factors. To increase the amount of data, we paired data sets based on the same or equivalent tracks. Thus, we simultaneously fit data sets SE1 and SE2, both of which analyzed unannealed induced tracks, but with different initial etching steps.*

This is another of the most ambiguous passages in the ms. In what sense were the data sets paired? What does that mean? Why try to fit models with 2-3 variables to data with only 3 to 5 steps? Why not try simpler models or use more etch steps to resolve these problems? And why combine data with different initial etch times? More control on experimental design would ameliorate these issues.

*We are working with the data we have, and will certainly do more in the future. Pairing is simply using two sets of etching steps on the same material (i.e. at the same level of annealing, often re-polishing the very same grains); we will clarify that. It's tough to get simpler than a model of 2 variables. Combining data with different initial etching times is actually a feature, not a bug, as it tests whether the model can explain a wider range of data. For example, as discussed in the paper, one might expect to get a different track length distribution if one selects tracks after 10 seconds and then measures them again after 10 more seconds of etching, versus if one just etches for 20 seconds and uses one's standard-operating-procedure selection criteria.*

Lines 241-245: *During fitting of the unannealed induced track data (SE1 and SE2), it became apparent that one pair of data points were exerting outsized control on the result. Since the 15-20 s and 20- 25 s steps for experiment SE2 feature the very similar mean length reduction and thus almost the same rate, in the context of our model forms assuming a linear rate decrease fits were forced to split this difference to the exclusion of closely fitting the rest of the data for SE1 and SE2. We thus excluded the 15-s mean length for SE2, effectively making the second etching step go from 10 s to 20 s*

I wonder if this presages graver issues within the data set. Why should this dataset be problematical? To simply reject these data seems rather arbitrary.

*One outlier in 28 experiments doesn't seem all that bad. We said we did what we did, and why we did it, so the reader can decide what to think about it.*

Line 268: *Figure 9 shows the parameter fits for Constant-core models, and Figure 10 for Linear models*

Here is the biggest single problem that I had with reviewing the ms. What on earth does this mean? This Figure shows a series of cross plots, without any description of what is being plotted or why, and what they mean. What does each dot represent in each plot? I find it totally unacceptable that the authors leave it to the reader to work this out.

*We will expand the figure caption; the information is nearby, in lines 249-252.*

Line 287: *It is remarkable that we are able to reproduce the mean length data in each etching step of all experiments simultaneously using these simple models of etching structure*

Give the number of adjustable parameters, each of which is free to vary as required until a fit is obtained, I don't see this as remarkable at all. And regarding the statement that mean length is reproduced, I guess that Figure 8 is supposed to show this, but a plot of measured vs fitted parameters would seem to be a more graphic depiction of this.

*There are only 2 or 3 adjustable parameters, which need to fit 4-7 data points in the 5 pairs of step-etch experiments. The reviewer should try it before he dismisses it. A residual plot is the clearest graphical way of comparing predictions to data and uncertainties. We could also do a typical 1:1 plot, but that would be less informative, and redundant.*

Line 329: *Fossil and unannealed induced tracks have slow core etching rates, while all annealed induced tracks have far higher rates.*

It is counter-intuitive that annealing would result in an increase in etch rates within the residual track region, particularly when spontaneous tracks do not show enhanced track rates. Such behaviour would suggest that heating increases the degree of damage, when all previous experience with lattice damage suggests that heating leads to a decrease in damage intensity. This suggests that there is a flaw in either the experimental design or

the model design from which this conclusion was derived. The authors seem to accept this observation at face value, but this unusual result deserves further investigation before any firm conclusions can be drawn.

*We found it counter-intuitive as well, and we devote lines 299-311 to our most likely suspect of a culprit, which is multiple impingements spuriously increasing etch rates. We have gathered more data on this, and are confident in our interpretation. We disagree that etching rate necessarily maps directly to a "degree of damage," however. Damage is complex, and many types of transformation are possible.*

The fitted values of VTmax and ΔxTmax in Table 2 show no consistent trends that might suggest these results have any meaning. It would be reasonable to expect these values to vary in a consistent manner as the degree of annealing progresses, but for annealing temperatures of 235, 270 and 280°C, VTmax varies from 4.0 to 1.77 and back to 3.36, while ΔxTmax changes from 0.19 to 11.35 and back to 0.76. It is hard to imagine why the distribution of damage should go through such contortions within a limited range of annealing temperatures. These results suggest that the fitted values are simply empirical numbers without any real meaning. For the Linear model, values of VTmax change from 4.09 to 3.54 to 3.59 over the same range, compared to the value for induced tracks of 1.70, and again it is difficult to see how these numbers can be accepted as having any physical reality. This I turn raises questions regarding the significance of other aspects of the results.

*We're just showing the results – those are the fits to the data. We can add more text to justify our two model forms as reasonable attempts at simple functions. We agree that the fluctuations in the core size of the constant-core model are a likely indicator of shortcomings in the Constant-core model. We are less alarmed by the Linear model patterns; some variation can simply reflect that we represented a complex, curved change in etching rates as a line.*

Incidentally, it would be useful to show error bars in Figure 11, as per normal practice.

*We shall do so.*

Lines 331-332: *This may be responsible for some component of the mismatch in annealing fossil versus induced tracks in laboratory experiments (Wauschkuhn et al., 2015a),*

As mentioned earlier, there is plenty of published evidence that there is no mis-match. The authors appear to have adopted this observation with no critical assessment. At the very least, the authors should mention that evidence to the contrary exists, and preferably also to explain why they discount that evidence.

*We discussed this previously in this response.*

Line 337: *treatments of fission-track lengths are all based on a line segment model*

Any mention of the line segment model should refer to Rex Galbraith's (2005) book.

*OK, we can include this one as well.*

Lines 457-457: *The vT(x) model framework provides a quantitative basis for evaluating whether the etching procedures used today are the most effective at their goal, which is to provide high numbers of reproducible and informative measurements to constrain thermal histories.*

I would dispute that the goal is simply to provide high numbers of measurements. Surely the goal should be to obtain data that provides the most accurate thermal history constraints possible. This depends not only on track length measurements but also on fission track age determinations and also on allowing for different etching rates in apatite grains on a grain mount. The vT(x) model described in the ms focusses on only one of these aspects.

*We discussed this previously in this response.*

References not cited in the ms under review:

Duddy, I.R., Green, P.F., Laslett, G.M., 1988. Thermal annealing of fission tracks in apatite 3. Variable temperature behaviour. Chem. Geol. 73, 25–38.

Green, P.F. and Duddy, I.R. 2018. Apatite (U-Th-Sm)/He thermochronology on the wrong side of the tracks Chemical Geology 488, 21–33

Green, P.F., J Duddy, I.R. & Japsen, P., 2018. Multiple episodes of regional exhumation and inversion identified in the UK Southern North Sea based on integration of palaeothermal and palaeoburial indicators. In:

Bowman, M. & Levell, B. (eds) 2018. Petroleum Geology of NW Europe: 50 Years of Learning – Proceedings of the 8th Petroleum Geology Conference, 47–65.

Spiegel , C, Kohn, B.P., Raza, A., Raiuner, T., Gleadow, A.J.W. 2007. The effect of low temperature exposure on apatite fission track stability: a natural annealing experiment in the deep ocean. Geochimica et Cosmochimica Acta 71:4512- 4537.

---

## Author Comment (AC2) · 13 Dec 2020

**Answers to comments on manuscript Confined fission track revelation in apatite: how it works and why it matters**

Richard Ketcham and Murat Tamer

Our answers are below in *red italics*.

**Answers to comments by Raymond Jonckheere and Bastian Wauschkuhn**

We should declare that we have a competing model, and thus a possible prejudice against, or in favour, of the reviewed manuscript. It was our intention not to review it, but we accepted at the insistence of the authors, and because of its interest to those in our group involved in related studies. Our evaluation is critical but we believe not unfair. The public comments will set us right, if need be.

General comments

This manuscript reports calculations and numerical simulations of fission-track etching in apatite based on a variable track etch rate vT(x), isotropic apatite bulk etch rate vB, and certain track selection criteria. It concludes with a discussion of the implications for apatite fission-track da-ting and modelling.

An astounding fact considering the first author's impressive publication record is that the manuscript is almost unreadable. Even with foreknowledge of the concepts involved and of earlier publications, we struggled to understand certain sections. This was not helped by the constant use of ill-defined or undefined notions and meaningless word strings (e.g., semi-track penetration calculation). There appears to be no attempt to be understood, let alone to be exact, and readers less familiar with the subject can make heads nor tails of the manuscript, much less evaluate its significance.

*We apologize for the difficulty caused; perhaps we can attribute it partly to Covid-19-related stresses. In any event, we acknowledge, as in our response to the other reviewer, that some things could be explained more clearly, and in a paper like this a small hang-up can greatly affect the reader's ability to see how the steps lead to the larger picture.*

A second striking fact is that the manuscript presents a track etch model that appears to exist in a vacuum, disconnected from all existing knowledge of latent tracks, etching of tracks and minerals, and from all published step-etch data, except for those of Tamer and Ketcham (2020), e.g., for the 5.5 M etch (Carlson et al., 1999; Jonckheere et al., 2007, 2017; Tamer et al., 2019), the comparable 5.0 M etch (Laslett et al., 1984; Green et al., 1986; Barbarand et al., 2003; Moreira et al., 2010), and, given that the model is a first-order approximation, for lower etchant concentrations as well.

*We're not entirely sure what the reviewers are asking for on this point – a review of all existing knowledge on latent tracks and etching? The reviewers themselves have produced thorough reviews, in their papers, and we refer to those. With the exception of Jonckheere et al. (2017), all prior step-etch studies have measured a fresh, random sample of tracks after each etching step, and we created our model to specifically interrogate data in which the same set of tracks is measured in each step, as such data provide a much clearer signal. It probably would not be too hard to extend our approach to earlier etching studies, but would require additional assumptions, most notably the need to estimate, and perhaps evolve, the analyst's decision-making process at every etch step, rather than only the first, which both of our reviews agree is fraught enough as it is.*

One respect in which we found the manuscript lacking is a post-factum assessment of what it all signifies. The known statistical properties of surface tracks and confined tracks are a basic ingredient of the model. A second ingredient is the principle that to etch a track-in-track, the etchant penetrates down a host track, crosses to the confined track and etches it from their intersection point outwards. It seems not unlikely that this explains the broad traits of Figures 2-4, and, with scaling based on measured lengths, some experimental results as well, without the need for a specific vT(x) model. A foreseeable consequence is an excess of "under-etched" tracks which is culled down to ≪10% based on for the most part unexplained ad hoc selection criteria. This results in etch rates (vTmax) ranging by a factor of ~3 and core lengths (ΔxTmax) by a factor of >50 between samples. Dazed by the mathematical acrobatics, the reader is left wondering what it all means. Is this a hard numerical result, and which of the several factors above weighed most on the outcome, the fundamental geometrical configuration, the etch rate model, or the selection criteria?

*We can add such a summary.*

From our standpoint, the proposed model is too approximate and dependent on questionable assumptions to be confident about the numerical results. On the other hand, it highlights the paramount importance of etching protocols and selection criteria for apatite fission-track dating and modelling.

*Again, we're not sure what the reviewers are getting at, or asking for. A sensitivity analysis of all model inputs?*

Specific comments

4.1. Etching structure. Given the title: "*Confined fission track revelation …*", and Figure 1, showing a confined track with two endpoints, one could be forgiven for thinking that the calculations refer to etching of confined tracks. It turns out, after a full page of baffling equations suspended in mid-air, that the calculation was not that of the lengths of confined tracks at all but of those of a semi-tracks. The confined track length is then

obtained as the sum of complementary semi-tracks. One needs to inform the reader from the start about what is going on, i.e. how one is going about solving a problem. With that information and a full explanation of all the symbols (and simpler notation) one need not bother about (1)-(5) as the conclusions are rather obvious from the outset.

*We agree that we can improve how we introduce our approach to the problem. We do explain all of our symbols; we don't know which explanations the reviewers are finding insufficient. We're not sure what line of reasoning indicates that we "need not bother" with equations 1-5; are the reviewers suggesting that they should just be entirely relegated to the Appendix? Similarly, we have no idea what the reviewers mean by "the conclusions are rather obvious from the outset."*

Figure 2 suggests a more interesting observation that is not discussed: the results for the "Constant-core model" (A, C) and the "Linear model" (B, D) are identical. It is possible that this is due to a conservation principle, in the sense that if A and B are far enough separated, then the time to etch from A to B is independent of the etch rate function vT(x) between A and B, but only depends on the area under vT(x). A consequence of this observation is that if the two end-member models predict the same evolution of (semi-)track length with etch time (cf. Figure 2), then how are step-etch mean-length data going to distinguish between them, or between them and other models?

*The two models are similar, but not identical. It's actually the area under 1/vT(x) that must be the same, and if there is only one etching step then the models are indeed indistinguishable. However, if there is a second etching step (leading to points A+dA and B+dB), and then a third, etc. then the structure imposed by the two models constrains how these additional increments can be accommodated. The same goes for a different track, etched to two different points (C and D), perhaps over a different time, etc.*

4.2. Semi-track penetration and confined track revelation. Much as the preceding discussion of the track etch rate and apatite etching, the lead up to Figure 3 (line 150) reads as a perfunctory dismissal of the accomplishments of earlier scientist as too trivial to cite. This entire section could have read: "*Semi-tracks have a sin(2δ) dip angle distribution and a homogeneous etchable length distribution (Dakowski, 1978; Laslett et al., 1982)*". Then some modelling happens and something is plotted in Figure 3: relative semi-track penetration and relative confined track revelation against depth (below the initial or etched surface?). Even with foreknowledge it is not possible to guess at the significance of Figure 3. What is concluded from Figure 3 does not seem to require modelling at all: "*~10% of tracks reach ≲1 μm*" and "*252Cf tracks are shorter, … but efficient*". Yes, see Dakowski (1978): ¾ of semi-tracks reach ≤½ a full track length deep, and most have near-zero depths.

*Yes, we could have made this section briefer and just referred to the equations, but we wanted to take it a little bit slower for the sake of readers not as familiar as the reviewers with the mathematics of track revelation. Figure 3 is developing the first stage of the overall model; the curves are*

*dependent not only on the known mathematics (and we can cite Dakowski here), but also the vT(x) structure. We are not really trying to conclude anything from these figures. However, they do provide, in our opinion, an interesting visualization of semi-track penetration, and confined track revelation, through time, which helps develop an intuition for not only where tracks reach, but when they get there. These figures are also helpful in understanding the subsequent contour diagrams (Fig 4, 7).*

4.3 Confined track intersection. Line 158: "*weighted as cos δ*": see Dakowski (1978). It is hard to comment; on the one hand the concept is simple (hence likely the shorthand reporting) but, on the other, it is impossible to puzzle out what exactly has been done or what exactly is shown in Figure 4 (CDF?). Its discussion again refers to issues that seem to require no modelling at all: "*only about half of the tracks remain after surface intersecting track are excluded*". Yes, a slab of thickness L below a unit of surface contains L N tracks of which ½ L N intersect it (Fleischer et al., 1975).

*The reviewers are incorrect in their translation here, as the model shown here only includes tracks within the angular interval the analyst might measure; and we are also including tracks with a range of depths, but very much weighted toward tracks near the polished surface, because that's where intersections are concentrated as semi-track etching progresses (see Figure 3).*

*We also note that in these critiques the reviewers are making the same omission in earlier treatments that we want this contribution to begin to correct: the element of time, and the fact that all aspects of the track intersection and etching process evolve in time, not just track length and shape. This has been known for a long time, of course (e.g., Jonckheere et al. 2007) but this paper is the first we're aware of that puts the entire process together. The straightforward reading of Figure 4 is: here is where and when track intersections occur.*

4.4 Confined track selection. Fleischer et al. (1969), Paretzke et al. (1973) and a great number of other scientists published calculations of etch-time dependent track profiles for varying vT(x) and isotropic vB.

*We mention Fleischer et al. (1969) in Section 2, and mistakenly omitted re-citing that work in section 3. We consider the model in section 5 rather primitive, and exclusively done for illustration sake.*

Line 185: the isotropic-vB case is not "simplified" compared to Aslanian et al. (in press), but contradicts it, as well as fundamental etching theories (e.g., Heimann, 1975). The definition of "bulk etch rate" is different.

*The reviewers are argumentative here; one person's simplification is another's heresy. We are already saying that their work will provide a better answer, but because they are still using undescribed "graphical construction methods" (Aslanian et al., 2020) rather than providing a time-evolving calculation, we could not reproduce their work for this illustration.*

Line 198: "*we constructed … an operator bias function*". Despite the reserve one has to ask: where does it end? It brings to mind Murphy's variable constant, which multiplied with anything gives the desired result. How decisively does Figure 6A differ from a straight ramp, or even a step function? If there is an actual basis for it, please discuss it; if not, all that follows from 6A is a possible artefact.

*As also expressed in our other response, here we are faced with a problem of quantifying something nobody has seen the need to try to quantify before: which barely-etched tracks will an analyst see and select, and which ones will either be missed or rejected. Unfortunately, if one wants to reproduce a measured track length distribution based on latent track etching structure, there is no getting around somehow accounting for which tracks are measured and which ones are not.*

*Furthermore, we believe we make a fairly convincing case that lack of a quantitative treatment of the issue of which tracks are measured and which are not underlies the AFT community's difficulties with reproducibility of length measurements. This may also get to the crux of the mental barrier that both reviews appear to reflect – they substitute virtue (implicitly: "good analysts only measure fully etched tracks") for empiricism ("what, exactly, is a fully etched track?").*

*As outlined in the text, we took the first-step measured data, and constructing a function that reproduced the left, or short, side of the track length distribution, and we furthermore used the same function for all such measurements in this study to limit the degree of license. The vT(x) model is responsible for the right, long side of the distribution, and so the results have an anchor in reality – too long, or too short, and we cannot match the data. Furthermore, with subsequent etching steps, both the left side and the right side of the distribution has to be generally reproduced to match the mean measured length; another anchor.*

*We thus reject the reviewers' exaggerated dismissal.*

Figure 5 is to all intents and purposes identical with Figure 4 of Fleischer et al. (1969); the equations are also not dissimilar. It is worth noting that the assumed gradual increase of track length is contradicted by the step-etch data of Jonckheere et al. (2017, in part measured by the present co-author), demonstrating that confined track lengths increase in fits and starts, reflecting a discontinuous latent-track structure, as reported in numerous apatite studies (Paul and Fitzgerald, 1992; Paul, 1993; Li et al., 2010; 2011; 2012; 2014, etc.), as well as nigh the entire literature on latent ion tracks (not counting the amusing contributions from the Canberra group who posit a cylindrical track and then shoot their ions straight through the crystal so that the track ends are missing). One can of course think of vT(x) as averaging over a certain section, but this should be stated explicitly.

*We can state this explicitly.*

Figure 6. It is worth commenting on the true significance of the observed "excellent agreement". It would appear that the invasive, ad hoc filters (6A and 6D) would transform just about any underlying distributions (light blue in 6B and 6E) into almost negligible residual fractions (dark blue in 6B and 6E) with the general Gaussian appearance of the measurements presented for comparison.

*As expressed above, and in the text, this is not the case. The "ad hoc" filters are only responsible for the left side of the distributions; the right side is an outcome of the vT(x) model.*

Figure 7 is interesting, but, of course, underlies the combined reservations concerning the steps leading up to it.

*One can also look at Figure 7 as a set of potentially testable predictions.*

A puzzling fact about section 4 is that 4.1 sets up two competing ad hoc models for the variation of the etch rate vT(x) along the tracks (in fact one end-member and an generalized member of a single model, which is assumed to be the same for tracks produced by all fission fragments and energies). Sections 4.2 to 4.4 then present modelling results, which seem not to require the equations in 4.1, and never again mention which model (or parameter fit) was used, much less how the models differ.

*This was indeed an unfortunate omission, which we will address. Frankly, in these figures, the differences between model forms is not really recognizable (as was largely the case in Figure 2). The principal difference is in their ability to reproduce the measured length data. The equations in section 4.1 are required to etch both semi-tracks and confined tracks as a function of time.*

4.5 Fitting step-etching data. This section presents a complicated account of different modelling issues. It would help to reproduce Figure 1 of Tamer and Ketcham (2020) to present the reader with visual map of the data. It is worrying that there appears to be a need to exclude one data point, that the iterative fitting procedure does not seem to converge towards unique best-fit solutions, and that such solutions appear to not be strictly reproducible. Please discuss what that actually signifies.

*We can add the figure, and write a short paragraph summarizing how sections 4.1-4.4 go into a model realization. It "signifies" that we were not able to come up with a set of mathematical expressions that deterministically generate a track length distribution as a function of etching structure and time, and thus used a Monte Carlo approach.*

5. Results. One cannot help thinking that the discussion of Figures 8-11 could be so much better structured if, instead of the artificial opposition of "two models", one of which is an end member of the other, there would have been just one constant-core model with two end members, a linear-vT(x) model ($\Delta$xTmax = 0) and a constant-vT(x) model ($\Delta$xTmax = Llat). The latter is at least relevant to the ceremonial dismissal of the model unfairly attributed to Laslett et al. (1984) in Tamer et al. (2019).

*We recognize the reviewers' desire for more order, but we disagree with the prescription – we are responding to what the data say about these two possibilities. Why would we want to force an end-member that doesn't fit most of the data?*

*The reviewers should re-read part 8 of Laslett et al. (1984). It refers specifically to a single vT, and says the track ends are "first resolved" when the "full etchable range" is reached, and that "in practice most confined tracks are slightly over-etched." These statements are all, strictly speaking, incorrect. We believe that this helped develop a misplaced confidence and complacency concerning our ability to identify a "fully etched" track, which in turn has led to poor reproducibility.*

6. Discussion. "*It is remarkable that we are able to reproduce … using these simple models of etching structure*". Yes indeed, on condition (1) that the predictions in the first place depend on the assumed model, and in negligible measure on the geometrical framework and assumptions concerning track selection, (2) there is a negligible self-referral between the data used for calibration of the models and those adduced for testing them. Perhaps the careful term "reproduce" reflects concern about the fact that these overlap, which, in our view, affects the significance of the results.

*We made the statement in (a) some amazement that everything worked, and (b) recognition that there is much that can be improved. One has to start somewhere.*

"*We take these successes as an indication of the overall correctness of our characterization of confined track etching*". This is too facile; it could be a Ptolemaic model, which, admittedly based on false assumptions, nevertheless manages to account for isolated appearances (individual experiments) given suitable eccentricities and epicycles. The authors need to make a serious effort to make transparent the network of mutual influences between the geometrical framework, the actual etch model, the various selection criteria, calibration data, and the data used for verification, so as to to reassure the reader that their model is not in effect a self-sustaining mirage hovering in mid-air.

*We were transparent in the one "self-referral", which was to estimate the left/short side of the length distribution for the first etch step based on observations, and we minimized the influence of this by using the same rules across all experiments. We will clarify this. Concerning Ptolemaicism, every other aspect of our model is based on accepted physical and mathematical principles of fission track behavior.*

It could be argued in favour of the model that, following annealing and etching together, pre-annealed induced tracks are longer than fossil tracks (Wauschkuhn et al., 2015; Figure 15). Perhaps they etch faster?

*Certainly, etching faster should cause some difference, but we don't yet have the information to estimate how much of one – in other words, any such estimate would be highly artificial. First, we do not know the vT/vB of the analyst (or, if vT/vB is the optimal way for parameterizing analyst*

*track selection; it's more a proposal at this point). Second, we have no data yet on how laboratory annealing affects the etching velocity in spontaneous tracks.*

Line 358: "*A thorough re-evaluation of this biasing based on measurements…*". This appears to overlook the fact that the (length) bias function implemented in modelling programs is based on experimental data.

*Experimental data of laboratory-annealed induced tracks; this is why we have to study fossil tracks.*

Lines 370-380. "*As vT/vB rises the proportion of accepted tracks increases …*". Do we interpret this accurately as: "Including shorter tracks decreases the mean length and increases the standard deviation (range)"?

*Somewhat accurately – one of the points of Figure 15A to provide some more specificity on the estimated trade-off.*

Line 386: "*… the mean latent track length of 17 μm indicated by our data and model*". More reliable estimates bracket the average latent track length in apatite between ~18 and ~21 μm (Jonckheere, 2003a; 2003b; Jonckheere et al., 2017). A discussion of established concepts such as etching threshold and range deficit relevant to this issue. On a related point: one could cite Figure 1 of Jonckheere (2003a) in support of the linear model, on the assumption that vT(x) is proportional to dE/dx.

*We generated a new model using the current version of SRIM, which we can add as a figure (while citing previous related work); the profile actually looks a bit more like a cross between linear and constant-core. We also note that the "more reliable estimates" only characterize the track in the moment of formation, and ambient-temperature annealing may affect it to some degree. We can add such discussion.*

Line 406: "*tip appearance depends on … mount preparation, polishing, and cleaning technique, microscope optics, and captured image quality*". This should include etching protocol and apatite composition.

*We will add these.*

Lines 450-455: "*These differences in latent versus measured lengths …*". The discussion omits established facts such as the etching threshold and range deficit (Fleischer et al., 1975; Iwano and Danhara, 1998).

*We can add such text, but elsewhere; we were taking the shortcut of considering the "latent" track as the zone of enhanced etchability, a shortcut that has also been used by our forebears (e.g., Laslett et al., 1984). But we can be more precise, or specify the sense in which we are using it.*

Lines 450-455: "*By stopping etching as soon as the curve of length versus time in step-etch experiments was reached …*". No, … as soon as the data showed a constant rate of increase

of the mean track length, taken to be the average bulk etch rate. The same principle used for calculating your bulk etch rate.

*An incomplete sentence on our part; we'll fix it.*

Lines 460-462: "*… a rate defined by vB, which may be obtainable from etch figure measurements parallel and perpendicular to the c axis (Tamer and Ketcham, 2020a)*". Long-established etching theories hold that measurements of surface features are not suitable for estimating etch rates (Jonckheere et al., 2019).

*We are utilizing an empirical approach to address a practical problem.*

Lines 497-498: "*Figure 17 shows the near-surface portion of the penetration model …*": Figures 3 and 17 remain obscure; they could be etch-time-dependent depth distributions, but a better ex-planation is needed.

Lines 524-525: "*If … geological annealing results in different etching rates than laboratory annealing …*". It could be argued that, after simultaneous annealing, pre-annealed induced tracks are longer than fossil tracks (Wauschkuhn et al., 2015; Figure 15), perhaps because they do etch faster?

*Again, we do not yet have data on how high laboratory annealing temperatures affect etching in fossil tracks. But, this is a good reason to get such data.*

Lines 530-549: The outlook section reads as an appeal for more and better data and community involvement, but is rather short on scientific detail and tends to drift off into a dreamy utopia. It can be deleted. It is at this stage more important to look back and re-evaluate the results, than to look forward.

*We disagree, completely. Getting more data on more types of tracks and more types of apatite, is not dreamy – it's work that we feel is well justified.*

Line 551: "*a range of detailed step-etching data …*". This is a rather exaggerated designation of a handful of mean track lengths (Table 1) and some intersection depths. The imbalance between the "comprehensive" model and the limited data for calibration and testing is a conspicuous weakness of this work.

*We note here that the reviewers criticize our amount of data in the next breath after they criticize our recommendation to get more and better data. We also note that the measurements we present constitute a lot of work.*

Line 554: "*Along-track etching velocity … varies … among fission tracks …*". The intended meaning is probably among different types of tracks (spontaneous, induced, annealed), not between individual tracks.

*Actually, all individual tracks, especially fossil ones which have each undergone different amounts of annealing.*

Line 563: "*Most variation …*". Barring sample preparation accidents, does one need an etch model to conclude that lower mean lengths and higher standard deviations result from including shorter tracks? The vT/vB criterion is not progress, because it applies to isotropic detectors, not to apatite.

*Having something where previously there was nothing is progress. We leave the door wide open for something better, but it's also possible that a simple solution will be the best.*

Lines 575-613: "*Appendix: vT(x) Etching Model Equations*". This appendix contains a long list of repetitive piecewise integrals of simple functions, that are not used to obtain the results in this manuscript, instead produced by numerical simulations. The appendix thus seems superfluous to the manuscript, and may be either deleted or included as a supplement. We have not checked these equations. It is on reflection possible that the modelling uses the equations but this is no-where made clear.

*This is a truly bizarre comment. Of course they are used to obtain the results of this manuscript; they are how the numerical simulations calculate etching of each semi-track and confined track as a function of time. They define the variables we use to fit the data, and plot in Figures 9-11.*

*Is this nowhere made clear? Line 138-139 states, for example, "The semi-track penetration calculation thus consists of randomizing some number (typically $10^5$) lengths, dips, and $x_{int}$ points, and then using the etching model to trace each semi-track's penetration into the grain surface." We can make it yet more explicit ("… etching model defined by Equations 1 and 2 and a given $v_{Tmax}$, $L_{lat}$, and $\Delta x_{Tmax}$…"), but what other etching model did the reviewers think we were using?*

Matters of detail relating to the writing, figures and captions are too numerous to be addressed at this stage.

---

## Referee Comment (RC3) · Raymond Jonkheere (Referee) · 18 Dec 2020

**TECHNISCHE UNIVERSITÄT BERGAKADEMIE FREIBERG**

[Figure]

Final comments on the authors' replies to our review of gchron-2020-31

**Confined fission track revelation in apatite: how it works and why it matters**

Richard A. Ketcham and Murat T. Tamer

Our essential comments are contained in our review (RC2), to which the authors have replied (AC2). The manuscript discussion forum invites us, or at least allows us, to react to these replies. In the absence of comments from other scientist, this could result in an endless back and forth between authors and reviewers until a consensus is reached or the deadline expires and the editors must arbitrate. It appears to us that this could be an exhausting and unproductive process, as it is most improbable that agreement will ever be reached on what is, or is thought to be, right or wrong. Since our scientific differences are not the real issue, we *recommend that the manuscript be considered for publication*, with such corrections as the authors and editors think necessary.

**Manuscript**

Nevertheless, we one last time wish to draw attention to some points, unrelated to scientific content. Various issues contribute to the fact that the manuscript makes an unfavourable impression on an attentive scientist. There is an apparent disdain for the reader, who is expected to guess at the intended meanings of word strings that appear to have been made up on the run, with indifference. This could be avoided by rewriting the manuscript with the aim to be understood.

It is inevitable that readers will evaluate the approach followed in this manuscript as a quick fix. The claim that the goal is to determine track etch rates is unconvincing. Price et al. (1973) and Green et al. (1978) performed *actual detailed measurements of $v_T$* along ion tracks in minerals more difficult to investigate than apatite. Furthermore, suspicious emphasis is placed on incidental geometrical observations. The road to $v_T$ is paved with precarious assumptions and precipitous leaps. The most unlikely and limited data (the mean lengths of etched confined tracks accessed via surface-intersecting host tracks and an intervening apatite segment) are used for estimating the model parameters. The resulting $v_T$ profiles have at best the most tenuous of connections with real fission-track or apatite properties. It cannot be otherwise, considering how they are arrived at. The computer-generated $v_T$-models are used to calculate etched-track geometries, of which a subset is selected for calculating the mean track lengths. Instead of that, the calculated geometries could have been compared with countless images of step-etched tracks, eliminating the need for inventing selection criteria. Within the current concept, it would in fact be most efficient to start from observed track geometries, and trace back their individual $v_T$ profiles, i.e. perform the reverse calculation from that illustrated in Figure 5. The authors are doubtless aware that it does not work. This should however also alert them to the weaker links in their approach.

The reader requires more detailed, unambiguous information. E*xactly* which cores does Table 2 describe? Fission tracks are created by nuclides with different masses, charges and energies, resulting in variable and anisotropic lengths. Are all tracks in one sample assumed to be the same? If not, are, e.g., the listed core lengths averages over all tracks in a single run? Must a set of related cores be assumed in order to allow for length variation, or is it an automatic outcome of the separation between the host and confined tracks and the locus of their intersections? How does length anisotropy emerge in an isotropic model? Are the cores perhaps scaled to different latent track lengths at the start; if so, how? Is the latent track envisaged as the traditional line segment of finite length, with variable $v_T$, but without structure, etching threshold or range deficit? Or are the $v_T$ models in Figure 1 somehow also compatible with recent observations of unetched tracks?

How is it that a model predicting step-etch data cannot be applied to single-step experiments? Most datasets indeed do not present lengths for consecutive etch times, but the model should nevertheless fit the data. If, on the other hand, the fits depend on the observation function of one analyst to the point that all other data are excluded from consideration, then what is the significance of four digit etch-rate estimates? How are the step-etch experiments dealt with? Are the emersion, rinsing of the etchant and etch products, and re-immersion in fresh etchant without effect?

The authors are modest in their replies *(this is a first step; all models are wrong but some are useful),* but this does not help their manuscript at all. To abuse their image: yesterday we stood at the edge of the precipice, but today we made a great leap forward. It would greatly benefit the reception of this manuscript if the authors addressed the questions that all trackers will puzzle about.

The authors are right that this is the first calculation of its kind. However, in our opinion, no-one will be interested unless they can bring themselves to address the readers' obvious concerns. This could turn this manuscript from a wild shot into a considered and considerate paper that professionals will *want to read* and will appreciate for its *thoughtful assessment* of its methods and results.

**Replies**

1. The *conceptual model* fitted to the Laslett et al. (1984; Figure 8) step-etch data (albeit not to strictly identical confined track samples) corresponds, in terms of the present manuscript, to an average constant-core model with $\Delta x_{Tmax}$ = 11.1 μm, $\Delta x_{Tmax-B}$ = 2.1 μm, $V_{Tmax}$ = 1.67 μm/s, and $V_B$ = 0.04 μm/s.

2. With some hesitation, the authors concur with the first reviewer that our geological context of the KTB is open to question. We wish that once, just once, a critic would come up with a single scientific fact.

3. Our last comment concerning the need for an appendix with model equations is not as bizarre as the authors think. More than 30 years ago one of us wrote a track-etch simulation program to investigate if the Dartyge et al. (1981) track model could account for some properties of etched confined tracks (it can). There was no host track and no distance to bridge to the confined track. The confined track itself was represented by a vector, with each position x corresponding to a fraction of a micron along the track that could be etched at a cost $\sim 1/v_T(x)$. This approach could handle discontinuous $v_T(x)$-profiles resulting from the local densities of extended and point defects, much more complex than those in the present manuscript, without needing to solve equations.

Freiberg, 17 December 2020,

R. Jonckheere
B. Wauschkuhn

---

## Referee Comment (RC4) · Andrew Gleadow (Referee) · 14 Jan 2021

Review of gchron-2020-31

Confined fission track revelation in apatite: how it works and why it matters Richard A. Ketcham and Murat T. Tamer

General Comments

This paper reports on new variable VT fission track etching models for apatite and considers in detail its implications for confined fission track length measurements and their application in thermochronology. The models involve a number of clearly acknowl-

edged simplifying assumptions, such as omitting the effects of anisotropic etching and annealing, but nonetheless provides a number of important insights into track etching behaviour and explanations for a range of previously observed phenomena. The title is an apt description of the content. The models are then applied to an existing empirical track length data set and achieve considerable success in explaining their characteristics. In my opinion the paper represents an important advance in our understanding of fission track etching behaviour with significant implications for practical applications in fission track thermochronology.

In essence, the paper explores the consequences of varying the track etching rate, VT, along a fission track rather than being constant along its entire length. It does not purport to be a fully comprehensive model of the detailed form and distribution of etched tracks in apatite, along the lines of the models being developed by Raymond Jonckheere and his colleagues, but rather studies whether variable VT produces first-order predictions that can be usefully compared with observations. In this, I think it has been remarkably successful and has made a strong case that such variability is both real and has a significant influence on measurements. The particular form of the simplified models considered are somewhat arbitrary but conform broadly to what is known about track structure in apatite and other minerals. In fact, because the models do not include the specific crystallographic characteristics of apatite, the implications could probably also be generalised to track etching in other minerals.

In order to understand the revelation of sub-surface confined tracks, the paper also considers the etching characteristics of surface semi-tracks from which they are progressively intersected in TINT measurements. The study represents the first quantitative exploration of the implications of this two-stage etching process, which has previously been understood only in the vaguest terms. This semi-track modelling component of the paper leads to its own conclusions, independently of their role in revealing confined tracks, such as the estimates made in 6.7 of the track counting efficiency for apatite. Doubtless these estimates could be refined by including crystallographic anisotropy in

more complex models but the first-order agreement with independent estimates is very encouraging.

Another outcome of modelling the joint development of semi-tracks and confined tracks is to show that the great majority of confined tracks are intersected at shallow depths (section 4.3, Fig 4), with the consequence that most confined tracks will necessarily lie at low dip angles in order to avoid intersecting the surface. This explains why the usual restriction to measuring only 'horizontal' confined tracks, usually taken to mean dips of <10°, actually includes a high proportion of all the available tracks. It is noted in line 158 that a confined track dip of 25° is close to the maximum observed in the Tamer and Ketcham 2020b data set and a similar observation is a significant feature of our own dip measurements on confined tracks in a range of samples (Li et al, 2018 Amer. Mineral. 103, 430) which show that ∼70% of confined tracks in a range of apatites have dips <10° and virtually all are <30°. The model also explains the observation of Li et al. that confined tracks with the greatest dips have slightly shorter mean lengths, probably due to a greater average intersection depth and later etching start times.

The first major conclusion of the paper that VT is indeed variable along the length of fission tracks is, I think, well-made and unlikely to be controversial, indeed such variability has long been accepted by many, but without any clarity as to the implications. The second group of conclusions about laboratory annealed tracks etching faster than either fossil tracks or freshly induced tracks is more surprising and unlikely to be readily accepted at face value. Given that these are based solely on the modelling of a very limited data set, and the simplifying assumptions involved, I would suggest at least a more cautious and tentative statement of these conclusions acknowledging such uncertainties. The result is certainly interesting and points to the need for further study to be certain that this is a general phenomenon and not an experimental or modelling artefact.

One very important conclusion is that the very poor reproducibility of track length measurements in previous interlaboratory comparisons is probably centred on just one factor – the criteria used by analysts to determine which tracks are acceptably etched for measurement. Previously, the observed discrepancies could be attributed to a range of possible analytical, calibration and training factors, compounded by the difficulty of standardisation in track length measurements. The new model results, however, suggest that attention should be focused primarily on standardising track selection criteria in order to reach a much-needed concordance in length measurements across different laboratories.

Having said all this, I actually found the paper quite complex and surprisingly difficult to read in many places. I spent considerable time trying to understand some sections and some of the diagrams, despite considerable familiarity with the subject material. I think the overall readability needs to be improved if the paper is to be accessible to a wider audience. I make some specific suggestions below, but I think there are some general principles that could help.

First, a number of unfamiliar terms are used based on the geometry of the tracks which would be greatly clarified by an additional diagram where these are labelled. These terms include 'Impingement point' in Fig 2 (maybe 'Intersection point' would be clearer), the 'Relative semi-track' penetration and 'Confined track penetration' in Fig 3 and 'Intersection depth' (Fig 4). While these are defined at various places in the text, a diagram that showed them all in one place would greatly assist the reader in understanding the following diagrams. Perhaps this new key diagram might include a latent and etched semi-track and one or more confined tracks intersected at different stages of development, annotated with all the parameters used in the subsequent discussion.

Second, some basic information is simply missing such as the etchant used – presumably this was 5.5M HNO3, but this is not actually mentioned anywhere in the paper. It is not reasonable for the reader to have to look up the original source paper to know this, especially when etching conditions are actually part of the discussion and the etching times are central to the text. There is also no indication of how the track densities might have differed between the different experiments and which ones were Cf-irradiated,

which must exert a major control on some of the parameters, such as the number of intersections per track.

Third, the paper also assumes that the experiment codes (SE1, SE2 etc) are sufficient to alert the reader to the particular track characteristics involved. These are summarised in Table 1, but even there they do not appear to be complete. For example, only one of the experiments (SE3) in Table 1 is indicated to have involved Cf-irradiation, but it is apparent from the text that several others at least also involved this procedure. It is very hard to keep all of the different experimental details in mind when looking at the results, making them more difficult to comprehend. A good example is Figure 12, where histograms of the intersection depths are given for each experiment. There are clear differences between these, but it requires considerable work of detection to figure out how these relate to the track characteristics. Perhaps these histograms could be grouped under sub-headings or annotated with 'spontaneous', 'unannealed induced', 'annealed induced' so that commonalities in the different groups would be obvious. More explanatory and clearer figure captions would also help.

Another example is Figure 13 B and D, which show the etching start time for the modelled confined tracks, but the discussion arising from this diagram is about the track etching time thereby requiring the reader to make a mental inversion to make sense of the diagram. Why not simply plot the inverted track etching time (20 sec, minus the start time).

In summary, I believe that this paper is an important contribution and provides a number of significant insights into track etching behaviour that have important implication for the practice of fission track thermochronology. I recommend acceptance for publication after significant revision.

Specific Comments:

Line 105: 'Only simple modes are justifiable at . . .'

Line 125: '. . .shows the development of total confined track length. . .' (is this the intention?)

Line 139: '. . .of lengths, dips, and. . .'

Line 201: Is this 'power-law increase' arbitrary, based on the model, or derived from the actual measurements?

Line 206: '. . .corresponding case for standard track selection'. Also, what does 'standard track selection' mean here?

Line 207: I see how a selection criterion of VT/VB<=12 can be applied to the modelling results, but how can such a criterion be applied in practice in anisotropic crystals, where the form of the track tips are determined largely by fast and slow etching facets (Jonckheere et al 2019), rather than the idealised tip shapes in Fig 5.

Line 210: '. . .also predicts the intersection depth and etching. . .'

Line 211: '. . .distribution of the modelled track lengths. . .', also 'It is clear that. . .'

Line 218: '. . .first a semi-track length distribution. . .'

Line 219: '. . .then a distribution of confined track lengths. . .'

Line 221: '. . .function is the reduced chi-squared. . .' (or 'a' – still a clumsy sentence)

Line 241-2: '. . .were exerting a disproportionate control. . .'

Line 242: '. . .for experiment SE2 feature a very similar mean. . .'

Line 243: The expression of this whole line is clumsy and difficult to understand.

Line 278-285: I am really not sure what this paragraph is trying to say.

Line 299: '. . .impingement happened first for each track, where more than one is present.'

Line 344: 'both revealed using Cf semi-tracks' – this is not indicated in Table 1.

Line 454-5: I have no problem in considering a change in etching protocol and I think achieving standardisation in this area should be a community-wide objective, but the aim of 'allowing tracks at all levels of annealing to etch more completely' is probably impossible to achieve in practice and the concept of 'fully etched tracks' likely to remain a mirage. The disparity in compositionally controlled etching rates is so great between grains in many samples that it is essentially impossible to achieve an identical degree of etching in each. I think standardised procedures and calibration of the degree of etching is a more achievable goal.

Andrew Gleadow

University of Melbourne, 14 January 2021

––––––––––––––––––––––––––––––

---

## Referee Comment (RC5) · Edward Sobel (Referee) · 19 Jan 2021

General comments I have read through both the manuscript and the 3 presently existing comments and replies (no reply to RC4 at this moment). I will try not to reiterate what the other reviewers have noted. I think that they have done a good job in addressing weaknesses in the ms, although not always in a manner which makes it easy to see how to revise the text. In general, the study has the potential to be an important work examining how confined track length and termination shape evolves as a function of etch time. It is clearly not the final answer to this topic, nor do the authors claim that it is. However, it is a complicated problem which has been studied for over 35 years, so

why expect a quick solution:-) Yes, progressive steps forward are useful. In this case, technology has made it possible to track track growth in ways that previous generations could not measure - this has lead to the ability to improve our understanding. We all (try to) build upon previous work - that is the goal! I certainly learned a great deal about how and where confined tracks form and lengthen - and how few of them are actually measurable. The discussion of how track tips become more defined as they become better etched and how defining these shapes may lead to better inter-operator agreement is an important point in the study.

Specific comments In general, I concur with the other reviewers that the text is challenging to read and needs significant rewriting to make the points easier to understand. It would help if the introduction provided a better roadmap of what will be presented in the ms. Renaming the sample numbers with abbreviations matching the experimental conditions might help the reader to keep track of what is being done. E.g., SE1 could be renamed IU1 (Induced Unannealed). Transition sentences explaining what is about to be presented - such as 'observed data will be compared to numerical results' - would help the reader to follow the text. Some figures would benefit from captions that stated whether the plots depict observed or modeled data. In general, the captions are too short and do not sufficiently explain what is being shown (and why). Better labelling of figures would also help. Rather than just writing text in the caption, place information on the individual panels as well. For instance - add "randomly oriented unannealed induced tracks" and 'Cf tracks' to the individual panels in fig. 3. I agree with Gleadow's comment that a figure showing all geometries and terms is required. A table defining all terms would also help. The study of annealing of fission tracks has been advancing in fits and starts for over 55 years. A fair number of references from this evolution are included. Most fission trackers have probably not read most of these papers nor are they deeply attuned to the unresolved problems. Yes, they should be aware; however, a more robust introduction to the open problems would be useful for many readers. The text is written in a very compact form, with the authors apparently assuming that the readers are quite familiar with the topic. I encourage them to expand the text. This

is a paper which can at present only be read by a specialist. Perhaps this is true of most scientific papers, and this is not a bad thing. However, this ms requires a large amount of prior knowledge because many points are taken for granted rather than being explained. The introduction should clearly explain what tracks are and how they etch (damage 1st, then bulk etching in 3D). A sentence about how tracks form wouldn't hurt either. At the moment, I think that many non-specialist readers wouldn't even finish reading the introduction. Note that by non-specialist, I mean a fission-tracker who isn't deeply into methodology. Repeating methodologic information from the experimental paper (Tamer and Ketcham, 2020b) would be useful - how exactly was the step-etching done? This is pretty fundamental for understanding this dataset. And as many institutions do not subscribe to Elsevier journals, it is not always a 1 minute task to get this reference.

The discussion is much easier to follow than the preceding sections.

59-60 More recent work has documented enhanced but continuously diminishing etching velocity in the region along tracks beyond where most current etching protocols reach (Jonckheere et al., 2017). Could it be that as etchant travels along a longer/deeper track, that the strength of the acid and hence the etch rate decreases? This is suggested by the difference in measured track length of 20 sec etching of SE3 (3 steps, 14.89 microns) versus TK20 (1 step, 14.43 microns) and SE2 (2 steps, 20 sec, 16.19 microns) versus SE1 (1 step, 20 sec, 15.77 microns). A parallel question - does the strength of all possible etchants (e.g., 1.6 vs 5.5 mol nitric acid) have the same relationship to both bulk etching rate and track revelation rate? Or is there a difference between these rates as the acid is changed? This is relevant for both slow versus fast etch acid recipes as well as changes in acid strength as it penetrates farther into the crystal. These questions likely reveal my ignorance about the etching literature; however, answers or guesses about these questions might be helpful for improving the ms, particularly as the average reader is probably as ignorant as I am about this topic. The 2nd point is partly addressed in 413-423.

87-88 There was no clear indication of vB varying with track orientation (Tamer and Ketcham, 2020b). This was commented on by other reviewers. Please add a comment in the text about this observation, similar to the response to reviewer.

92 In cases where it was difficult to determine if an intersection truly occurred due to interfering features, we conservatively included it. Following Green's comment and your reply, it would be useful to explain your explanation of conservative here.

129-130 Variation in impingement point alone is likely responsible for some component of the observed variation in track lengths. How many tracks would one have to measure for such variability to become irrelevant. I.e., is this concern real and important for actual data collection or is this just an issue for interpreting this data set?

180-185 Adding photos of actual track tips would help to explain this useful concept. Naturally, anisotropy is quite important here, but this approach provides a way to go forward for better defining criteria about when a track can be measured. The argument that operator's track identification choice is responsible for much of the length variability observed in the inter-lab comparison is a key result of this study.

207-208 The tracks have a range of tip development (Fig. 6D), and only selecting those with vT/vB $\leq$ 12 (Fig. 6E) results in an excellent match to the measured data (Fig 6F). Excellent seems overstated. Adding text to the caption to point out what one is supposed to notice (this agreement) would help.

480 efficiency suggests that a carefully controlled preheating step could greatly increase confined track numbers, potentially without affecting lengths and thus paleothermal information Potentially is the key word in this sentence. If it did affect lengths, wouldn't this be fatal? It seems unlikely that the length reduction could be sufficiently well-constrained, particularly for detrital apatites, which are more likely to exhibit kinetic variability.

fig 2 caption - please add definition of xint in caption to help reader understand the

figure. It is not defined in the text. It appears to mean the position where the track began to etch / was intersected by a semi-track. The reader shouldn't have to interpret how a parameter is defined - it must be clearly stated. Ah - it is defined on line 159 - far too late. It is quite hard to see the difference between the 2 sets of figures; therefore, the difference should be described in the figure caption. If there isn't a real difference, then say so in the caption so that the reader doesn't get frustrated.

Fig. 8 is quite hard to understand. Write a useful caption! Fig. 9 shows 15 plots. The caption says nothing. Do you really expect the reader to look at this and understand your point? Ok, you have plotted 5 different data sets, each with the same 3 cross-plots. Conclusions? What is significant? Yes, if I have lots of time, I can try to figure out your point, and remind myself what are the differences between these 5 sets. However, if you want readers to understand your points, it would be better if you guide them. Otherwise, many people will give up and get nothing from at least this part of the paper. And ultimately, articles which demand too much patience by the reader are not yet ready to be published. Please increase your font sizes. Look at the text below the color scale on this figure. Do you really think that it can be read (including the subscript) without strong magnification?

Technical corrections 105 Only simple models justifiable at this point because our data consist of a very limited number of experiments. Add 'are' 150 Figure 3 shows examples of penetration and revelation rate. add 'calculated' 178 which will develop as a function etching velocity add 'of the' 199-200 Etching time must be greater than 3 s, after which track observability the probability of selecting a track is represented as $((L - 4.5)/5.5)3$; A word or 2 are missing here 220 The merit function is reduced chi-squared () This is too brief -for instance: We used a reduced chi-squared () value for the merit function. 230 Thus, we simultaneously fit data sets SE1 and SE2, At a minimum, point to table 1, so the reader can be reminded what SE1 and SE2 are. 233-234 Similarly, we co-fit SE3 with a single-step 20s measurement of unannealed fossil tracks from Tamer and Ketcham (2020a). Please name this dataset here (TK20). 236 After several

trials, we settled on a vT/vB of 12, why? 242-244 adding commas to this sentence might help to make it understandable on the 1st reading. 270 add micron after ∼0-0.2 Table 1 notes need to define more column headers. Fig. 11 - Explain what velocity gradient means in the caption. What is the database here? Fig. 13 - What do lighter and darker bars on histogram mean? Presumably the same as a similar fig. 6. State that - don't just expect the reader to remember (or to have read the entire ms).

Edward Sobel Universitaet Potsdam, Jan. 19, 2021

---

## Referee Comment (RC6) · Paul. Green (Referee) · 21 Jan 2021

The comments by different Reviewers of this ms show a high degree of consistency in highlighting a large number of issues, particularly emphasising the poor standard of presentation associated with this manuscript in terms of the lack of explanation in both the text and Figure captions. It should not be necessary for a reviewer to read a number of other papers in order to review a new paper, but in this case it is not possible to follow the work from the ms alone. I spent way too much time in reviewing this ms and I'm sure the other reviewers did too. I particularly congratulate Ed Sobel on his thorough and thoughtful comments. My overall response to the ms is to ask what

practical consequence this work could have. I understand that any progress in under-
standing the fundamentals of track revelation is useful in principle, but at the end of the
day, in practical application we have a grain mount containing various apatite species
with different etch rates, all etched together, for one etch time and with one etchant.
If analytical data in unknowns and age standards and calibration data for kinetic mod-
els are generated using the same conditions, then reliable results should be obtained.
This raises a huge number of practical questions, but they are not directly relevant to
the present discussion. One issue that still puzzles me is the repeated assertion, in
the paper under review and in earlier papers in the chain, that VB is not anisotropic.
Surely the etch figures in a prismatic surface show that the etch rate is higher along the
c-axis than perpendicular to it. Can anyone explain this conundrum? I see little point
in responding to most of the authors' comments on my review. The consistent opinion
of all reviewers is that this is an extremely poorly presented manuscript, and requires
significant revision before it could be considered acceptable for publication. Perhaps
if the authors take all the comments into account the final product may eventually be
easier to understand and appreciate. However I would like to respond to one comment:
In their response to my Review, Ketcham and Tamer suggest that I was misdirected in
my criticism of the Wauschkuhn et al. (2015) study. I was not. In my review I dis-
cussed the basic motivation of the Wauschkuhn et al. study, in terms of their ideas
concerning the KTB borehole (as expressed in the title of their paper). I was not dis-
cussing the particular aspect that Ketcham and Tamer refer to in their comments, as
illustrated in Figure 15 of Wauschkuhn et al., casting doubt on the principle of equiv-
alent time. As I was reviewing the ms by Ketcham and Tamer I did not see the need
to provide additional comments on any other aspect of the Wauschkuhn et al. study,
apart from their basic premise. In regard to the evidence in Figure 15 of Wauschkuhn
et al. (2015), the data presented there appear to fully support the validity of equivalent
time. My reading of that Figure is that the induced tracks that were pre-annealed do
not begin to start shortening again until heated at a temperature above that used in
the initial treatment. At higher temperatures, both induced and pre-annealed induced

populations give similar track lengths, which is just what is predicted from equivalent time. Regarding the comparison with spontaneous tracks, Wauschkuhn et al. (2015) acknowledge that they cannot be compared directly with induced tracks because "the fossil track population is not a single-length population, and contains somewhat shorter and somewhat longer tracks than a population of induced tracks pre-annealed to the same mean length". Duddy et al. (1988) provided clear experimental evidence confirming the validity of equivalent time, based on measurements of mean track length in apatites that had undergone various variable temperature annealing treatments. In common with Wauschkuhn et al. (2015), Ketcham and Tamer in the paper under review fail to mention this and proceed to cast doubt on the concept. In their response to my review, Ketcham and Tamer state "The experiment in Wauschkuhn was designed very specifically to test the equivalent time hypothesis in a way Green (1988) did not." This comment is either deliberately misleading or betrays a basic lack of understanding. The Green (1988) study was not designed to test equivalent time. The Duddy et al. (1988) study was designed to test equivalent time and the concept passed with flying colours. It is unacceptable to cast doubt on the validity of equivalent time without citing published evidence that clearly validates the concept. The fact that they discussed Duddy et al. (1988) in a previous paper, as noted by Ketcham and Tamer in their Response, is irrelevant to the present discussion, and only highlights the lack of full documentation in the ms under review.

Paul Green. January 2021.

---

## Referee Comment (RC7) · Raymond Jonckheere (Referee) · 25 Jan 2021

**TECHNISCHE UNIVERSITÄT BERGAKADEMIE FREIBERG**

[Figure]

Two remarks related to the author-reviewer discussion around gchron-2020-31

**Confined fission track revelation in apatite: how it works and why it matters**

Richard A. Ketcham and Murat T. Tamer

The new comments on the above manuscript prompt us to respond to two issues connected to our work.

1. P. Green (21.01.21): "*One issue that still puzzles me is the repeated assertion, in the paper under review and in earlier papers in the chain, that $V_B$ is not anisotropic. Surely the etch figures in a prismatic surface show that the etch rate is higher along the **c**-axis than perpendicular to it. Can anyone explain this conundrum?*".

Although it is indeed an issue in several reviews, we believe that it is not a major problem. The isotropic bulk etch rate ($V_B$ = 0.022 ± 0.004 μm s$^{-1}$) comes from (anneal)-etch-anneal-etch experiments (Tamer and Ketcham, 2020)[1]. After a first 10 s etch, the remaining unetched damage was annealed, after which the tracks were re-etched (+10 + 5 s). The rate of track length increase was assumed to be twice the bulk etch rate in the direction of the confined track. This is right by existing models of fission track etching. It is however wrong by established theories of crystal growth and dissolution. In contrast to $V_B$, a growth or dissolution (or etch) rate $V_R$ is the rate of displacement **of a crystallographic plane as a whole** in a perpendicular direction. It follows that concave forms, as in this case the ends of confined tracks, become bounded by the slowest-etching planes. In general, these correspond to the low-index planes, in apatite to the basal and prism planes (Figure 1). The basal and prism face both have an etch rate $V_R \approx 0.5$ μm min$^{-1}$ (Aslanian et al., 2020). Some geometry reveals that the rate of length increase of an *etched-annealed* track is then between (2×0.5) and (2×√2×0.5) μm min$^{-1}$ (0.017-0.024 μm s$^{-1}$; for Durango apatite etched in 5.5 M HNO$_3$ at 21 °C), depending on orientation. This is consistent with the minimum rate of length increase of *unannealed* tracks (Aslanian et al., 2020; Figure 7c). The higher values reported there are evidence of damage beyond the endpoint of the first etch step (30 s; 5.5 M HNO$_3$; 21 °C), i.e., damage which was annealed in Tamer and Ketcham (2020). It is also worth noting that, according to crystal growth and dissolution theories, convex forms like track-surface intersections become bounded by fast-etching faces (hence the etch pits). It follows that:

(1) One cannot use the length increase of fission tracks for measuring the bulk etch rate in the track direction.

(2) One cannot use the dimensions of track-surface intersections (*Dpar*; *Dper*) for estimating surface etch rates.
* * *
[1] EAE1-3; SE4-6 exhibit a similar trend between 20 and 30 s etching, which is however more difficult to understand.

(3) We leave aside the question of the significance that can be attached to the "roundness" of the track tips.

[Figure]

**Figure 1.** Calculated ends of etched confined tracks with various **c**-axis angles in an apatite prism face. The clover leaf is the envelope of the anisotropic etch rates $V_R$ of the crystallographic planes perpendicular to the etch rate vectors (leaf radii); the apatite **c**-axis runs from top to bottom (after Jonckheere et al., 2019; Figure 11).

2. P. Green (21.01.21): *"In regard to the evidence in Figure 15 of Wauschkuhn et al. (2015), the data presented there appear to fully support the validity of equivalent time. My reading of that Figure is that the induced tracks that were pre-annealed do not begin to start shortening again until heated at a temperature above that used in the initial treatment. At higher temperatures, both induced and pre-annealed induced populations give similar track lengths, which is just what is predicted from equivalent time.*

In our opinion, this reflects an increasing resistance to annealing but not the effect of equivalent time as such. Duddy et al. (1988, p. 25) write:

> *"[…] the 'principle of equivalent time' […] assumes that at any moment, a track which has been annealed to a certain degree […] behaves during further annealing in a manner which is **independent of the conditions which caused the prior annealing**, but which depends only on the degree of annealing that has occurred, and the prevailing conditions of temperature and time".*

The authors refer to Goswami et al. (1984,), who, about the independent pathway principle, as applied to track densities (s), write (p. 124):

> *"We […] assume that once a value s is reached, then **all annealing pathways that can lead to the value of s are equivalent**".*

Both these formulations are expressions of the Markov property that a future state depends on the present state and the future conditions but not on how the present state came about. Thus no conditions at all can be attached to the past, because the purpose of equivalent time is that the past may be ignored, *whatever it was*. In the case of the Wauschkuhn et al. (2015) experiment, the pre-annealing of the induced tracks might just as well have been for 10 s at 500 °C.

That said, the experiment does not contradict the Duddy et al. (1988) data or their interpretation (except that the principle is formulated in terms of *"a track"* - i.e., each track - whereas its proof rests on the mean track lengths). Wauschkuhn et al. (2015) also do not call into question its application to geological annealing of fossil tracks in T,t-path modelling. All that the result shows is that a population of induced tracks of a certain mean length undergoes less shortening than a population of fossil tracks of the same mean length. Strictly, this is indeed a violation of equivalent time, but not one that precludes its application to fossil *or* to induced tracks.

Wauschkuhn et al. (2015) admit that their fossil track population is not in all respects identical to that of pre-annealed induced-tracks. However, the equivalent time principle is expressed in terms of the (mean) track lengths, nothing else. In particular, the different origins of fossil and pre-annealed induced tracks (irradiation followed by one isothermal annealing step as opposed to accumulation, a track at a time, during variable geological annealing) are immaterial as long as the track lengths | mean lengths | length distributions (?) are the same. The pertinent empirical fact is that two *almost* identical track populations nevertheless do not anneal at similar rates.

Its significance is that it suggests a difference between fossil and induced tracks (Price et al., 1973; Gleadow et al., 1983; Durrani and Bull., 1987; Tamer and Ketcham, 2020). This could be pertinent to the length-density relationship of fossil tracks, the reliance on curvilinear equations to explain geological track lengths, and, failing sufficient curvature, the world-wide exhumation. At the outside, one could speculate that the apparent increasing resistance to annealing, producing fanning Arrhenius diagrams, is in fact a side-effect of an increasing track etch rate $v_T$. Before dismissing the Wauschkuhn et al. (2015) result we should at least repeat the experiment.

Freiberg, 25 January 2021,

R. Jonckheere
B. Wauschkuhn

---

## Author Comment (AC3) · 13 Feb 2021

**Final author response on manuscript Confined fission track revelation in apatite: how it works and why it matters**

Richard Ketcham and Murat Tamer

13 February 2021

The main thrust of the combined comments is that we need to spend more time setting the stage for the problem, including presenting the Tamer and Ketcham (2020) data and discussing more previous work on etching rates and why they are thought to vary, including etching rate being linked to the energy loss rate of the fission particle. We also need to more thoroughly describe how our model works, including a summary figure. In our initial submission we tried to go quickly through these initial parts to keep the paper relatively short, but evidently too quickly and not carefully enough. Other good suggestions are photos of developing track tips, and exploring model predictions on the dip distribution of confined tracks. All told, we anticipate this will require considerable new text, and 4-5 new figures, resulting in a substantially longer contribution, but we believe a more accessible and understandable one.

We have responded to the initial comments by Jonckheere and Green previously. We have transferred all subsequent comments below (except the final one by Jonckheere and Wauschkuhn (RC7), which is addressed to Green's remarks and not our contribution), and our answers are in *red italics*.

**Answers to final comments by Raymond Jonckheere and Bastian Wauschkuhn (RC3)**

Our essential comments are contained in our review (RC2), to which the authors have replied (AC2). The manuscript discussion forum invites us, or at least allows us, to react to these re-plies. In the absence of comments from other scientist, this could result in an endless back and forth between authors and reviewers until a consensus is reached or the deadline expires and the editors must arbitrate. It appears to us that this could be an exhausting and unproductive process, as it is most improbable that agreement will ever be reached on what is, or is thought to be, right or wrong. Since our scientific differences are not the real issue, we *recommend that the manuscript be considered for publication*, with such corrections as the authors and editors think necessary.

*We are gratified that the referees recommend consideration for publication after corrections.*

Nevertheless, we one last time wish to draw attention to some points, unrelated to scientific content. Various issues contribute to the fact that the manuscript makes an unfavourable impression on an attentive scientist. There is an apparent disdain for the reader, who is expected to guess at the intended meanings of word strings that appear to have been made up on the run, with indifference. This could be avoided by rewriting the manuscript with the aim to be understood.

It is inevitable that readers will evaluate the approach followed in this manuscript as a quick fix. The claim that the goal is to determine track etch rates is unconvincing. Price et al. (1973) and Green et al. (1978) performed *actual detailed measurements of $v_T$* along ion tracks in minerals more difficult to investigate than apatite. Furthermore, suspicious emphasis is placed on incidental geometrical observations. The road to $v_T$ is paved with precarious assumptions and precipitous leaps. The most unlikely and limited data (the mean lengths of etched confined tracks accessed via surface-intersecting host tracks and an intervening apatite segment) are used for estimating the model parameters. The resulting $v_T$ profiles have at best the most tenuous of connections with real fission-track or apatite properties. It cannot be otherwise, considering how they are arrived at. The computer-generated $v_T$-models are used to calculate etched-track geometries, of which a subset is selected for calculating the mean track lengths. Instead of that, the calculated geometries could have been compared with countless images of step-etched tracks, eliminating the need for inventing selection criteria. Within the current concept, it would in fact be most efficient to start from observed track geometries, and trace back their individual $v_T$ profiles, i.e. perform the reverse calculation from that illustrated in Figure 5. The authors are doubtless aware that it does not work. This should however also alert them to the weaker links in their approach.

*We are also grateful for their comments on pertinent prior work on documenting etching rates (Price et al. 1973; Green et al. 1978), and will include this work in a revised introductory section that will better set the stage for the work we present. It appears that the reviewers are referring to approaches their research group is using, which we continue to see as complementary.*

The reader requires more detailed, unambiguous information. E*xactly* which cores does Table 2 describe? Fission tracks are created by nuclides with different masses, charges and energies, resulting in variable and anisotropic lengths. Are all tracks in one sample assumed to be the same? If not, are, e.g., the listed core lengths averages over all tracks in a single run? Must a set of related cores be assumed in order to allow for length variation, or is it an automatic outcome of the separation between the host and confined tracks and the locus of their intersections? How does length anisotropy emerge in an isotropic model? Are the cores perhaps scaled to different latent track lengths at the start; if so, how? Is the latent track envisaged as the traditional line segment of finite length, with variable $v_T$, but without structure, etching threshold or range deficit? Or are the $v_T$ models in Figure 1 somehow also compatible with recent observations of unetched tracks?

*Our revised introduction/background will put our linear and constant-core models into the context of variable particle energies and stopping characteristics, which tend to feature first gradual and then faster decreases in energy loss rate of the fission particles, which have been hypothesized (e.g. Price et al 1973) as being linked to along-track etch rate.*

How is it that a model predicting step-etch data cannot be applied to single-step experiments? Most datasets indeed do not present lengths for consecutive etch times, but the model should nevertheless fit the data. If, on the other hand, the fits depend on the observation function of one analyst to the point that all other data are excluded from consideration, then what is the significance of four digit etch-rate estimates? How are the step-etch experiments dealt with? Are the emersion, rinsing of the etchant and etch products, and re-immersion in fresh etchant without effect?

*Our model can be, and is, applied to single-step experiments (e.g. experiments TK20; EAE1,2,3), but single-step experiments do not contain enough information to constrain the model, unless perhaps one had a lot of different single-step experiments etched for different times. We already discuss the experimental uncertainties, and report appropriate confidence intervals.*

The authors are modest in their replies *(this is a first step; all models are wrong but some are useful),* but this does not help their manuscript at all. To abuse their image: yesterday we stood at the edge of the precipice, but today we made a great leap forward. It would greatly benefit the reception of this manuscript if the authors addressed the questions that all trackers will puzzle about.

The authors are right that this is the first calculation of its kind. However, in our opinion, no-one will be interested unless they can bring themselves to address the readers' obvious concerns. This could turn this manuscript from a wild shot into a considered and considerate paper that professionals will *want to read* and will appreciate for its *thoughtful assessment* of its methods and results.

**Replies**
1. The *conceptual model* fitted to the Laslett et al. (1984; Figure 8) step-etch data (albeit not to strictly identical confined track samples) corresponds, in terms of the present manuscript, to an average constant-core model with $\Delta x_{Tmax}$ = 11.1 µm, $\Delta x_{Tmax-B}$ = 2.1 µm, $V_{Tmax}$ = 1.67 µm/s, and $V_B$ = 0.04 µm/s.

*We don't believe these parameters strictly match the Laslett et al. (1984) model; we plotted up our model with these parameters assuming instantaneous access of the acid to the tracks (which we think they are assuming), and the result does not resemble the figure ($V_{Tmax}$=1.1 µm/s works better). There are a number of factors that make the Laslett data difficult to model, the largest one being that they measured TINCLEs, making it difficult to estimate when the acid would reach the track – undoubtedly faster than for TINTs, but still requiring an estimate of V-cleavage. The Laslett data may also include tracks in later etching steps that were unobservable during the early ones. As for the conceptual model, we believe that the reviewers are over-interpreting that they connected their data points with a curved line, while ignoring what is actually stated in the text.*

2. With some hesitation, the authors concur with the first reviewer that our geological context of the KTB is open to question. We wish that once, just once, a critic would come up with a single scientific fact.

3. Our last comment concerning the need for an appendix with model equations is not as bizarre as the authors think. More than 30 years ago one of us wrote a track-etch simulation program to investigate if the Dartyge et al. (1981) track model could account for some properties of etched confined tracks (it can). There was no host track and no distance to bridge to the confined track. The confined track itself was represented by a vector, with each position x corresponding to a fraction of a micron along the track that could be etched at a cost $\sim 1/v_T(x)$. This approach could handle discontinuous $v_T(x)$-profiles resulting from the local densities of extended and point defects, much more complex than those in the present manuscript, without needing to solve equations.

*These replies are conversational in nature, and do not require responses, at least in the context of evaluating this manuscript. We do note that we included in this submission the complete computer codes for generating our results.*

**Answers to comments by Andrew Gleadow (RC4)**

General Comments

This paper reports on new variable VT fission track etching models for apatite and considers in detail its implications for confined fission track length measurements and their application in thermochronology. The models involve a number of clearly acknowledged simplifying assumptions, such as omitting the effects of anisotropic etching and annealing, but nonetheless provides a number of important insights into track etching behaviour and explanations for a range of previously observed phenomena. The title is an apt description of the content. The models are then applied to an existing empirical track length data set and achieve considerable success in explaining their characteristics. In my opinion the paper represents an important advance in our understanding of fission track etching behaviour with significant implications for practical applications in fission track thermochronology.

*We are grateful for the reviewer's endorsement.*

In essence, the paper explores the consequences of varying the track etching rate, VT, along a fission track rather than being constant along its entire length. It does not purport to be a fully comprehensive model of the detailed form and distribution of etched tracks in apatite, along the lines of the models being developed by Raymond Jonckheere and his colleagues, but rather studies whether variable VT produces first-order predictions that can be usefully compared with observations. In this, I think it has been remarkably successful and has made a strong case that such variability is both real and has a significant influence on measurements. The particular form of the simplified models

considered are somewhat arbitrary but conform broadly to what is known about track structure in apatite and other minerals. In fact, because the models do not include the specific crystallographic characteristics of apatite, the implications could probably also be generalised to track etching in other minerals.

*We again are grateful for the comment, and particularly for pointing out the benefits of our general approach for applying to other minerals.*

In order to understand the revelation of sub-surface confined tracks, the paper also considers the etching characteristics of surface semi-tracks from which they are progressively intersected in TINT measurements. The study represents the first quantitative exploration of the implications of this two-stage etching process, which has previously been understood only in the vaguest terms. This semi-track modelling component of the paper leads to its own conclusions, independently of their role in revealing confined tracks, such as the estimates made in 6.7 of the track counting efficiency for apatite. Doubtless these estimates could be refined by including crystallographic anisotropy in more complex models but the first-order agreement with independent estimates is very encouraging.

Another outcome of modelling the joint development of semi-tracks and confined tracks is to show that the great majority of confined tracks are intersected at shallow depths (section 4.3, Fig 4), with the consequence that most confined tracks will necessarily lie at low dip angles in order to avoid intersecting the surface. This explains why the usual restriction to measuring only 'horizontal' confined tracks, usually taken to mean dips of <10°, actually includes a high proportion of all the available tracks.

*We thank the reviewer for this important observation as well.*

It is noted in line 158 that a confined track dip of 25° is close to the maximum observed in the Tamer and Ketcham 2020b data set and a similar observation is a significant feature of our own dip measurements on confined tracks in a range of samples (Li et al, 2018 Amer. Mineral. 103, 430) which show that ~70% of confined tracks in a range of apatites have dips <10° and virtually all are <30°. The model also explains the observation of Li et al. that confined tracks with the greatest dips have slightly shorter mean lengths, probably due to a greater average intersection depth and later etching start times.

*These are interesting and pertinent points, but we note that, in the models we present, we actually limit the maximum dip to 25°, and we do not explore the relative preservation of tracks as a function of dip angle. In the revision we can explore this a little further, but are doubtful that we can fully explain this empirical observation, because there is likely to be an analytical bias against looking for, finding, and being able to confidently measure confined tracks at high dips.*

The first major conclusion of the paper that VT is indeed variable along the length of fission tracks is, I think, well-made and unlikely to be controversial, indeed such

variability has long been accepted by many, but without any clarity as to the implications. The second group of conclusions about laboratory annealed tracks etching faster than either fossil tracks or freshly induced tracks is more surprising and unlikely to be readily accepted at face value. Given that these are based solely on the modelling of a very limited data set, and the simplifying assumptions involved, I would suggest at least a more cautious and tentative statement of these conclusions acknowledging such uncertainties. The result is certainly interesting and points to the need for further study to be certain that this is a general phenomenon and not an experimental or modelling artefact.

*We believe we were cautious in our wording, but we can revisit it. However, we do note that the pattern of faster etching for annealed tracks does hold up over four unannealed and six annealed track experiments, each of which is essentially an independent test.*

One very important conclusion is that the very poor reproducibility of track length measurements in previous interlaboratory comparisons is probably centred on just one factor – the criteria used by analysts to determine which tracks are acceptably etched for measurement. Previously, the observed discrepancies could be attributed to a range of possible analytical, calibration and training factors, compounded by the difficulty of standardisation in track length measurements. The new model results, however, suggest that attention should be focused primarily on standardising track selection criteria in order to reach a much-needed concordance in length measurements across different laboratories.

*We agree.*

Having said all this, I actually found the paper quite complex and surprisingly difficult to read in many places. I spent considerable time trying to understand some sections and some of the diagrams, despite considerable familiarity with the subject material. I think the overall readability needs to be improved if the paper is to be accessible to a wider audience. I make some specific suggestions below, but I think there are some general principles that could help.

First, a number of unfamiliar terms are used based on the geometry of the tracks which would be greatly clarified by an additional diagram where these are labelled. These terms include 'Impingement point' in Fig 2 (maybe 'Intersection point' would be clearer), the 'Relative semi-track' penetration and 'Confined track penetration' in Fig 3 and 'Intersection depth' (Fig 4). While these are defined at various places in the text, a diagram that showed them all in one place would greatly assist the reader in understanding the following diagrams. Perhaps this new key diagram might include a latent and etched semi-track and one or more confined tracks intersected at different stages of development, annotated with all the parameters used in the subsequent discussion.

*We can add such a figure.*

Second, some basic information is simply missing such as the etchant used – presumably this was 5.5M HNO3, but this is not actually mentioned anywhere in the paper. It is not reasonable for the reader to have to look up the original source paper to know this, especially when etching conditions are actually part of the discussion and the etching times are central to the text. There is also no indication of how the track densities might have differed between the different experiments and which ones were Cf-irradiated, which must exert a major control on some of the parameters, such as the number of intersections per track.

*We can transfer this information from our previous paper, as also requested by other reviewers.*

Third, the paper also assumes that the experiment codes (SE1, SE2 etc) are sufficient to alert the reader to the particular track characteristics involved. These are summarized in Table 1, but even there they do not appear to be complete. For example, only one of the experiments (SE3) in Table 1 is indicated to have involved Cf-irradiation, but it is apparent from the text that several others at least also involved this procedure. It is very hard to keep all of the different experimental details in mind when looking at the results, making them more difficult to comprehend. A good example is Figure 12, where histograms of the intersection depths are given for each experiment. There are clear differences between these, but it requires considerable work of detection to figure out how these relate to the track characteristics. Perhaps these histograms could be grouped under sub-headings or annotated with 'spontaneous', 'unannealed induced', 'annealed induced' so that commonalities in the different groups would be obvious. More explanatory and clearer figure captions would also help.

*We can include this information.*

Another example is Figure 13 B and D, which show the etching start time for the modelled confined tracks, but the discussion arising from this diagram is about the track etching time thereby requiring the reader to make a mental inversion to make sense of the diagram. Why not simply plot the inverted track etching time (20 sec, minus the start time).

*We can consider this, but there is a reason for them being plotted this way which we could also make more clear: these diagrams are extensions of the set begun in Fig 4A,B and Fig 7A, in which each represents a subset of the previous one showing which intersected tracks remain valid and detectable for measurement.*

In summary, I believe that this paper is an important contribution and provides a number of significant insights into track etching behaviour that have important implication for the practice of fission track thermochronology. I recommend acceptance for publication after significant revision.

Specific Comments:

*Most of these are wording changes that we can address during revisions. We respond to the longer comments.*

Line 105: 'Only simple modes are justifiable at …'

Line 125: '…shows the development of total confined track length…' (is this the intention?)

Line 139: '…of lengths, dips, and …'

Line 201: Is this 'power-law increase' arbitrary, based on the model, or derived from the actual measurements?

*It is the result of a rough fitting process, which we can describe further.*

Line 206: '…corresponding case for standard track selection'. Also, what does 'standard track selection' mean here?

*Track selection as it would be done when attempting to measure "fully etched tracks"*

Line 207: I see how a selection criterion of VT/VB<=12 can be applied to the modelling results, but how can such a criterion be applied in practice in anisotropic crystals, where the form of the track tips are determined largely by fast and slow etching facets (Jonckheere et al 2019), rather than the idealised tip shapes in Fig 5.

*This consideration is why we explicitly and repeatedly leave the door open for a better summary parameter to be determined. However, that being said, although the tip forms might be "largely" determined by fast and slow etching directions, they will also be determined by vT(x) in a way not characterized by Jonckheere et al. (2019), who only depict the tips of fully-etched tracks. Essentially, vT(x) will determine when those facets at the track tips can start to develop.*

Line 210: '…also predicts the intersection depth and etching…'

Line 211: '…distribution of the modelled track lengths…', also 'It is clear that…'

Line 218: '…first a semi-track length distribution…'

Line 219: '…then a distribution of confined track lengths…'

Line 221: '…function is the reduced chi-squared…' (or 'a' – still a clumsy sentence)

Line 241-2: '…were exerting a disproportionate control…'

Line 242: '…for experiment SE2 feature a very similar mean…'

Line 243: The expression of this whole line is clumsy and difficult to understand.

Line 278-285: I am really not sure what this paragraph is trying to say.

*We can work on it. This is another case of first-ever-of-its-type data, and we can't fully explain it, so we want to leave a marker.*

Line 299: '…impingement happened first for each track, where more than one is present.'

Line 344: 'both revealed using Cf semi-tracks' – this is not indicated in Table 1.

Line 454-5: I have no problem in considering a change in etching protocol and I think achieving standardisation in this area should be a community-wide objective, but the aim of 'allowing tracks at all levels of annealing to etch more completely' is probably impossible to achieve in practice and the concept of 'fully etched tracks' likely to remain a mirage. The disparity in compositionally controlled etching rates is so great between grains in many samples that it is essentially impossible to achieve an identical degree of etching in each. I think standardised procedures and calibration of the degree of etching is a more achievable goal.

*We agree and disagree. Etching "more completely" is not an impossibility – "more" is quite relative and achievable. The idea, at its simplest, is to have a higher percentage of tracks be more fully etched, which our model indicates is achievable as expressed in some of our discussion about Figure 13.*

**Answers to comments by Edward Sobel (RC5)**

General comments I have read through both the manuscript and the 3 presently existing comments and replies (no reply to RC4 at this moment). I will try not to reiterate what the other reviewers have noted. I think that they have done a good job in addressing weaknesses in the ms, although not always in a manner which makes it easy to see how to revise the text. In general, the study has the potential to be an important work examining how confined track length and termination shape evolves as a function of etch time. It is clearly not the final answer to this topic, nor do the authors claim that it is. However, it is a complicated problem which has been studied for over 35 years, so why expect a quick solution :-) Yes, progressive steps forward are useful. In this case, technology has made it possible to track track growth in ways that previous generations could not measure - this has lead to the ability to improve our understanding. We all (try to) build upon previous work - that is the goal! I certainly learned a great deal about how and where confined tracks form and lengthen - and how few of them are actually measurable. The discussion of how track tips become more defined as they become better etched and how defining these shapes may lead to better inter-operator agreement is an important point in the study.

*We are grateful for the endorsement*

Specific comments In general, I concur with the other reviewers that the text is challenging to read and needs significant rewriting to make the points easier to

understand. It would help if the introduction provided a better roadmap of what will be presented in the ms. Renaming the sample numbers with abbreviations matching the experimental conditions might help the reader to keep track of what is being done. E.g., SE1 could be renamed IU1 (Induced Unannealed).

*A good suggestion that we will implement.*

Transition sentences explaining what is about to be presented - such as 'observed data will be compared to numerical results' – would help the reader to follow the text. Some figures would benefit from captions that stated whether the plots depict observed or modeled data. In general, the captions are too short and do not sufficiently explain what is being shown (and why). Better labelling of figures would also help. Rather than just writing text in the caption, place information on the individual panels as well. For instance - add "randomly oriented unannealed induced tracks" and 'Cf tracks' to the individual panels in fig. 3. I agree with Gleadow's comment that a figure showing all geometries and terms is required. A table defining all terms would also help. The study of annealing of fission tracks has been advancing in fits and starts for over 55 years. A fair number of references from this evolution are included. Most fission trackers have probably not read most of these papers nor are they deeply attuned to the unresolved problems. Yes, they should be aware; however, a more robust introduction to the open problems would be useful for many readers. The text is written in a very compact form, with the authors apparently assuming that the readers are quite familiar with the topic. I encourage them to expand the text. This is a paper which can at present only be read by a specialist. Perhaps this is true of most scientific papers, and this is not a bad thing. However, this ms requires a large amount of prior knowledge because many points are taken for granted rather than being explained. The introduction should clearly explain what tracks are and how they etch (damage 1st, then bulk etching in 3D). A sentence about how tracks form wouldn't hurt either. At the moment, I think that many non-specialist readers wouldn't even finish reading the introduction. Note that by non-specialist, I mean a fission-tracker who isn't deeply into methodology. Repeating methodologic information from the experimental paper (Tamer and Ketcham, 2020b) would be useful - how exactly was the step-etching done? This is pretty fundamental for understanding this dataset. And as many institutions do not subscribe to Elsevier journals, it is not always a 1 minute task to get this reference.

The discussion is much easier to follow than the preceding sections.

*All of these points are well taken. Among our concerns when writing this paper was that it was getting to be too long, imparting a different kind of length bias against people reading it, which is why we economized on some of our introduction and description. We also note that we received critiques both ways; the first comment by Jonckheere and Wauschkuhn took us to task for spending too much time explaining important angular biasing effects when we could have just given an*

*equation and referenced Dakowski (1978), an important but under-read paper. In our revision we will err on the side of explaining more.*

59-60 More recent work has documented enhanced but continuously diminishing etching velocity in the region along tracks beyond where most current etching protocols reach (Jonckheere et al., 2017). Could it be that as etchant travels along a longer/deeper track, that the strength of the acid and hence the etch rate decreases? This is suggested by the difference in measured track length of 20 sec etching of SE3 (3 steps, 14.89 microns) versus TK20 (1 step, 14.43 microns) and SE2 (2 steps, 20 sec, 16.19 microns) versus SE1 (1 step, 20 sec, 15.77 microns).

*This is an interesting idea, on which we are aware of no relevant data. We don't think this is a significant effect, however, because (a) we can and do explain all of these data without such a mechanism, as an outcome of track selection; (b) we see different etching rates within tracks of similar length but different origin (SE3 unannealed spontaneous versus SE6 lightly annealed induced); (c) our EAE experiments measure the same bulk etch rate for a 5-second versus 10-second etch.*

A parallel question - does the strength of all possible etchants (e.g., 1.6 vs 5.5 mol nitric acid) have the same relationship to both bulk etching rate and track revelation rate? Or is there a difference between these rates as the acid is changed? This is relevant for both slow versus fast etch acid recipes as well as changes in acid strength as it penetrates farther into the crystal. These questions likely reveal my ignorance about the etching literature; however, answers or guesses about these questions might be helpful for improving the ms, particularly as the average reader is probably as ignorant as I am about this topic. The 2nd point is partly addressed in 413-423.

*We believe that etchant strength does affect etching anisotropy (Dpar/Dper), and thus expect that it would also affect the relationship between bulk etch rate and along-track rate. However, we are not aware of a published study that has this information for apatite. We can add a cautionary note, but prefer not to speculate too much.*

87-88 There was no clear indication of vB varying with track orientation (Tamer and Ketcham, 2020b). This was commented on by other reviewers. Please add a comment in the text about this observation, similar to the response to reviewer.

*We will do this.*

92 In cases where it was difficult to determine if an intersection truly occurred due to interfering features, we conservatively included it. Following Green's comment and your reply, it would be useful to explain your explanation of conservative here.

*We will do this.*

129-130 Variation in impingement point alone is likely responsible for some component of the observed variation in track lengths. How many tracks would one have to measure for such variability to become irrelevant. I.e., is this concern real and important for actual data collection or is this just an issue for interpreting this data set?

*We don't really think it's an issue so much as an inevitable part of the data. No number of measurements would make it go away, unless, say, one screened tracks based on impingement point (which we don't think would be worth it).*

180-185 Adding photos of actual track tips would help to explain this useful concept. Naturally, anisotropy is quite important here, but this approach provides a way to go forward for better defining criteria about when a track can be measured. The argument that operator's track identification choice is responsible for much of the length variability observed in the inter-lab comparison is a key result of this study.

*We will compile such a figure.*

207-208 The tracks have a range of tip development (Fig. 6D), and only selecting those with vT/vB < 12 (Fig. 6E) results in an excellent match to the measured data (Fig 6F). Excellent seems overstated. Adding text to the caption to point out what one is supposed to notice (this agreement) would help.

*One does need to mentally rescale the dark blue histogram bars, but we think the match is pretty darn good…*

480 efficiency suggests that a carefully controlled preheating step could greatly increase confined track numbers, potentially without affecting lengths and thus paleothermal information Potentially is the key word in this sentence. If it did affect lengths, wouldn't this be fatal? It seems unlikely that the length reduction could be sufficiently well-constrained, particularly for detrital apatites, which are more likely to exhibit kinetic variability.

*The idea is that a heating treatment could possibly be found that results in no length reduction but a different etching rate. We don't think investigating this idea would be too difficult, as it's not hard to conduct experiments where one aliquot undergoes slight heating and the other does not. For example in the experiment from Wauschkuhn et al. (2015) that's already been discussed in this review process, heating at 1h for 175°C results in no reduction in mean spontaneous track length in Durango apatite. Might it be enough to accelerate etching? Potentially…*

fig 2 caption - please add definition of xint in caption to help reader understand the figure. It is not defined in the text. It appears to mean the position where the track began to etch / was intersected by a semi-track. The reader shouldn't have to interpret how a parameter is defined - it must be clearly stated. Ah - it is defined on line 159 – far too late. It is quite hard to see the difference between the 2 sets of figures; therefore, the difference should

be described in the figure caption. If there isn't a real difference, then say so in the caption so that the reader doesn't get frustrated.

*OK; actually, we can incorporate this definition into the figure requested by Andy Gleadow schematically illustrating the entire model.*

Fig. 8 is quite hard to understand. Write a useful caption! Fig. 9 shows 15 plots. The caption says nothing. Do you really expect the reader to look at this and understand your point? Ok, you have plotted 5 different data sets, each with the same 3 cross-plots. Conclusions? What is significant? Yes, if I have lots of time, I can try to figure out your point, and remind myself what are the differences between these 5 sets. However, if you want readers to understand your points, it would be better if you guide them. Otherwise, many people will give up and get nothing from at least this part of the paper. And ultimately, articles which demand too much patience by the reader are not yet ready to be published. Please increase your font sizes. Look at the text below the color scale on this figure. Do you really think that it can be read (including the subscript) without strong magnification?

*Admittedly, with those we are counting on the fact that this is being published in electronic format, where zooming is trivial; the possibly too-ambitious goal was to fit everything on one page. We'd be interest to hear form the journal editor on their recommendations in this respect.*

Technical corrections 105 Only simple models justifiable at this point because our data consist of a very limited number of experiments. Add 'are' 150 Figure 3 shows examples of penetration and revelation rate. add 'calculated' 178 which will develop as a function etching velocity add 'of the' 199-200 Etching time must be greater than 3 s, after which track observability the probability of selecting a track is represented as $((L - 4.5)/5.5)3$; A word or 2 are missing here 220 The merit function is reduced chi-squared () This is too brief -for instance: We used a reduced chi-squared () value for the merit function. 230 Thus, we simultaneously fit data sets SE1 and SE2, At a minimum, point to table 1, so the reader can be reminded what SE1 and SE2 are. 233-234 Similarly, we co-fit SE3 with a single-step 20s measurement of unannealed fossil tracks from Tamer and Ketcham (2020a). Please name this dataset here (TK20). 236 After several trials, we settled on a $vT/vB$ of 12, why? 242-244 adding commas to this sentence might help to make it understandable on the 1st reading. 270 add micron after ~0-0.2 Table 1 notes need to define more column headers. Fig. 11 - Explain what velocity gradient means in the caption. What is the database here? Fig. 13 - What do lighter and darker bars on histogram mean? Presumably the same as a similar fig. 6. State that - don't just expect the reader to remember (or to have read the entire ms).

*We can implement these corrections.*

**Answers to further comments by Paul Green (RC6)**

The comments by different Reviewers of this ms show a high degree of consistency in highlighting a large number of issues, particularly emphasising the poor standard of presentation associated with this manuscript in terms of the lack of explanation in both the text and Figure captions. It should not be necessary for a reviewer to read a number of other papers in order to review a new paper, but in this case it is not possible to follow the work from the ms alone. I spent way too much time in reviewing this ms and I'm sure the other reviewers did too. I particularly congratulate Ed Sobel on his thorough and thoughtful comments.

*We also note that Dr. Sobel was supportive of the importance of this work.*

My overall response to the ms is to ask what practical consequence this work could have. I understand that any progress in understanding the fundamentals of track revelation is useful in principle, but at the end of the day, in practical application we have a grain mount containing various apatite species with different etch rates, all etched together, for one etch time and with one etchant. If analytical data in unknowns and age standards and calibration data for kinetic models are generated using the same conditions, then reliable results should be obtained.

*This depends on what one means by "reliable" and "should." Recent studies have documented quite thoroughly that using the same conditions does not necessarily lead to the same results, and our work provides a testable answer as to why, which is a very useful first step in solving the problem – a very practical consequence. Another practical consequence is providing a potential quantitative basis for optimizing etching, rather than assuming that we learned everything we need to know 40 years ago. A final practical consequence is to improve our ability to construct a physically based, rather than empirical, understanding of annealing. As Andy Gleadow cogently pointed out in his review, confined track revelation "has previously been understood only in the vaguest terms," and a vague understanding of the principal observational data is not conducive to developing a rigorous understanding of what those data say.*

This raises a huge number of practical questions, but they are not directly relevant to the present discussion. One issue that still puzzles me is the repeated assertion, in the paper under review and in earlier papers in the chain, that VB is not anisotropic. Surely the etch figures in a prismatic surface show that the etch rate is higher along the c-axis than perpendicular to it. Can anyone explain this conundrum?

*In their response (RC7), Jonckheere and Wauschkuhn provide an explanation based on etching theory. For our part, we could object to what appears to be a repeated willful misrepresentation – we never claim that VB is not anisotropic, we just state that our experiment showed no clear indication of anisotropy.*

I see little point in responding to most of the authors' comments on my review. The consistent opinion of all reviewers is that this is an extremely poorly presented

manuscript, and requires significant revision before it could be considered acceptable for publication. Perhaps if the authors take all the comments into account the final product may eventually be easier to understand and appreciate. However I would like to respond to one comment: In their response to my Review, Ketcham and Tamer suggest that I was misdirected in my criticism of the Wauschkuhn et al. (2015) study. I was not. In my review I discussed the basic motivation of the Wauschkuhn et al. study, in terms of their ideas concerning the KTB borehole (as expressed in the title of their paper). I was not discussing the particular aspect that Ketcham and Tamer refer to in their comments, as illustrated in Figure 15 of Wauschkuhn et al., casting doubt on the principle of equivalent time. As I was reviewing the ms by Ketcham and Tamer I did not see the need to provide additional comments on any other aspect of the Wauschkuhn et al. study, apart from their basic premise. In regard to the evidence in Figure 15 of Wauschkuhn et al. (2015), the data presented there appear to fully support the validity of equivalent time. My reading of that Figure is that the induced tracks that were pre-annealed do not begin to start shortening again until heated at a temperature above that used in the initial treatment. At higher temperatures, both induced and pre-annealed induced populations give similar track lengths, which is just what is predicted from equivalent time. Regarding the comparison with spontaneous tracks, Wauschkuhn et al. (2015) acknowledge that they cannot be compared directly with induced tracks because "the fossil track population is not a single-length population, and contains somewhat shorter and somewhat longer tracks than a population of induced tracks pre-annealed to the same mean length". Duddy et al. (1988) provided clear experimental evidence confirming the validity of equivalent time, based on measurements of mean track length in apatites that had undergone various variable temperature annealing treatments. In common with Wauschkuhn et al. (2015), Ketcham and Tamer in the paper under review fail to mention this and proceed to cast doubt on the concept. In their response to my review, Ketcham and Tamer state "The experiment in Wauschkuhn was designed very specifically to test the equivalent time hypothesis in a way Green (1988) did not." This comment is either deliberately misleading or betrays a basic lack of understanding. The Green (1988) study was not designed to test equivalent time. The Duddy et al. (1988) study was designed to test equivalent time and the concept passed with flying colours. It is unacceptable to cast doubt on the validity of equivalent time without citing published evidence that clearly validates the concept.

*We find this critique unwarranted. Not all tests are by design (e.g., "she was tested in battle"). As we said in our earlier response, and as Jonckheere and Wauschkuhn say in their follow-up comment, Duddy et al. (1988) only tested the equivalent time hypothesis on induced tracks, not spontaneous ones. We did not imply that the experiments in Green (1988) were designed to test equivalent time, but they were the first experimental cross-validation of confined length annealing between spontaneous and induced tracks, so they were a test all the same. Moreover, it is a test the hypothesis appeared to pass, as the outcome led to the recommendation by Green (1988) to*

*normalize spontaneous lengths with unannealed induced lengths, supporting the idea that the two types of tracks can be considered interchangeably. Wauschkuhn et al. (2015) provided a more direct test, and as discussed in the J&W follow-up reply (RC7), the equivalent time concept as originally stated did not pass it. We are aware of one other unpublished study that repeats the Wauschkuhn et al. (2015) result, and our present study suggests a new distinction between fresh induced tracks annealed in the laboratory and those that have evolved over millions of years. We concur with J&W that further follow-up is warranted.*

The fact that they discussed Duddy et al. (1988) in a previous paper, as noted by Ketcham and Tamer in their Response, is irrelevant to the present discussion, and only highlights the lack of full documentation in the ms under review.

**Comment on remarks by Jonckheere and Wauschkuhn (RC7)**

*These remarks concern Green's review, not our paper, and so we do not see a need to respond, other than to agree with them, and to thank them for saving us the effort with their thorough response.*

---

## Author Response (AR2)

**Ketcham and Tamer responses to reviewer for R1 (Edward Sobel)**

**Reviewer comments:**
I have thoroughly read the revised version of Ketcham and Tamer's 'Confined fission track revelation in apatite: how it works and why it matters'. It is still a slow read because I have to think a lot about what is being presented. That is certainly a function of my expertise. However, unlike my reading of the first submission, I almost always understood and appreciated the authors' point. Below I have noted a few places where minor text editing might help the next reader. In general, I was happy with how Ketcham and Tamer addressed the reviewers' comments. In particular, the new figs are quite helpful and the renamed samples are easier to keep track of. I think that this is a really thought-provoking study which forces grey hairs like myself to think hard about how the AFT method works and how it should be taught to new workers. I think that this will be a valuable paper for future workers as well as a motivation for additional studies.
Sincerely,
Edward Sobel

Figs 1, 2, 10a, 15 - please add tic marks on the axes.
Fig. 6 'Semi-track penetration and confined track revelation calculated at time steps of 0.2 s and depth steps of 0.2 885 µm. Lines correspond to relative penetration of semi-tracks and revelation of confined tracks at etching times every second from 1 to 20 s, ' (and text around 205).
I'm sorry, but I am having trouble following this. I don't quite understand what is relative semi-track penetration. Please add a 1 sentence explanation in the caption to help slow readers like me. Same issue for fig. 20.
fig 10 caption - nice fig! It would help if the caption noted that these are pairs of figures for each length - both ends. It took me a moment to figure this out because the lengths in the short etch times are so hard to see. 'when analyst is accepting most tracks found '
Perhaps when analyst accepts most tracks that are found
Fig. 11 caption: 'Contour diagrams of model predictions of unannealed induced fission tracks selected by the analyst for measurement etching the grain mount for 20 seconds. '
something doesn't seem right at the end of this sentence.
107-9 - please note the conditions used for full annealing.
309 - add 'to' forced to minimize
335 add micron after 0.1
Discussion - 1st pp - please cite the relevant figs to make it easier for the reader to look at the data.
378-9 this is 0.04-0.21 microns of additional length per intersection, yes? Please slightly rewrite.
599 please cite fig. 20e here.
609 end of sentence: 2 characters are corrupted on my pdf.

**Author responses:**
All requested corrections have been made.  We made a small number of additional changes to correct a mis-numbered figure reference, and slightly improve the wording in a few places, but none that changed the scientific content.